# Revisiting wind speed measurements using actively heated fiber optics: a wind tunnel study

Justus G.V. van Ramshorst[1,4], Miriam Coenders-Gerrits[1], Bart Schilperoort[1], Bas J.H. van de Wiel[2], Jonathan G. Izett[2], John S. Selker[3], Chad W. Higgins[3], Hubert H.G. Savenije[1], and Nick C. van de Giesen[1]

[1]Delft University of Technology, Water Resources Section, Stevinweg 1, 2628 CN Delft, The Netherlands
[2]Delft University of Technology, Geoscience and Remote Sensing, Stevinweg 1, 2628 CN Delft, The Netherlands
[3]Oregon State University, Biological and Ecological Engineering, 116 Gilmore Hall, Corvallis, Oregon 97331, USA
[4]University of Göttingen, Bioclimatology, Büsgenweg 2, 37077 Göttingen, Germany

**Correspondence:** Justus van Ramshorst (justus.vanramshorst@uni-goettingen.de)

**Abstract.** Near-surface wind speed is typically only measured by point observations. The Actively Heated Fiber-Optic (AHFO) technique, however, has the potential to provide high-resolution distributed observations of wind speeds, allowing for better spatial characterization of fine-scale processes. Before AHFO can be widely used, its performance needs to be tested in a range of settings. In this work, experimental results on this novel observational wind-probing technique are presented. We utilized a controlled wind-tunnel setup to assess both the accuracy and the precision of AHFO under a range of operational conditions (wind speed, angles of attack and temperature difference). The technique allows for wind speed characterization with a spatial resolution of 0.3-m on a 1-s time scale. The flow in the wind tunnel was varied in a controlled manner, such that the mean wind, ranged between 1 and 17 ms⁻¹. The AHFO measurements are compared to sonic anemometer measurements and show a high coefficient of determination (0.92-0.96) for all individual angles, after correcting the AHFO measurements for the angle of attack. Both the precision and accuracy of the AHFO measurements were also greater than 95% for all conditions. We conclude that the AHFO has potential to measure wind speed and we present a method to help for choosing the heating settings of AHFO. AHFO allows for characterization of spatially varying fields of mean wind. In the future, the technique could potentially be combined with conventional Distributed Temperature Sensing (DTS) for sensible heat flux estimation in micrometeorological/hydrological applications.

## 1 Introduction

This work presents the results of a wind tunnel study designed to test the novel Actively Heated Fiber-Optic (AHFO) (Sayde et al. (2015)) wind speed measurement technique in controlled airflow conditions. The primary aims of the experiment were to assess the directional sensitivity and signal-to-noise ratio of AHFO.

Wind speed is most commonly observed using in-situ point measurement techniques. As a result, the spatial distribution of field observations is limited. While it is possible to obtain distributed wind speed observations with remote sensing (e.g., Goodberlet et al. (1989); Bentamy et al. (2003)), the spatial resolution is too low for many micrometeorological applications.

Many field experiments assume Taylor's frozen flow hypothesis (Taylor (1938)) in order to estimate fluxes with similarity theory (e.g., Higgins et al. (2009); Kelly et al. (2009); Bou-Zeid et al. (2010); Patton et al. (2011)). However, similarity theory only holds for idealized homogeneous/stationary conditions, which are rarely met in practice, resulting in a model containing strong assumptions, which often leads to significant errors (Ha et al. (2007); Higgins et al. (2012); Thomas et al. (2012)). In real, non-idealized situations, even slight surface heterogeneities can lead to dramatic impacts on the spatial structure of the flow in the surface boundary layer. Further, even if perfect surface homogeneity was possible, other atmospheric (surface) conditions are often nonstationary (Holtslag et al. (2013)).

In the past decade, a new way to obtain spatially distributed measurements was introduced into environmental studies. High spatial resolution measurements could be used to directly check underlying spatial assumptions (e.g., full temperature and horizontal wind profiles) and would reduce the need for such assumptions in real-world cases. Distributed Temperature Sensing (DTS) technology measures temperature at high temporal and spatial resolution over distances of up to several kilometers by using Fiber Optic (FO) cables as sensors (Selker et al. (2006a); Selker et al. (2006b); Tyler et al. (2009)). High-end DTS can measure the temperature at a 1-s and 0.3-m resolution (Sayde et al. (2014)). The ability to report temperature at such high resolution has proven useful in many environmental studies (Selker et al. (2006a); Selker et al. (2006b); Tyler et al. (2008); Tyler et al. (2009); Steele-Dunne et al. (2010)), including atmospheric experiments (Keller et al. (2011); Petrides et al. (2011); Schilperoort et al. (2018); Higgins et al. (2018); Izett et al. (2019)). It has also been shown that it is possible to observe air temperature and thermal structure of near-surface turbulence with DTS (Thomas et al. (2012); Euser et al. (2014); Zeeman et al. (2015), Jong et al. (2015)).

In 2015, Sayde et al. (2015) introduced the AHFO technique where they aimed to use DTS to measure wind speed. The underlying concept of the proposed method is similar to that of a hotwire anemometer; however, instead of single point measurements, AFHO enables distributed measurements to be made at high spatial resolution. Instead of only passively measuring the temperature in the fiber (as is done with DTS), one segment of the cable is actively heated. The heated segment is positioned parallel to the unheated reference segment, with a small separation, in our case 0.1 m. The temperature difference between the heated and reference segment is measured, i.e., the heated fiber and the air temperature. The temperature difference between the cables depends on the energy input as well as on the wind speed of the ambient air, which determines the magnitude of the lateral heat exchange, through convective heat loss. By setting up an energy balance for the heated cable, one can estimate the magnitude of this convective heat transport, which leads to an estimate of the wind speed.

Results from a field study by Sayde et al. (2015) demonstrated promising performance of the AHFO technique, but they recommended further tests on two aspects to be performed in controlled airflow conditions. First, the heat transfer model assumes a flow normal to the axis of the fiber. Hence, non-normal angles of attack need to be accounted for by using directional sensitivity equations. Following the recommendations of Sayde et al. (2015) we tested different directional sensitivity equations from hotwire anemometry (Webster (1962); Hinze (1975); Perry (1982); Adrian et al. (1984)) in the controlled setting of

our experiment. Second, Sayde et al. (2015) highlight the importance of a sufficient signal-to-noise ratio when conducting measurements. They show that the temperature difference between the heated and reference segments gives a good estimate for this ratio. The influence of the directional sensitivity and the signal-to-noise ratio on the measurement accuracy and precision is investigated and the results are used to propose a method to estimate the precision for future experiments with AHFO, hence our work will improve the possibilities for successful application of AHFO in future field experiments.

Finally, in the future it will be interesting to perform outdoor tests with AHFO, for both micrometeorological and hydrological applications, as AHFO gives a lot of insights in spatially varying wind fields. AHFO can be especially interesting in non-homogenous field sites, like forests, which are already studied with other DTS applications (Schilperoort et al. (2018), Schilperoort et al. (2020)). Moreover, the ability to measure spatial varying wind fields has the potential to be useful for estimating sensible heat fluxes in a variety of atmosphere-vegetation-soil continuums, by applying Monin-Obukhov similarity theory (assuming no violation of its assumptions) to the measured vertical profile of the mean wind speed and temperature (Businger et al. (1971)).

An overview of the experimental setup is presented in Section 2, with the accuracy and precision of the AHFO experiments presented in Section 3. In Section 3.4, a method for estimating the precision of AHFO experiments is introduced, followed by a short note on future studies.

## 2 Experimental Set-Up and Methods

### 2.1 DTS and Signal-to-Noise ratio analysis

Based on the backscattered signal of a laser pulse inside fiber optic cables, a DTS machine measures temperature along a complete fiber optic cable (Selker et al. (2006a); Selker et al. (2006b)). A main source of noise in DTS data is white noise induced by the statistical variability in photon count from backscatter (optical shot noise). The white noise can be reduced by averaging over multiple measurements in either space or time, assuming the observed temperature is/stays (relatively) constant (van de Giesen et al. (2012)). Spatial resolution could be increased by making coils, however (sharp) bends could be a potential source of signal loss (Hilgersom et al. (2016)).

A sufficiently high signal-to-noise ratio is essential for measurement precision with DTS. In Sayde et al. (2015) it is shown that the signal-to-noise ratio can be described as: $(T_s - T_f)/T_{error}$, where $T_s$ and $T_f$ are the temperature (in K) of the heated cable segment and (unheated) reference segment (i.e., air temperature). Hence the signal-to-noise ratio is related to the $\Delta T$ (= $T_s - T_f$) and the measurement error of the DTS, $T_{error}$. A large $\Delta T$ is obviously desirable, however, $\Delta T$ cannot be increased infinitely. The power controller can only deliver a limited amount of power to heat the FO cable, which is especially relevant for the heating of long lengths of FO cable (i.e. several hundreds of meters of FO cable). Additionally larger temperature differences could cause that other ways of transferring energy (e.g., free convection, radiative heat loss and diffusion) become more dominant. The effect of $\Delta T$ is investigated by using three temperature differences during the experiment.

DTS temperature measurements contain a measurement error, which follows a normal distribution (Selker et al. (2006a)). With long FO cables this measurement error changes over the length of the cable and this error is also different for each DTS

machine. In this experiment a short FO cable is used, which is close to the calibration bath. Therefore, the measurement error is calculated based on the calibration baths, by taking the average of two baths where the mean standard deviation over the whole experiment is calculated. Given the fact the signal used is $\Delta T$, containing the difference of two temperature measurements of $T_s$ and $T_f$, $T_{error}$ becomes: $T_{error} = \sqrt{\sigma_{T_s}^2 + \sigma_{T_f}^2}$. In this experiment $\sigma_{T_s} = \sigma_{T_f}$, resulting in $T_{error} = \sigma_T \cdot \sqrt{2}$. In this experiment we used a single value, however in experiments with longer FO cables, one could calculate a $T_{error}$ changing along the cable (des Tombe et al. (2020)).

The effect of the signal-to-noise ratio is quantified, and an equation to estimate the precision is presented. The measurement precision is an indication of the variability of wind speed measurements (e.g., RMSD), as opposed to accuracy which describes a systematic measurement error for which can be compensated when using another device (in our case expressed by the bias).

## 2.2 Determination of Wind Speed

### 2.2.1 Original determination of Wind Speed by Sayde et al. (2015)

An energy balance is used to quantify the heat dissipation from the heated section, and therefore estimate the wind speed with DTS. The convective cooling can be converted to wind speed, because it is a function of wind speed and the temperature difference between the heated and unheated segments. The full energy balance (in W) for a cable segment volume of length, B, is given by Sayde et al. (2015), and schematically shown in Figure 1:

$$c_s \rho_v V \frac{dT_s}{dt} = P_s B + (\bar{S}_b + \bar{S}_d + \alpha_s \bar{S}_t)(1 - \alpha_f)2r\pi B + (\bar{L}_\downarrow + \bar{L}_\uparrow)\epsilon 2r\pi B - \epsilon\sigma T_s^4 2r\pi B - h(T_s - T_f)2r\pi B \tag{1}$$

Where, $r$ is the radius of the cable ($6.7 \cdot 10^{-4}$ m in our setup); $V$ is the volume of the cable segment ($\pi r^2 B$, in $m^3$), $c_s$ is the specific heat capacity of the FO cable (502 $Jkg^{-1}K^{-1}$) and $\rho_v$ is the FO cable density (800 $kgm^{-3}$). $P_s$ is the heating rate per meter of cable (in $Wm^{-1}$); and $B$ is the length of a cable segment (in m). $\bar{S}_b, \bar{S}_d$ and $\alpha_s \bar{S}_t$ (in $Wm^{-2}$) are the mean direct, diffuse and reflected short wave radiation fluxes, respectively, with $\alpha_s$ being the surface albedo of the ground; and $\alpha_f$ is the FO cable optic surface albedo. $\bar{L}_\downarrow + \bar{L}_\uparrow$ (in $Wm^{-2}$) are the average downward and upward longwave radiation fluxes, respectively; and $\epsilon$ is the FO cable surface emissivity. Based on the kind of stainless steel, emissivity values can range from 0.3 to 0.7 (Baldwin and Lovell-Smith (1992)); however, we assume a value of 0.5 (Madhusudana (2000)). $\sigma$ is the Stefan-Boltzmann constant, $5.67 \cdot 10^{-8}$ ($Wm^{-2}$ $K^{-4}$); and $\epsilon\sigma T_s^4$ is the outgoing longwave radiation of the fiber, i.e., $L_{fiber}$; $h$ is the convective heat transfer coefficient ($Wm^{-2}K^{-1}$).

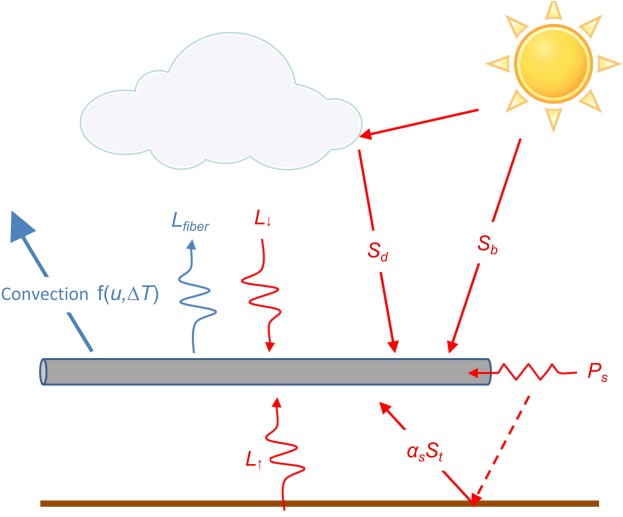

**Figure 1.** Schematization of the energy balance, from Sayde et al. (2015)

### Simplification

The energy balance is simplified, by dividing Eq. 1 by $2r\pi B$, which is equal to the surface area of the FO cable. The energy balance now no longer depends on $B$, meaning the length of FO cable segment does not need to be defined. The proposed final energy balance by Sayde et al. (2015) is as follows and in Wm$^{-2}$:

$$5 \quad \frac{c_s \rho r}{2}\frac{\mathrm{d}T_s}{\mathrm{d}t} = \frac{P_s}{2\pi r} + (\bar{S}_b + \bar{S}_d + \alpha_s \bar{S}_t)(1-\alpha_f) + (\bar{L}_\downarrow + \bar{L}_\uparrow)\epsilon - \epsilon\sigma T_s^4 - h(T_s - T_f) \tag{2}$$

where, $\rho$ is the FO cable density per meter of cable segment: 4.5 x 10$^{-3}$ kgm$^{-1}$.

### Convective heat transfer coefficient

The convective heat transfer coefficient $h$ (Wm$^{-2}$K$^{-1}$) can by means of the dimensionless Nusselt (Nu), Prandtl (Pr), and Reynolds (Re) numbers be expressed as function of the wind speed, $h = f(u_N)$. The Nusselt number is the ratio between the convective and conductive heat transfer, where the Nusselt number can be written as follows (Žukauskas (1972)):

$$\mathrm{Nu} = \frac{h d_s}{K_a} = C\mathrm{Re}^m \mathrm{Pr}^n \left(\frac{\mathrm{Pr}}{\mathrm{Pr}_s}\right)^{\frac{1}{4}} \tag{3}$$

with,

$$\mathrm{Re} = \frac{u_N d_s}{v_a} \tag{4}$$

$d_s$ is the fibers characteristic length ($2r$); $K_a$ is the thermal conductivity of air and $v_a$ the kinematic viscosity of air, respectively 0.0255 Wm$^{-1}$K$^{-1}$ and 1.5 x 10$^{-5}$ m$^2$s$^{-1}$ (Tsilingiris (2008)). $K_a$ and $v_a$ are assumed to be constant, due to the controlled conditions in the wind tunnel, but in field experiments this should be included as a variable, as $K_a$ and $v_a$ are temperature and relative humidity depend (Tsilingiris (2008)). $C$, $m$ and $n$ are empirical constants related to forced advection of heat by

air movement. In Sayde et al. (2015), $C$, $m$ and $n$ values of 0.51, 0.5 and 0.37 are set, based on (Žukauskas (1972)). Pr is the Prandtl number and can be seen as the ratio between kinematic viscosity and thermal diffusivity, which, we assume Pr to be constant (0.72) for our range of temperatures (12-35 °C), as in Tsilingiris (2008), with Pr$_s$ (the Prandtl number for the heated fiber segment), assumed to be the same as Pr, due to the small temperature differences (max. 6 K). Lastly, Re is the Reynolds number which is used to determine the flow regime of the air along the fiber segments, i.e., Re expresses if the flow regime is

laminar or turbulent. Combining Eq. 3-4 yields:

$$h = Cd^{m-1}\mathrm{Pr}^n \left(\frac{\mathrm{Pr}}{\mathrm{Pr}_s}\right)^{\frac{1}{4}} K_a v_a^{-m} u_N^m \tag{5}$$

The determination of the Nusselt number (Eq. 3) is only valid in the following ranges of Re (40-1000) and Pr (0.7-500). Re can be a limitation for higher wind speeds, especially when the diameter of the fiber is large, in our case wind speeds higher than approximately 11 ms$^{-1}$ would be out of range.

In the derivation of the energy balance (1), there is assumed to be no free convection, induced by heating of the air close to the cable, and no conduction of heat in the axial direction of the FO cable. It is also assumed there is no radiative exchange between objects close and parallel to the heated fiber, i.e., dispersion of heat from the heated cable to the reference cable is assumed to be negligible. Furthermore, a flow directed normal to the axis of FO cable is assumed by the proposed energy balance, i.e., for flow directed in a different angle, compensation is necessary to accurately estimate the wind speed.

**2.2.2  Revised simplified determination of Wind Speed**

Due to the setup inside the wind tunnel, as opposed to outdoor conditions, some simplifications can be made. The short wave radiation can be neglected because it is an indoor experiment (no sunlight). Furthermore, we assume that there is a in space uniform temperature inside the wind tunnel, due to the enclosed conditions. This means the incoming radiation is dependent on the air temperature, $T_f$. Assuming incoming ($\bar{L}_\downarrow + \bar{L}_\uparrow$) to be black body radiation (i.e., $L_{in} = \sigma T_s^4$), the net longwave radiation

loss for the fiber can be simplified accordingly by merging the incoming longwave and outgoing longwave radiation as:

$$(\bar{L}_\downarrow + \bar{L}_\uparrow)\epsilon - \epsilon\sigma T_s^4 \approx -\epsilon\sigma(T_s^4 - T_f^4) \tag{6}$$

One more additional change is made, based on our results obtained during testing of the performance of the AHFO technique. In processing of the obtained wind tunnel data it was found that by using the calculation of the Nusselt number from Žukauskas (1972), Eq. 3, a $\sim 20\%$ additional bias in calculating the wind speed was created. By using a more recent version for calculating

the empirical Nusselt number (Cengel and Ghajar (2014)), the bias in our study is reduced to $\sim 5\%$ Therefore, Eq. 7 is

proposed to calculate the Nusselt number, where the constants $C$, $m$ and $n$ are still used; however, with the values from Table 7-1 ($C = 0.683, m = 0.466$ and $n = 1/3$) in Cengel and Ghajar (2014), rather than those in Žukauskas (1972). Next to the improved fit, the range of Re over which the equation is valid is much wider (40-4000 compared with 40-1000), and therefore more applicable in future AHFO experiments.

$$\text{Nu} = C\text{Re}^m\text{Pr}^n = 0.683\text{Re}^{0.466}\text{Pr}^{1/3} \tag{7}$$

Consequently, the expression of $h$ changes as well.

$$h = Cd^{m-1}\text{Pr}^n K_a v_a^{-m} u_N^m \tag{8}$$

With the long- and short-wave radiation simplifications, the energy balance becomes:

$$\frac{c_s\rho r}{2}\frac{\mathrm{d}T_s}{\mathrm{d}t} = \frac{P_s}{2\pi r} - \epsilon\sigma(T_s^4 - T_f^4) - h(T_s - T_f) \tag{9}$$

By substituting the expression for $h$ (Eq. 8), we can rearrange Eq. 9 to obtain an expression for wind speed. Eq. 10 will be used to estimate the wind speed ($u_N$) in our wind tunnel study.

$$u_N = \left(\frac{0.5P_s\pi^{-1}r^{-1} - \epsilon\sigma(T_s^4 - T_f^4) - 0.5c_p\rho r\frac{\mathrm{d}T_s}{\mathrm{d}t}}{Cd^{m-1}\text{Pr}^n K_a v_a^{-m}(T_s - T_f)}\right)^{1/m} \tag{10}$$

## 2.3  Wind tunnel experiments

We conducted a series of experiments under tightly controlled airflow conditions to improve the applicability of AHFO in
experimental (field) research and to study the directional sensitivity and influence of the signal-to-noise ratio. The experiments presented were performed in a wind tunnel at Oregon State University (Low Speed Wind Tunnel of the Experimental Fluid Mechanics Research Lab - College of Engineering). This wind tunnel has a closed circuit, which means the air inside is recycled. The test section of the wind tunnel has a cross-section (height by width) of 1.23 by 1.52 m, and an undisturbed horizontal section of roughly 5 to 6 m which may be used for probing. During the experiment the heated and the unheated reference
cable segment were placed 8cm apart. The FO cable has two FO cores, hence, each cable segment could be sampled twice. For validation, an independent sonic anemometer (IRGASON+EC100 and CR3000, Campbell Scientific, Logan, UT,USA) was placed approximately 0.2 m downwind of the fibers, which measures the wind speed in 3 directions at 10 Hz. As the FO cables are very thin, it is assumed that these do not significantly disturb the measurement of the sonic volume (particularly at larger averaging times). All equipment was mounted using custom-designed support material.
The cable (AFL, Spartanburg, SC, USA) mounted in the wind tunnel consisted of a 1.34 mm outer diameter stainless steel casing that enclosed four multi-mode FO cores with a diameter of 250 µm (Figure A1). The electrical resistance per meter of

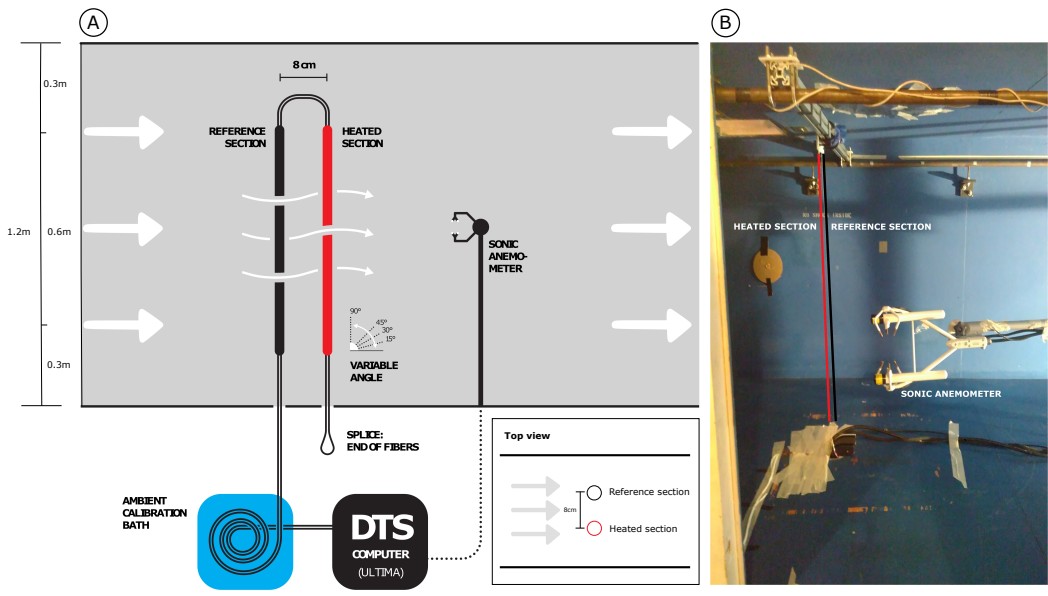

**Figure 2.** a) Schematic of the wind tunnel setup and b) photograph of the experimental setup in the wind tunnel.

stainless steel casing ($R_s$) is 1.67 ($\Omega$m$^{-1}$) and is constant along the length, where for the length of a cable segment ($B$, (m)), $R = R_s B$, where $R$ ($\Omega$) is the total resistance of a cable segment. Similary the heating rate is defined as $P_s = I^2 R_s$ (Wm$^{-1}$) per meter of cable segment, where $I$ (A) is the eletrical current. Only two FO cores were used and these were spliced at the end of the cable to create a duplexed FO core (using two FO cores in one cable), which results in double measurements for each measuring point along the FO cable, using a single-ended configuration (Hausner et al., 2011). Both the FO cores were connected to a Silixa Ultima DTS machine (Ultima S, 2 km range, Silixa, London, UK) outside the wind tunnel, however afterwards a single-ended configuration was used due to asymmetrical signal loss.

One cable segment was heated by connecting the stainless steel casing to a power controller (MicroFUSION uF1HXTA0-32-P1000-F040) by 12 AWG (copper) cables (3.31 mm$^2$), to heat the cable in a controlled way.

For calibration and validation of the DTS data, approximately 6 m of the FO cables was placed in a well-mixed ambient bath to calibrate the DTS temperature according to the single-ended method described by Hausner et al. (2011). The temperature was verified with one probe (RBRsolo$^2$ T, RBR Ltd., Ottawa, Ontario, Canada). A circulating aquarium pump was placed inside the bath, to prevent stratification.

In field experiments the wind speed and direction will vary, therefore different angles of attack and wind speeds are tested. Additionally different heating rates are used to quantify the importance of the signal-to-noise ratio. The following settings are used:

– **Angle of attack**: The cable was mounted at four different angles in the wind tunnel, resulting in different angles of attack to mean flow direction, in order to gain more insights into directional sensitivity. In Figure 2b the 90° set-up is visible,

however the cable was also mounted at a 45°, 30° and 15° angle, with respect to the floor of the wind tunnel (see: Figure 2a, inset). During all set-ups, the lower part of the FO cable was fixed to the opening in the bottom of the wind tunnel, while the upper end was attached to an extruded aluminum bar that was moved over the fixed horizontal bars, to achieve the desired cable angles.

– **Wind speed**: To test the performance for a range of wind speeds, ten different wind speeds were tested at every angle: 1, 3, 5, 7, 9, 11, 13, 15, 16 and 17 ms⁻¹. The AHFO wind speed measurements can be adjusted by comparing the AHFO wind speed to a reference sonic anemometer. The wind speed in the wind tunnel was fixed at a constant value to create a stable, non-turbulent, steady state flow (Appendix C).

– **Heating rate**: The magnitude of the current needed to create a given temperature difference is dependent on the cable

resistance and the wind speed, therefore the current is adjusted for each individual experiment. The current was fixed to create a temperature difference ($\Delta T$) of 2, 4 and 6 K between the heated and reference cable. Heating rates varied from 0.5-10 Wm⁻¹ during our setup.

In total, 120 (4 x 10 x 3) trials were conducted with the different parameters, each with a minimum duration of 10 minutes.

Temperatures along the FO cable were sampled at 0.125 m resolution with a sampling rate of 1 Hz. Splices connecting two

fiber optic cores are known to create an additional loss in signal, i.e., local higher attenuation (Tyler et al. (2009); van de Giesen et al. (2012)), this loss is normally independent of the direction. However, in our setup the signal loss of the splice connecting the fiber-optic cores of our cable at the end of the array was not the same in both directions. Due to this asymmetrical structure of the splice loss, only the data of one channel was used to ensure the quality of the results, as this channel showed a regular splice loss.

For each angle of attack only the 5 temperatures differences ($\times 2$ because of duplexed FO core) from the middle of the wind tunnel are used, to prevent using AHFO wind speed measurements with side/boundary effects. We investigated the consequences of extending the spatial range and found there is limited difference between these measurements (see Table D1). During this extended spatial range analysis we found out part of the 90° data contained additional noise which decreased the accuracy when everything was combined, and therefore we decided to take only 5 temperature differences for the 90°

calculations. Potential reasons for this additional noise could be the sharper bend for the 90° setup (Hilgersom et al. (2016)), also the FO cable is shorter for the 90° setup (due to the design of the setup), what means the fixations are closer to the middle of the cable causing local disturbances on the temperature measurements. In Table D2 an overview for the amount of measurements used for each setup is shown.

In our study we use the advantage of averaging over time and space, to reduce (white) noise in the DTS measurements

(van de Giesen et al. (2012); Selker et al. (2006b)). For clarity we therefore introduce three parameters: $n_{time}$, $n_{space}$ and $n$, where $n_{time}$ is the amount of measurements averaged over time and $n_{space}$ is the amount of measurements averaged over space and $n$ the total amount of measurements over time and space and can be expressed as: $n = n_{time} \times n_{space}$. In the machine specifications it is given that the sample resolution is $x_{sample} = 0.125$m. The highest actual spatial resolution is 0.3m, indicating a $n_{space} \geq 2$, according to the 10-90% rule as described in Tyler et al. (2009). In this paper we will first

work with $n_{space} = 10$ (for 90° $n_{space} = 5$) and finally we will propose an equation to predict the precision (See later Eq. 21) which is a function of $n_{space}$ and $n_{time}$. We first use $n_{space} = 10$, because for deriving the precision prediction an unique constant ($C_{DTS}$) is necessary. $C_{DTS}$ is derived from our measurements and can be used for predicting the precision in future experiments. $C_{DTS}$ is expected to be more accurate if the amount of (white) noise is reduced by averaging.

## 5  2.4  Directional sensitivity analysis

Equation 10 is derived for flows normal to axis of the cable (as in Figure 2b). Depending on the physical setup the wind will not always have a 90° angle compared to the axis of the cable, especially in outside atmospheric experiments. For angles smaller than 90° the wind speed will be underestimated, as the convective heat transfer is less efficient. While Sayde et al. (2015) adjusted the wind speed of the sonic anemometer using a geometric correction from hotwire anemometry (e.g., Adrian et al.
10  (1984)), we adjusted the measured DTS windspeed $u_N$ (eq. 10) to compare both wind speeds:

$$u_{DTS} = \sqrt{\frac{u_N^2}{\cos^2(\varphi - 90°) + k_{ds}^2 \sin^2(\varphi - 90°)}} \tag{11}$$

$k_{ds}$ is the directional sensitivity and $\varphi$ is the angle of attack of the wind with respect to the axis of the cable, ranging from 0° to 90°.

## 2.5  Accuracy and precision definition

15  The perfomance of our AHFO measurements will be assed by looking at the accuracy and precision. The accuracy ($\sigma_a$) is defined by the normalized difference of the AHFO and sonic anemometer wind speed measurements, Eq. 12.

$$\sigma_a(j) = \frac{\bar{u}_{DTS}(j) - \bar{u}_{sonic}(j)}{\bar{u}_{sonic}(j)} \tag{12}$$

Where $j$ is a specific wind speed setting, where $j = 1, 3, 5, 7, 9, 11, 13, 15, 16, 17$ ms$^{-1}$. And $\bar{u}$ is the average of all indivual measurements ($i$) for a given wind speed setting.
20  The precision ($\sigma_p$) is defined by the normalized RMSD between the AHFO and sonic anemometer wind speed measurements, 13.

$$\sigma_p(j) = \frac{\text{RMSD}}{\bar{u}_{sonic}(j)} = \frac{\sqrt{\sum_{i=1}^{k} \left( \left( u_{sonic}(i,j) - \bar{u}_{sonic}(j) \right) - \left( u_{DTS}(i,j) - \bar{u}_{DTS}(j) \right) \right)^2 \frac{1}{k(i)}}}{\bar{u}_{sonic}(j)} \tag{13}$$

## 3 Results and Discussion

### 3.1 Proposed directional sensitivity equation

During analysis of the wind tunnel data it was found that Eq. 11 was not giving satisfying results (e.g., a 22% bias between the 90° and 15° angle). In Adrian et al. (1984) it is shown that in hotwire anemometry a variety of theoretical and empirical formulas have been proposed in the past, in order to account for directional sensitivity. Alternatively, using the formula suggested by Bruun (1971) gives more satisfying results, diminishing the bias between the 90° and 15° angle to only a few percent. This is shown in the boxplot of Figure 3.

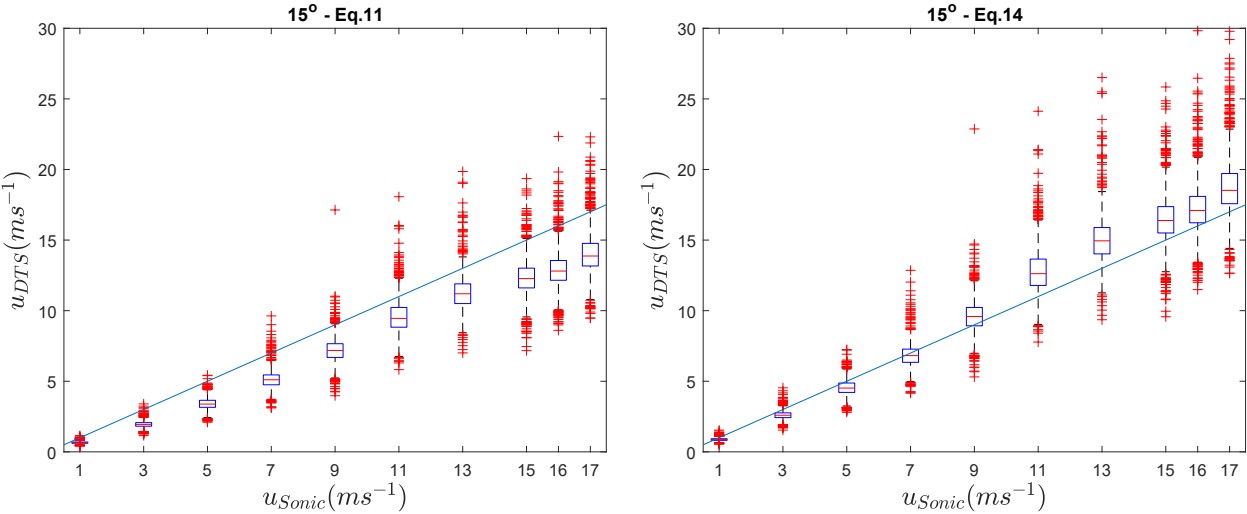

**Figure 3.** Directional sensitivity shown in boxplots for 15° angle, original Eq. 11 (a) and proposed Eq. 14 (b). The line represents the 1:1-line.

Therefore, Eq. 14 is used to account for directional sensitivity in our study, with the scaling exponent, $m_1$, able to be optimized during calibration of the AHFO measurements. The value for $m_1$ obtained during calibration of our set up was 1.05.

$$u_{DTS} = \frac{u_N}{\cos(\varphi - 90°)^{m_1}} \tag{14}$$

### 3.2 Accuracy and precision

In Figure 3b the AHFO wind speed measurements are compared to the velocity measured with the sonic anemometer. The comparison for all angles can be found in Figures B1 and B2. The wind speeds measured with AHFO are calculated using 10 temperature differences (duplex setup with $2 \times 5$ heated and reference measurements), i.e., for the 90° setup this is equivalent to a height of $\sim 0.675$ m in the wind tunnel.

Figure B1 shows the sample rate DTS data against the 1-s average sonic anemometer data, for the four different angles of attack. For all four angles the results are satisfying. The 90°, 45° and 30° angles slightly underestimate the wind speed. The 15°

angle is overestimated, especially at higher wind speeds. Figure B2 shows the same data set, but then combined for all angles, for a 1-s and temporally averaged 30-s resolution. A clear improvement of the precision is visible when temporal averaging is performed. Even though the directional sensitivity equation (Eq. 14) is not yet fully calibrated, the bias is negligible, with a coefficients of determinations ranging from 0.92-0.96, with a slope ranging from 0.91 to 1.14 and a intercept ranging from -0.70-0.64 $ms^{-1}$ (See Figure B1 for each angle). The wind speed measurement are the least accurate for the 15° angle of attack.

To get more insight in the quality of the results, a dimensionless analysis is performed. In Figure 4, the non-dimensional wind speed accuracy for the whole wind tunnel experiment is shown. For all combinations (120 individual cases of varying wind speed ($j$), angle and $\Delta T$, the accuracy is calculated according to Eq. 12. As can be seen in Figure 4, $\sigma_a$ depends on the spatial and temporal averaging of the FO data. The averaging time $n_{time}$ is defined as $n_{time} = t_{avg}/t_{sample}$, where $t_{avg}$ can only be an integer which is a multiple of $t_{sample}$. Spatial averaging is defined as $n_{space} = x_{avg}/x_{sample}$, where $x_{avg}$ can only be an integer which is a multiple of $x_{sample}$. In Figure 4 the accuracy is averaged over all wind speeds for each $\Delta T$ and angle combination, with $n_{space} = 10$ and $n_{time}$ varying from 1 to 30, resulting in 12 values for each time resolution.

For the data set ($n = 5 - 300$), the maximum $\sigma_a$ is $\pm 0.03$, which is promising for future applications. The $\Delta T = 6K$ should be the best performing heating setting, however this is not always the case and there are fluctuations between the heating settings, which could be due to neglecting small energy losses, like free convection due to heating of air close to the heated cable (Sayde et al. (2015)), which is temperature dependent. With such an energy loss included, the bias of each angle might change. Nevertheless, the bias is fairly constant after $n = 50$ with increasing averaging time, which means further analysis can probably increase the accuracy. The change in bias from $n = 5$ to $n = 50$ is due to the precision of our AHFO measurements, which increases with averaging over longer time (n increases) and is higher for a greater $\Delta T$. This difference is bigger for the 90° cases, as $n_{space} = 5$ instead of $n_{space} = 10$ for the other angles, indicating that spatial averaging also has an effect on the bias.

While the accuracy (bias) remains fairly constant over the averaging period, the relative precision, $\sigma_p$ improves significantly (Fig. 5). The precision is calculated for all 120 $\Delta T$, angle and wind speed combinations (where $j = 1, 3, 5, 7, 9, 11, 13, 15, 16, 17$ $ms^{-1}$), using Eq. 13.

For the calculation of the precision $u_{DTS}$, we considered the variability of the wind speed, even though small in the wind tunnel. We assumed that this variability is measured by the sonic anemometer measurements and we assume that this per definition is smaller than the variability of the DTS machine $u_{DTS}$ estimates. After applying Eq. 13 the variability of the DTS machine $u_{DTS}$ are obtained. For each of the 120 combinations, $\bar{u}_{sonic}(j)$ and $\bar{u}_{DTS}(j)$ are the average wind speeds for a $j$. $u_{sonic}(i,j)$ and $u_{DTS}(i,j)$ are single measurements for a $j$.

The precision was averaged over all wind speeds for each $\Delta T$ and angle combinations in Figure 5, which is justified because $\sigma_p$ is normalized by the mean wind speed, hence any linear dependency should be removed.

The precision improves to a $\sigma_p$ less than 0.05 by averaging over time, hence an increasing $n$. Improvement by averaging is expected due to the reduction of noise (van de Giesen et al. (2012)). As mentioned, the main source of noise in DTS data is

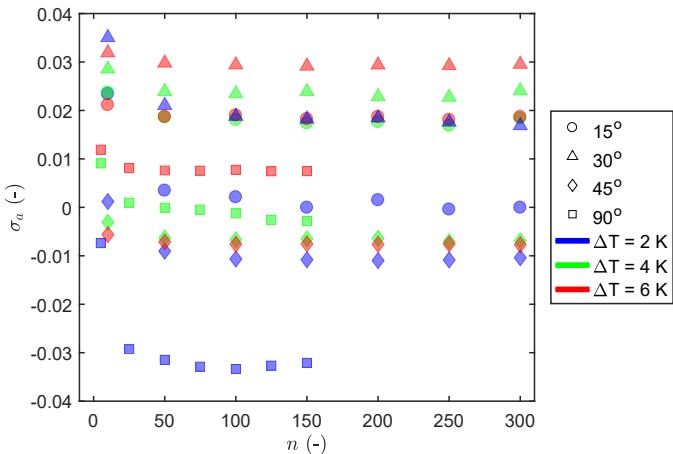

**Figure 4.** Bias in AHFO wind speed as a function of averaging period for different angles of attack, and different fiber heating. With $n$ varying from 5-300.

white noise, this explains the visible improvement of the precision by $\frac{1}{\sqrt{n}}$, as signal averaging is applied, where $n$ is the amount of measurements (Selker et al. (2006b); Kaiser and Knight (1979)).

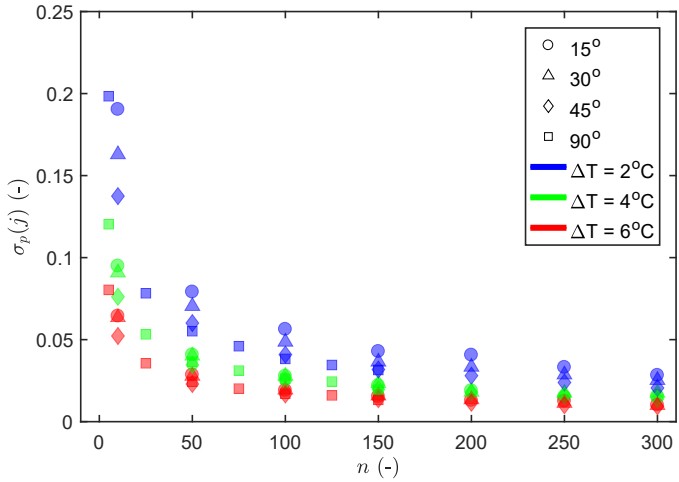

**Figure 5.** Precision of the AHFO wind speed measurements as a function of averaging period. With $n$ varying from 5-300.

### 3.3 Normalized precision independent of sampling settings

In order to remove the influence of different settings (such as the choice of $\Delta T$) and determine a general prediction of precision
5 in future experiments, we normalize the precision. First, the precision is normalized to $\Delta T$ (Figure 6a), by multiplying Eq. 13

by $\frac{\Delta T}{T_{error}}$, which can be written as Eq. 15.

$$\sigma_p(j,\Delta T) = \sigma_p(j)\cdot\frac{\Delta T}{T_{error}} \tag{15}$$

As a results, $\frac{1}{\sqrt{n}}$ dependence becomes even more clear, as shown by the black solid line showing $\frac{\bar{\sigma}_p}{\sqrt{n_{time}}}\times\frac{\Delta T}{T_{error}}$, where $\bar{\sigma}_p$ is the average of Eq.13, with $n_{space}=10$ (and the $n_{space}=5$ of 90° calculated as $n_{space}=10$ using the $\sqrt{n}$ rule) and $n_{time}=1$.

5 Second, the precision is also normalized to the $\frac{1}{\sqrt{n}}$ behavior, by multiplying Eq. 15 by $\sqrt{\frac{t_{avg}}{t_{sample}}}$, resulting in Eq. 16.

$$\sigma_p(j,\Delta T,n_{time}) = \sigma_p(j)\cdot\frac{\Delta T}{T_{error}}\sqrt{\frac{t_{avg}}{t_{sample}}} \tag{16}$$

$T_{error}$ and $t_{sample}$ are known and depend on the performance and setup of the DTS, in this case we use $T_{error}=0.32$ K and $t_{sample}=1$-s, calculated as described earlier. It appears that the precision by taking the average can be condensed in one number, 1.13, which we denote by the symbol $C_{int}$ (Figure 6b). Intermediate constant $C_{int}$ can be defined as, Eq. 17, with

10 $n_{space}=10$.

$$C_{int} = \overline{\sigma_p(j)\cdot\frac{\Delta T}{T_{error}}\sqrt{n_{time}}} = 1.13\pm0.13 \tag{17}$$

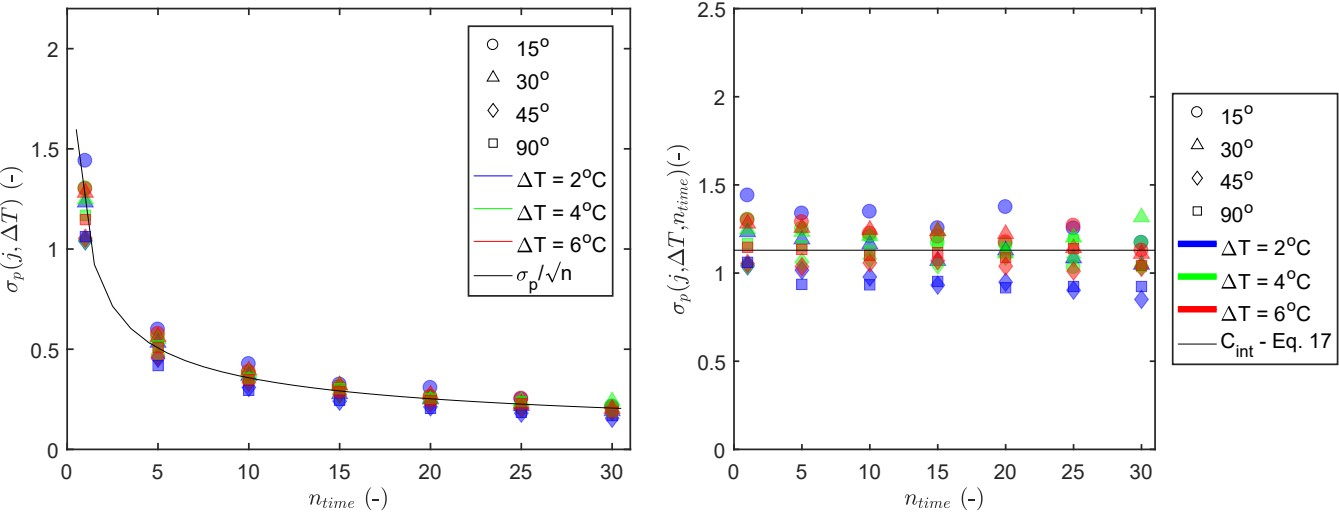

**Figure 6.** a) Precision of the AHFO wind speed measurements as a function of averaging period, independent of $\Delta T$; and b) Precision of the AHFO wind speed measurements as a function of averaging period. Independent of $\Delta T$ and averaging period. With $n_{space}=10$ and the $n_{space}=5$ of 90° calculated as $n_{space}=10$ using the $\sqrt{n}$ rule.

Finally, a final constant for a 1-s and 0.125-m resolution is desired, so it can be used for different kinds of DTS machines, also when a DTS machine has different sampling resolutions. By using the shown $\frac{1}{\sqrt{n}}$ dependency, we can convert $C_{int}$ into

$C_{DTS}$, by multiplying $C_{int}$ by $\sqrt{\frac{10}{1}}$, as $n_{space}$ is 10. This results in Eq. 18 with $n_{space}$=1 and $n_{time}$=1. $C_{DTS}$ is in our paper on purpose not calculated at once, but derived using $C_{int}$. As the wind speed in the middle of the wind tunnel can be assumed constant, we expect $C_{DTS}$ to be better by using 5 measurements in the middle of the wind tunnel instead of picking one of these 5.

$$ C_{DTS} = \overline{\sigma_p(j) \cdot \frac{\Delta T}{T_{error}} \sqrt{n_{time}}} \cdot \sqrt{n_{space}} = C_{int}\sqrt{10} = 3.57 \pm 0.41 \tag{18} $$

### 3.4 Precision prediction

At the start of a new AHFO experiment it is unknown how to make sure the signal-to-noise ratio is sufficient, such that $\sigma_p$ is small. However, given the result that the increase in precision behaves similar for each $\Delta T$ and the averaging time, it is possible to make a prediction for the precision of future work.

In outdoor experiments, the only setting which can be changed is the heating rate, $P_s$, which is assumed to be fixed at a single value. The idea behind the precision prediction is to guide the choice of a heating rate, such that a preferred precision is achieved for a known dominant wind speed range. As the wind speed outside will vary naturally, $\Delta T$ will change accordingly. Therefore, to obtain an expression where $P_s$ is the only unknown, $\Delta T$ first needs to be expressed as a function of the wind speed $u_N$ and the heating rate ($P_s$). This can be done by using Eq. 10. To obtain a first estimate, some assumptions can be

made. The numerator of Eq. 10 consists of three terms, of which the first one with heating rate ($P_s$) is dominant compared to the other ones, namely 10-100 times bigger. When these minor terms are neglected Eq. 10 can be simplified to:

$$ u_N = \left( \frac{0.5 P_s \pi^{-1} r^{-1}}{C d^{m-1} \mathrm{Pr}^n K_a v_a^{-m}(T_s - T_f)} \right)^{1/m} = \left( \frac{AP_s}{B \Delta T} \right)^{1/m} \tag{19} $$

With $A = 0.5\pi^{-1}r^{-1}$, $B = C(d)^{m-1}\mathrm{Pr}^n K_a v_a^{-m}$ and $\Delta T = T_s - T_f$, resulting in an expression for $\Delta T$ as a function of wind speed:

$$ \Delta T = \frac{AP_s}{Bu_N^m} \tag{20} $$

Knowing this expression of $\Delta T$, Eq. 18 can again be rewritten into Eq. 21 (assuming the difference between $u_{sonic}(i,j) - \bar{u}_{sonic}(j)$ and $u_{DTS}(i,j) - \bar{u}_{DTS}(j)$ is negligible), which expresses the precision estimate, with $P_s$ as only parameter which can be changed during an experiment.

$$ \sigma_p(j, n_{space}, n_{time}, P_s) = C_{DTS} \frac{BT_{error}u_N^m}{AP_s} \sqrt{\frac{1}{n_{space} \cdot n_{time}}} \tag{21} $$

Where $n_{space} \times n_{time}$ is the number of measurements over which the observed wind speed is averaged, in either space or time domain. By assuming that all constants are known from literature and the set-up, a first estimate of the error can be made

for every velocity or heating rate given. If a dominant wind speed range for a new project is known, an associated heating rate can be found, such that the error is sufficiently small.

As an example, Figure 7 shows the estimated precision for our experiment at 1-s ($n_{time} = 1$) and $\sim 0.675$-m ($n_{space} = 10$) resolution over a range of wind speeds and heating rates. If the diameter of the fiber is different, this is taken into account via term $A$ from Eq. 21, which includes the radius ($d = 2r$). Also, when a DTS machine with a different performance and setup is used, this can be implemented by calculating an appropriate $T_{error}$ accordingly. Of course different applications will demand different space-time averaging windows, depending on the scientific research question to be answered with AHFO, which option is included by $\sqrt{\frac{1}{n_{space} \cdot n_{time}}}$.

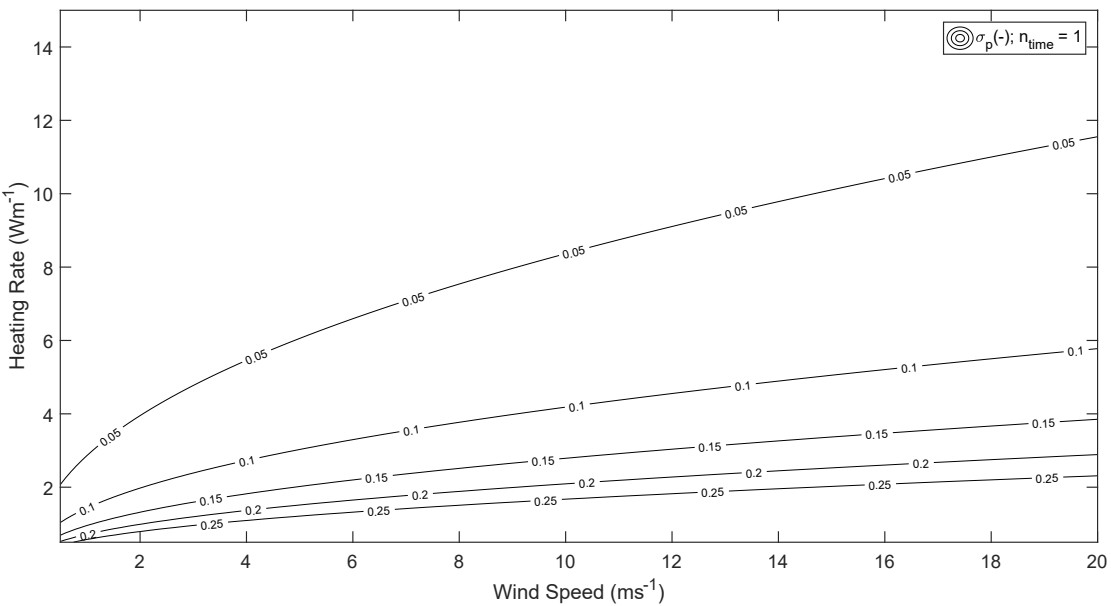

**Figure 7.** Expected precision (contour lines) for a given heating rate and wind speed as calculated from Eq. 21, with $n_{space} = 10$ and $n_{time} = 1$ and the angle of attack is 90°.

In outdoor experiments, the influence of the short and long wave radiation will be present. However, as long as the radiation is the same for the heated and non-heated segment, this does not influence the error estimation, as for the signal-to-noise ratio, $\Delta T$ is the most important factor. When the heated and reference fiber are close to each other, which is also needed for properly estimating the wind speed, both fibers will experience a similar contribution of external radiation, such that the overall $\Delta T$ will be relatively unaffected by this factor.

**Verification of the precision prediction**

For verification purposes the calculated precision (Eq. 13) is combined with the predicted precision (Eq. 21) in Figure 8. As can be seen in Figure 8, the precision of the AHFO system is estimated well and the one time standard deviation covers all calculated precisions. When using Eq. 21 one should consider $u_N$ is derived for a 90° angle of attack. If wind speeds with other angles of attack are expected, one should use Eq. 14 for prediction of the precision. $u_N$ is the measured wind speed normal to the FO cable and the measured wind speed is lower in case of an angle < 90°. In this case one should use $u_N = u_{DTS} \cdot \cos(\varphi - 90°)^{m_1}$. Concluding, with our prediction equation we can predict all our settings within a one standard deviation interval, showing general applicability.

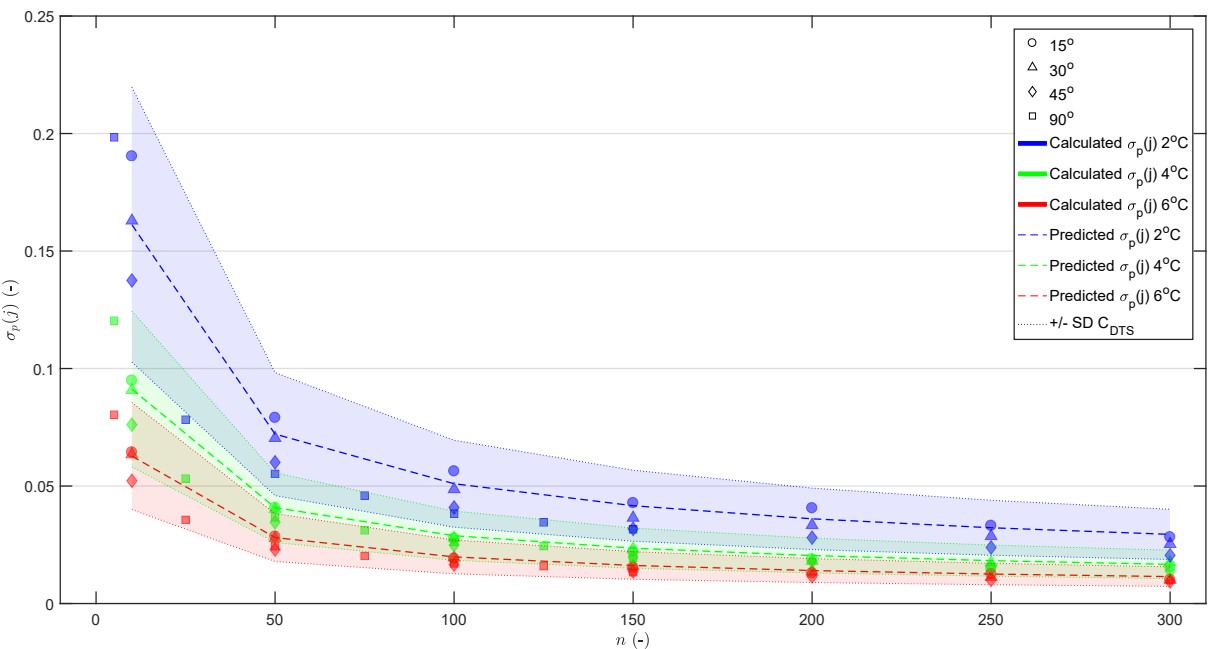

**Figure 8.** Verification of the precision function (Eq. 21). The predicted precision (dashed lines) is compared with the calculated precision from our experiment (Eq. 13). The dotted lines show the prediction with a ± standard deviation of $C_{DTS}$.

## 3.5 Considerations using AHFO outdoors

The experiments described here were performed in a controlled wind tunnel environment. When performing outdoor AHFO experiments, several factors need to be considered. First of all, during field experiments the relative humidity and temperature might have such a big range that assuming certain parameters (e.g., $K_a$ and $v_a$) as constant is not applicable anymore (Tsilingiris (2008)). Furthermore, for small wind speeds (e.g., $< 1$ ms$^{-1}$), the neglection of energy losses like free convection seems not entirely applicable, as this term becomes more dominant in comparison to forced convection. This is confirmed in our study, where it was visible that the response is different between a well ventilated and non-ventilated cable, hence the

accuracy is dependent on the wind speed. Although not shown in this paper, it seemed there was no time response difference between a vertical or horizontal mounted heated cable, however by mounting the cable in a horizontal or vertical direction, free convection might influence the temperature measurements as the heated air is moving upward.

Also, the flow in the wind tunnel is laminar and has less turbulence than in outdoor conditions (Appendix C). This is a good setting for calibration of the AHFO method, however in outdoor conditions (small scale) turbulence around the cable is something to take into account. Especially with smaller wind speeds the cooling by turbulence around the cable can be an additional heat loss component, which is not included in the energy balance and therefore could lead to overestimation of the wind speed. Furthermore, one should take into account that wet fibers, due to rain or dew fall, might have an altered heat loss.

It is shown that AHFO can give reliable wind speed measurements, however the precision and accuracy is not as good as with a sonic anemometer. The major addition of AHFO is the possibility to sample the wind speed with a high spatial distribution. It should be taken into account that the time resolution is lower than that of a sonic anemometer and therefore AHFO is less suitable for small scale turbulence, but larger scale turbulence (>1-s; >0.3-m) can potentially be fully captured with a 2D/3D setup with distributed measurements. Despite the high potential resolutions (1-s and 0.3-m) the user should consider to average in either the space or time domain to enhance the precision of the obtained data. The choice for averaging over space or time should be made based on the researched topic.

Finally, when measuring in the field, the use of high quality reference point measurements (e.g., sonic anemometer) is recommended, for example to be able to compensate for possible biases. Using a vertical set-up of the fibers would reduce the need for compensating for the angle of attack, as the mean wind speed is mostly parallel to the surface. However, in complex terrains as for example inside canopies, one ancillary device could be not enough due to the high variability of the wind field. In such a case, a more complex 3D set-up of DTS/AHFO (Zeeman et al. (2015)) could be an indication of the angle of attack. Also recently, a new method is under development which tries to measure the angle of attack with a single cable, using micro structures attached to the fiber (Lapo et al. (2020)).

## 4  Conclusions

Through a series of controlled wind tunnel experiments, new insights into the accuracy and precision of the newly introduced AHFO wind speed measuring technique were obtained. With high spatial (0.3-m) and temporal (1-s) resolution, the AHFO wind speed measurements agreed very well with the sonic anemometer measurements, with coefficients of determination of 0.92-0.96. It is also shown that the AHFO technique has the possibility to measure with a precision and accuracy of 95%. Some additional work is needed, as there still is a small overestimation, which may be caused by neglecting some energy fluxes, such as free convection due to heating of the air close the heated cable. Furthermore, it is possible to optimize the directional sensitivity compensation by extended calibration. Compensating for the directional sensitivity requires ancillary measurement devices in order to measure the angle of attack, however in complex terrains as for example inside canopies, one ancillary device could be not enough due to the high variability of the wind field.

The error prediction equation (Eq. 21) is an important result of this work that will aid in the design of future experiments. This design tool helps with choosing a heating rate for the actively heated fiber in order to be able to create a sufficiently high precision. Based on the prevalent wind speeds of a potential field experiment site, a first estimate of an associated sufficient heating rate can be calculated. Due to the way this design tool is constructed, it can be a good first estimate for all kinds of fibers, DTS precisions, and user preferred spatial and temporal resolutions.

The AHFO technique can reliably measure wind speeds under a range of conditions. The combination of high spatial and temporal resolution with high precision of the technique opens possibilities for outdoor application, as the key feature of the AHFO is the ability to measure spatial structures in the flow, over scales ranging from one meter to several kilometers. In the future, the technique could be useful for micrometeorological and hydrological applications, allowing for characterization of spatial varying fields of mean wind speed, such as in canopy flows or in sloping terrain.

*Author contributions.* Justus van Ramshorst prepared and performed the experiments, worked on analyzing the data and writing the manuscript. John Selker and Chad Higgins assisted with the experiments and analyzing the data and contributed to the manuscript. Miriam Coenders-Gerrits, Bart Schilperoort, Bas van de Wiel and Jonathan Izett helped with analyzing the data and contributed to the manuscript. Huub Savenije and Nick van de Giesen contributed to the manuscript.

*Competing interests.* The authors declare that they have no conflict of interest.

*Acknowledgements.* Many thanks for the practical assistance of Cara Walter and Jim Wagner with the AHFO/DTS setup and appreciation for the people of the OPEnS LAB for assisting with the assembling of parts. This project was partly funded by NWO Earth and Life Sciences, Veni project 863.15.022, The Netherlands. We are also grateful for the funding by Holland Scholarship and CTEMPs.

## Appendix A: FO cable schematization

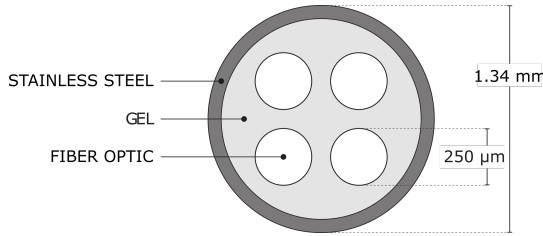

**Figure A1.** Cross-section of the FO cable

## Appendix B:  Comparison of AHFO and sonic anemometer wind speed

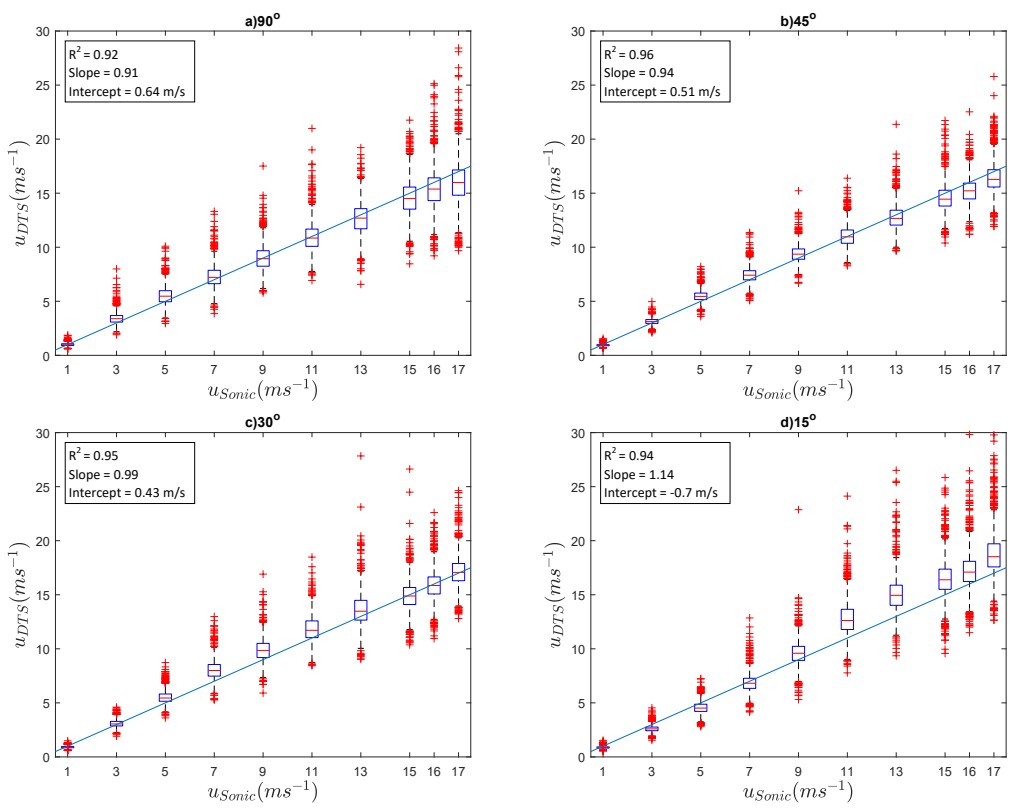

**Figure B1.** Comparison of AHFO and sonic anemometer wind speed at a 1-s temporal resolution, for the four different angles of attack. a) 90°, b) 45°, c) 30°, and d) 15°. $n_{space} = 10, n_{time} = 1$. The line represents the 1:1-line.

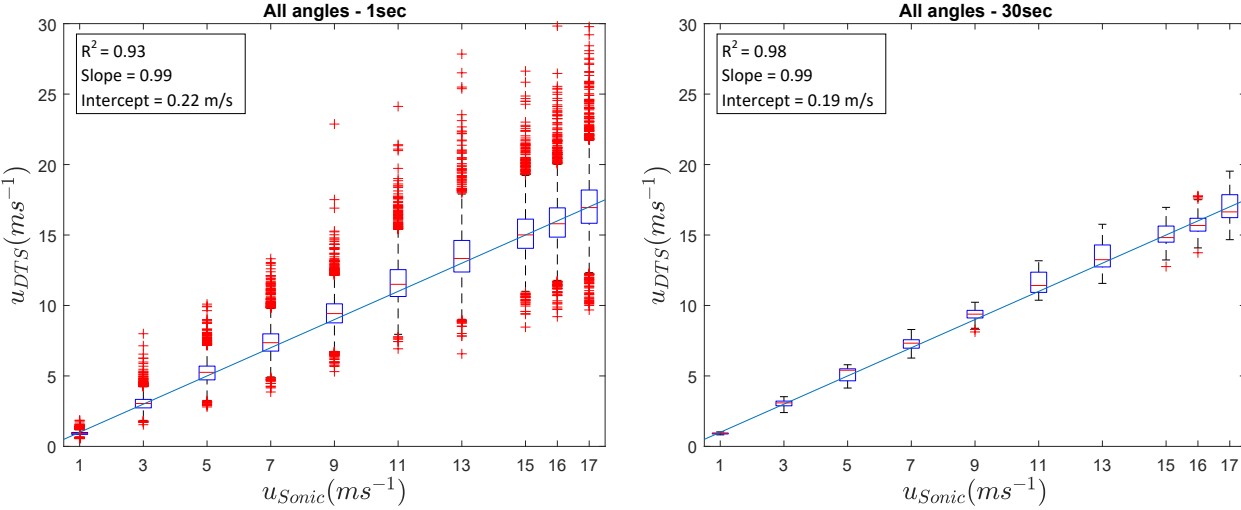

**Figure B2.** Comparison of AHFO and sonic anemometer wind speed, combining all angles of attack at a 1-s(a) and 30-s(b) resolution. $n_{space} = 10, n_{time} = 1$ and 30. The line represents the 1:1-line.

## Appendix C:  Wind tunnel flow characteristics

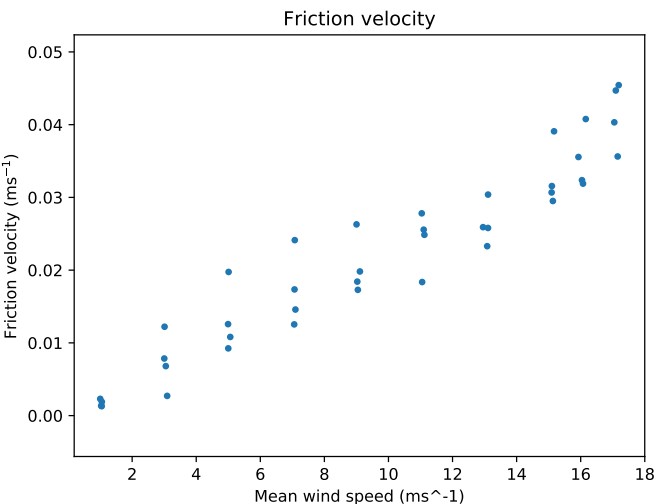

**Figure C1.** Friction velocity (ms$^{-1}$) in the wind tunnel during AHFO experiment.

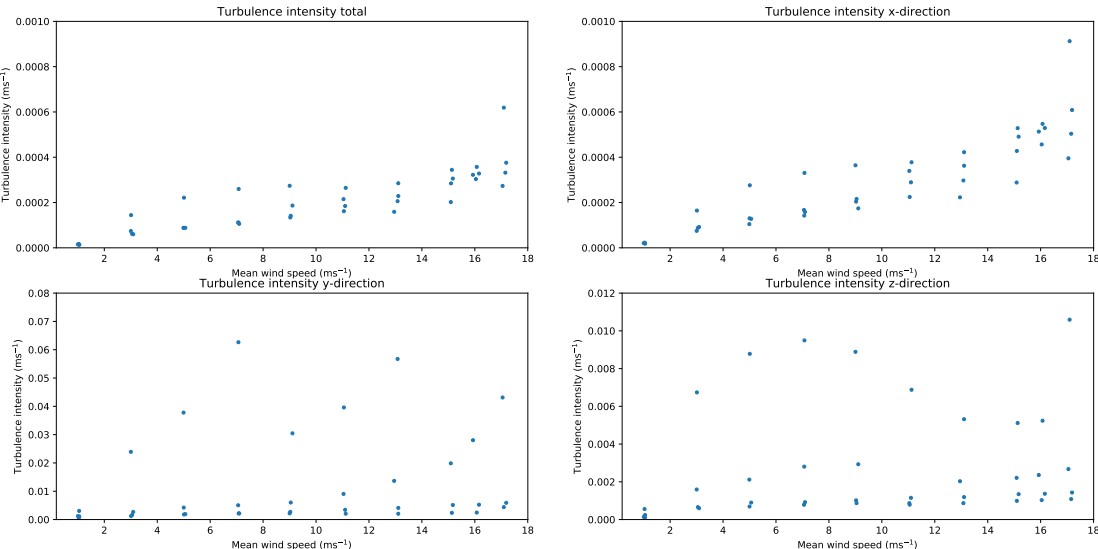

**Figure C2.** Turbulence intensity (variance devided by mean wind speed) $(\text{ms}^{-1})$ in the wind tunnel during AHFO experiment. The x-direction is in the flow direction. The y-direction is the width direction. The z-direction is the height direction.

**Table D1.** Standard deviation $\sigma_{space}$ of 5 pairs of AHFO measurements (duplex configuration) per wind speed, and its normalized standard deviation. It shows that the normalized standard deviation is $\approx 3\%$ no matter if one takes the top, mid-top, center, mid-bottom, or bottom pair.

| u (ms $^{-1}$) | 1 | 3 | 5 | 7 | 9 | 11 | 13 | 15 | 16 | 17 |
|---|---|---|---|---|---|---|---|---|---|---|
| $\sigma_{space}$ (ms $^{-1}$) | 0.033 | 0.092 | 0.147 | 0.181 | 0.235 | 0.312 | 0.323 | 0.445 | 0.526 | 0.544 |
| Normalized $\sigma_{space}$ (%) | 0.033 | 0.031 | 0.029 | 0.026 | 0.026 | 0.028 | 0.025 | 0.030 | 0.033 | 0.032 |

For each angle and power rate, the $u_{DTS}$ was calculated with only the two temperature differences (duplex configuration) of the top of wind tunnel, or the mid-top, center, mid-bottom, or bottom of the wind tunnel (thus $n_{space} = 2$). From these 5 pairs we calculated the standard deviation $\sigma_{space}$ per wind speed.

## Appendix D: Amount of measurements

**Table D2.** Amount of temperature differences for each setup ($n_{space}$)

| Angle (in °) | # of $\Delta T$ measurements ($n_{space}$) |
|---|---|
| 15 | 10 |
| 30 | 10 |
| 45 | 10 |
| 90 | 5 |

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
