# Peer review of "Revisiting wind speed measurements using actively heated fiber optics: a wind tunnel study"

_Atmospheric Measurement Techniques, 2019_

## Referee Comment (RC1) · Anonymous Referee #1 · 20 Apr 2019

The study of van Ramshorst et al. investigated the actively heated fiber-optic (AHFO) technique and estimated its accuracy and precision under controlled airflow conditions by comparing to a three-dimensional ultrasonic anemometer. A very valuable error prediction equation for the wind speed measurements at different heating rates were developed, as the heating rate can be a limiting factor for long cables. This equation is also accounting for averaging over space or time which further increases precision. They conclude that AHFO measurements are reliable in outdoor deployments when correcting the measurements for directional sensitivity with a ultrasonic anemometer, choosing the right heating rate and spatial or temporal averaging. Distributed temperature sensing (DTS) measures temperatures along a fiber-optic cable spatially

continuously and can be used in various fields. Especially for atmospheric research this technique offers new insight into the temperature field and thus was implemented in many studies. By using the AHFO technique, wind speed measurements can be added to the system. As the community using the DTS and AHFO technique is growing, the study of van Ramshorst et al. is important for users to be aware of the accuracy, precision and limitation of this technique. Hence, the paper is valuable for our community. The introduction to the determination of wind speed is nicely done, however, I think it can be organized more reader friendly. The overall structure of the paper is logic, but could be reorganized and shortened. In my opinion some figures are redundant. The development of the error prediction equation needs clarification. I could not differentiate results from discussion. Further, I am missing some turbulence statistics of the wind tunnel (friction velocity, velocity aspect ratio, turbulence intensity in different directions,...) to give an estimate how representative the turbulence within the wind tunnel is to outdoor turbulence. I recommend to accept the submitted manuscript after major revisions. I attached a supplement with further details.

Please also note the supplement to this comment:
https://www.atmos-meas-tech-discuss.net/amt-2019-63/amt-2019-63-RC1-supplement.pdf

---

## Referee Comment (RC2) · Anonymous Referee #2 · 25 Jun 2019

**General comments**

The manuscript describes a controlled laboratory evaluation of a recently introduced technique for wind speed measurements using actively heated fiber-optic cables combined with fiber-optic temperature sensing (AHFO-DTS), similar to hot-wire anemometry. The evaluation considers the wind speed, the angle between the mean flow and the sensing cable and the temperature offset between heated and unheated sections of the sensing cable in the experimental design. The results include a simplified model to help plan the heating requirements for experiments in similar conditions.

The study highlights aspects to consider for planning real-world applications of AHFO-DTS and as a major outcome shows that the potential bias due to sensing cable pitch angle may be low and previously used constants should be reconsidered. However, not all laboratory outcomes can be immediately translated to a real-world field application. The authors added a section to discuss this, but should elaborate on the specific conditions in the wind tunnel (compared to real-world) in more detail, to help the reader. The need to include certain outcomes, particularly time averages of wind speed estimates, is unclear to me. Removing those would improve the focus of the text/figures and reduce redundancy. White noise can be mitigated by spatial/temporal averaging, but the introduction does not clearly mention the relevance for this study; I think that the outcomes for high-end resolutions of AHFO-DTS (currently 1Hz, 0.3m) are conclusive enough without it. After major revisions the manuscript should be considered for publication, and I expect it to be a very helpful contribution for those interested in spatial wind speed measurements using this novel technique.

**Specific comments**

In their current form, Fig 4, 5 and 6 do not adequately highlight the differences between the applied corrections or experimental settings. Actually, the uncorrected regression (Fig. 4a) fits the range of observations in the center, making it look like a better fit. Please improve the figures.

The reason to present results for different averaging periods was not clearly introduced. I suspect those averaging details can be left out and this would help focus the result/discussion section (remove Fig 6 through 9). The precision and accuracy are most meaningful at the highest resolution, for combinations of different angle, wind speed and temperature offset (and heating rate) settings, and could perhaps be summarized in a single figure or table.

A fixed temperature difference between the reference and heated probe would require

a feedback system that adjusts the heating rate according to previous or expected wind conditions.

1. In the laboratory setup, was this adjustment in heating rate made manually or automatically? How accurate could this be set and were heating rate readjustments made during the 10-min steady periods? The text suggests that under low angles of attack, the heat exchange is less efficient and, consequently, a lower heating rate can be applied to achieve a 2/4/6 K differences compared to the reference. Please discuss the impact of such variable heating rates on the results.

2. In real-world applications, with more variable wind and radiation conditions, such a feedback system may become challenging. A constant heating rate (variable temperature difference) is perhaps more practical. Therefore, could you also present your results expressed as a function of heating rate, instead of fixed temperature offset?

Section 2.2.2 includes both a modification - a different set of constants - and a simplification of the set of equations from Sayde et al (2015) for applications in a wind tunnel. This is not reflected by the section title. Perhaps move the proposed modification (with figure) to the results section or rephrase the section title.

Why were there no (additional) reference measurement made downwind of the heated cable? In a controlled environment, this could help identify feedback between ref and heated cables in relation to separation distance. In a real-world application, would this make a setup less sensitive to wind direction shifts, or is the relative position of both sections irrelevant? Given the long-standing expertise among the authors, could further recommendations be made for follow-up evaluations of AHFO-DTS inside or outside the wind tunnel?

**Technical corrections**

- p1l1/p1l15: Either Active or Actively, choose one for the AHFO definition.
- p2l21: remove the comma after '(2015)'

- p3l16: 'cable (which encloses the FOs)': suggestion 'FO cable'

- p3l27: 'angles of attach': suggestion 'angles of attack with the mean flow'

- p3l27: Were the ref and heated cable also 8 cm apart in mean wind direction under these slanted angles? From the following sentences this is not immediately clear. If not, would it matter if they would come closer together? Please explain.

- p3l33-p4l1: 'The wind speed in the wind tunnel was fixed at a constant value to create a steady state flow.': perhaps remove, or rephase to be more specific. Do you mean: The engines in the wind tunnel were set to a constant rate, generating a steady (often laminar) flow after some time and, as a result, a dynamic steady state heat exchange of the FO cables with the moving air.

- p4l2: This paragraph contains a listing of three different experimental settings. Please help the reader by presenting them as such. 'First,... Second,. . .' or other rephrasing.

- p4l2: 'Furthermore, for all wind speeds and angles the temperature difference between the reference and heated section, delta T, wat set at 2, 4 and 6 K in order to evaluate the importance of hotwire signal magnitude.'

- p4l8: 'machine': maybe 'instrument'?

- p4l9: 'One cable segment was heated.', suggestion: remove this first sentence and rephrase the second sentence 'The stainless-steel casing of the heated section was ...'

- p4l10: Steel or other wire material?

- p4l11: a temperature difference should be reported in K (Kelvin). Also, those values were already mentioned. suggestion: cahnge ' to a fixed level, either 2, 4 and 6 °C, depending on the setup' to 'at fixed levels'.

- p5l4: what is an 'ambient bath'?

- p4l7/p5l8: double ended DTS measurement was applied (p4) but not used (p5). Please put these details together in a single paragraph. Are there quantitative criteria to reject the double ended method using the arguments here, perhaps described in literature (citation)?

- p5l21-p5l22: Somewhat vague. 'An energy balance ... advective energy transport from the heated cable, ...'. suggestion: 'An energy balance method ... heat dissipation from the heated section, ...'. The heat dissipation may not be a fully advective process. p6l14: please rephrase, introduce the abbreviation last, suggestion: 'The Nusselt number, Nu, is the ...'

- Match the citation style according to journal guidelines, throughout the text. Particularly the year notation with nested parenthesis seems odd, p2l3 should be "... (e.g., Goodberlet et al., 1989)".

- Fig. 4, Fig. 5, Fig. 6: Add a meaningful indication of the statistics of these point clouds - add means, or convert the presentation to boxplots - or remove the figures.

- p10l1-p10l21: contains details already mentioned before and details that should better be placed in an introduction.

- Eq 12 and 13: where is $u_N$ defined?

- p10l26: Is the center of the wind tunnel cross section also the center of the ref/heated FO cable sections? Why use a section of 0.9 m and not 0.3 m, the advertised resolution of the system, nearest to the location of the sonic anemometer sensor path? Please show that the extended spatial range did not have an impact on the outcomes. Also, by taking a fixed length of FO, the observed positions in the cross section of the tunnel changed with angle; the 15 degrees angle of attack setting would only integrate approximately 0.2 m vertically. Since the bottom of the fiber was attached to the tunnel (p3l26-30), were the positions of the center of the ref/heated sections determined at different positions along the optical-fiber length for each angle?

- p10l26: 'Only the temperatures from the middle of the wind tunnel are used, to prevent using data with side/boundary effects.' If the center of the observed sections, independent of the angle of the cable, were centered at the same position in a cross-section of the tunnel as the sonic anemometer, please state this explicitly in the method section.

- p10l27: Referring to 'data' here is not very specific, please rephrase. suggestion: 'AHFO-DTS derived wind speed estimates'.

- p10l28: Averaging to 30 sec, including Fig. 6+7, shows no new information: better to leave it out?

- Fig. 5c: In the review copy, it seems like the 16 m/s data included values that had not reached steady state yet (variability in $U_{sonic}$, < 16 m/s). Could you please verify?

- p12l11: 'The precision is calculated for all 120 àT , angle and wind speed combination, using Eq. 15.' Please refer to the equation after it has been defined.

- p13;14: Are these numbers for $T_{error}$ computed, based on the calibration bath sections? Or a specification of the instrument?

---

## Author Comment (AC1)

*23 July 2019*

This document contains a point by point reply to review 1 and 2 of the paper by Justus van Ramshorst et al. (DOI: 10.5194/amt-2019-63). The manuscript is already partly revised and restructured for clarification of some misunderstandings and is added after the replies to both reviews. Most figures still need to be changed and the wind tunnel statistics will be added later.

In this document you can find:

**REVIEW 1**

Review on Manuscript of van Ramshorst et al. 2019 April 20, 2019

**Review report: Wind speed measurements using distributed fiber optics: a wind tunnel study**

Author of the paper: van Ramshorst et al.
Journal: Atmospheric Measurement Techniques
Manuscript DOI: 10.5194/amt-2019-63

**General Comments**

The study of van Ramshorst et al. investigated the actively heated fiber-optic (AHFO) technique and estimated its accuracy and precision under controlled airflow conditions by comparing to a three-dimensional ultrasonic anemometer. A very valuable error prediction equation for the wind speed measurements at different heating rates were developed, as the heating rate can be a limiting factor for long cables. This equation is also accounting for averaging over space or time which further increases precision. They conclude that AHFO measurements are reliable in outdoor deployments when correcting the measurements for directional sensitivity with a ultrasonic anemometer, choosing the right heating rate and spatial or temporal averaging. Distributed temperature sensing (DTS) measures temperatures along a fiber-optic cable spatially continuously and can be used in various fields. Especially for atmospheric research this technique offers new insight into the temperature field and thus was implemented in many studies. By using the AHFO technique, wind speed measurements can be added to the system. As the community using the DTS and AHFO technique is growing, the study of van Ramshorst et al. is important for users to be aware of the accuracy, precision and limitation of this technique. Hence, the paper is valuable for our community.

*We appreciate your acknowledgment for the importance of our study with AHFO.*

The introduction to the determination of wind speed is nicely done, however, I think it can be organized more reader friendly. The overall structure of the paper is logic, but could be reorganized and shortened. In my opinion some figures are redundant. The development of the error prediction equation needs clarification. I could not differentiate results from discussion. Further, I am missing some turbulence statistics of the wind tunnel (friction velocity, velocity aspect ratio, turbulence intensity in different directions to give an estimate how representative the turbulence within the wind tunnel is to outdoor turbulence. I recommend to accept the submitted manuscript after major revisions.

*Thanks for your feedback, in the comments below we will reply to each comment.*

**Title and structure of the paper**

Throughout the paper the abbreviation AHFO is used, however the title uses "distributed fiber optics". The title may also incorporate that is not the first paper using the AHFO technique. I propose "Revisiting wind speed measurements using actively heated fiber optics: a wind tunnel study" or similar.

*We agree with the suggestion and will revise the title accordingly in the final manuscript*

I would propose another order of the sections. After the introduction, I would start with the introduction to the DTS technique and the signal-to-noise ratio (Section 2.4), then introduce the energy balance of the fiber (Section 2.2), then introduce the experimental setup (Section 2.1), because then the reader already know why two fibers are needed, why spatial averaging is potentially important, etc. However, this order is a minor point and could also be chosen differently.

*Thank you for the valuable comment, we agree that your suggestion is more logical for the reader and we revise the order into: 2.4, 2.2, 2.1. We would like to add 2.3 (line 1-10) as 2.2.3 were the need for an angle of attack correction is explained. 2.3 (line 11-end) will be added to the results as new proposed angle of attack correction.*

Afterwards I would not differentiate between results and discussion. The discussion was insufficient, as I could find no references comparing the work to other studies nor testing or discussing the error prediction equation. I propose to have the following sections instead of Results and Discussion: Directional Sensitivity, Accuracy of AHFO, Precision of AHFO, Error Prediction Equation, Outdoor Deployment of AHFO. Then finally the conclusions.

*We agree with the suggestion of reordering the sections and we revise the order into on chapter 3 with the following subsections: 3.1 (2$^{nd}$ part 2.3), 3.2 (old chapter 3), 3.3 (4.1), 3.4 (4.2) and 3.5 (4.3).*

I also have specific comments on the following sections:
- Determination of Wind speed (complete Section 2.2):
The reader is barely able to follow. The equations are introduced in Section 2.2.1, but in the following section the introduced equations are altered or simplified. I suggest to define subsections each concerning one part of the energy balance of the fiber-optic cable, similar to the study of Sayde et al. (2015). Within each subsection all assumptions and simplifications should be noted. This should also shorten Section 2.2.

*Considering your comments, we agree that this is a complex section. We did not want to repeat the entire paper of Sayde et al. (2015), however we only summarize its main findings. Nonetheless to guide the reader we propose the combining of 2.2.1. and 2.2.2 into one single section, with the following order and special subsections as follows:*

- *Original energy balance by Sayde et al.*
  - *Simplification of the energy balance*
  - *Advective heat transfer coefficient (h) including modifications and assumptions.*
- *Revised simplified determination of wind speed*

*We hope this is structure helps to clarify (see revised manuscript)*

DTS and Signal-to-Noise ratio analysis (Section 2.4):
I don't see the motivation of this sections besides describing the measurement principle of the DTS technique and sources of noise. Also some sentences are not precise and remain unclear (p.10 l.11, l.17-18, l.20-21) or could be removed (p.10 l.18-19).

*We would like to provide the reader some basic information about DTS and more importantly we introduce white noise (temporal and spatial averaging) and the importance of high enough deltaT Furthermore:*
- *p.10 1.11: we decided to remove this sentence because it does not add any used information.*
- *17-18: we meant with this sentence that huge temperature differences could create complications related to energy transfer, for example: buoyancy forces become larger. We clarify this.*
- *20-21: we clarified this.*
- *18-19: we changed the sentence*

Using AHFO outdoors (Section 4.3):
The last paragraph belongs into the introduction
*We agree and will add it in the introduction after line 5 page 3 (new manuscript).*

**Questions on deriving the error prediction equation (Sect.4.1&4.2)**

The main goal and strength of this paper is the development of the error prediction equation depending on the wind speed, the heating rate and accounting for averaging over space and time. However, in my opinion, the development of this equation have to be presented in a more reader friendly way and the assumptions and the validity of this equation has to be reconsidered. I have some questions concerning the intermediate constants and Eq. 20:

*We understand were not clear in our presentation which lead to confusion in this entire section and relates to many of your questions. Therefor we rewrote our results and discussion section (new manuscript). We first explained the revised directional sensitivity equation (3.1). Accuracy and precision are presented in section 3.2. In section 3.3 we normalize the precision results. In 3.4 we present the precision prediction.*

The parameter γ is introduced representing σp at 1 s temporal, and 10 measurement spatial, resolution, hence a specific, empirically derived σp.
-> Does this mean that the authors averaged over 10 $u_{DTS}$ measurements spatially? Or did the authors average the temperature differences over 10 spatial measurements? Or did the authors average the temperature of the unheated and heated cable over 10 spatial measurements and from that computed the temperature difference and thus $u_{DTS}$?

*The latter is the case. We averaged 5 heated and 5 non heated temperature measurements and calculated the temperature difference and successively U_DTS from this for every second. We will clarify this in the text. We hope we clarify this with parameters m and k presented in the new manuscript. $n_{space}$ stands for the amount of spatial measurements, in our case 5. And $n_{time}$ stands for the amount of temporal measurements, in our case 1.*

in Eq. 16 γ stands for a specific, empirically derived σp, however, when included in
Eq. 20 the same γ is representing σp depending on n, Ps and $u_N$, which does not seem
logical to me. How can the authors defend this?

*γ is a function of $n_{space}$, $n_{time}$, u and Ps. We removed the use of gamma and introduced equation 15-17*
*which hopefully clarifies.*

$\sqrt{\dfrac{t_{avg}}{t_{sample}}}$ is $\sqrt{n}$, with n being the number of measurements over which $u_{DTS}$ is averaged over time. This is
not mentioned in the text.

*This is correct, we explained this in the text by introducing parameter $n_{time}$.*

 Following this, Eq. 16 can be rewritten:

$$C_{int} = \gamma \frac{\Delta T}{T_{error}} \sqrt{n} \qquad\qquad\qquad (1)$$

$C_{int}$ is then presented as an intermediate constant, which is also shown in Fig. 9b. How-
ever, is $C_{int}$ the mean over all ΔT and attack angles? Fig. 9b shows a spread of $C_{int}$
from 1.3 up to 2.

*Yes, this is the mean of all ΔT and attack angles, as we want to present a new constant which can be*
*used as first prediction for all measurements.*

Further, did the authors use the constant ΔT or the actually measured
ΔT? Please clarify or use different symbols.

*We used the measured ΔT. We set the power input such that the measurements had roughly 2, 4 and 6*
*°C difference.*

"By using the shown $\sqrt{\dfrac{1}{n}}$ dependency, we can easily convert $C_{int}$ into $C_{DTS}$" (p.15 l.1-2):

- what is $C_{DTS}$ compared to $C_{int}$ or what does it represent?

*$C_{int}$ is now defined by equation 18 in the revised manuscript and only considers the temporal averaging.*
*$C_{DTS}$ also considers the spatial averaging, see equation 19.*

*See section 3.3 in the revised added manuscript*

- is n representing the space or time domain in this context?

*For this question we refer to the revised manuscript were we redefined n as $n_{space}$ * $n_{time}$, which means n*
*is the amount of measurements in time and space.*

$C_{DTS}$ is computed "by multiplying $C_{int}$ by $\sqrt{\frac{10}{1}}$ as n is 10 times less" (p.15 l.2), what

is a confusing statement. In my understanding $\sqrt{\frac{10}{1}}$ is $\sqrt{\frac{x_{avg}}{x_{sample}}}$ with x representing the

space domain. This is $\sqrt{n_{space}}$ with nspace being the number of measurements over which
$u_{DTS}$ is averaged over space. Hence I derive the following equation for $C_{DTS}$:

$$C_{DTS} = C_{int}\sqrt{n_{space}} = \gamma \frac{\Delta T}{T_{error}} \sqrt{n * n_{space}} \qquad (2)$$

*We rewrote the calculation of $C_{DTS}$ and introduced the spatial and temporal $n_{time}$ and $n_{space}$, like you suggested.*

in Section 4.2 the goal was to have an estimation for σp. Therefore, Eq. 16 and Eq. 17
are combined, σp is inserted for γ (which is a point of discussion for me as mentioned
earlier), solved for ΔT, and inserted in Eq. 19 to derive Eq. 20 when solving for σp.

-> when I did the proposed evolution from Eq. 16 to Eq. 20, I got a factor of $\frac{1}{\sqrt{n}}$, not $\sqrt{\frac{1}{n}}$

*Mathematically $\frac{1}{\sqrt{n}}$, and $\sqrt{\frac{1}{n}}$ are the same.*

-> in Eq. 20 n "is the number of measurements over which the observed wind speed is
averaged, in either space or time domain" (p.15 l.22). However, I interpret this n as
n + nspace and not n * nspace (as shown in my Eq. 2). Hence, I think $C_{DTS}$ is computed
incorrectly.

*Please consider the revised manuscript, which should clarify that n indeed is $n_{space} * n_{time}$ as in equation 2 of your review.*

-> are the authors proposing that there is also a $\frac{1}{\sqrt{n\_space}}$ dependency for σp? I am missing a figure
showing this.

*By averaging over space and time the white noise in DTS measurements can be lowered. If the temperature difference and therefor wind speed is (roughly) the same over a spatial distance, averaging over space has the same effect as averaging over time. With averaging over time also uses the assumption that the wind speed stays roughly constant (and in our case the wind speed is constant over time).*

-> this changes Fig. 10 completely

*As mentioned above, due to our unclearness our equations were misinterpreted and figure 10 does not change completely.*

The authors should consider to use different symbols for the time and space domain like
$n_{time}$ and $n_{space}$ or similar

*We agree and introduced $n_{time}$ and $n_{space}$, and n as you suggested (see the revised manuscript)*

Why did the authors choose γ representing the empirical derived σp when averaging over 10 measurements spatially? As the effect of spatial averaging is also under study in this paper, I would derive σp without spatial averaging, so the lowest possible precision, and then investigate the effects of spatial averaging. Also the derived $C_{int}$ and $C_{DTS}$ seem to be biased by this decision.

*We added some explanation to section 2.3 and 3.3 in the revised manuscript where we give an explanation:*

*Section 2.3: Page 9 line 18 until page 10 line 2.*
*Section 3.3: Page 4 line 15 until 22*

In my understanding $C_{DTS}$ is derived empirically when choosing Ps depending on $u_N$ to have a constant ΔT and $C_{DTS}$ is a mean over all experiments.

*This is correct.*

This is not considered in Eq. 20 nor further discussed. Is $C_{DTS}$ representing σp for a constant Ps during different wind speeds, even though the experimental design was different and $C_{DTS}$ is a mean over all experiments? Was there an experiment done using a constant Ps verifying the error prediction function?

*We did not perform such an experiment. The aim of our study is to determine the heating rate required to have a desired σp for roughly the highest expected wind speed for your application outdoors. For the lower wind speeds the σp will be better, as the temperature difference will be higher and therefor σp.*

Do the authors suggest to use a different Ps depending on wind speed? In my opinion it might be not useful in field deployment with quickly varying wind speeds to change the heating rates constantly as the fiber-optic cable needs to reach steady state and there is also a response time between the changed heating rate and DTS measurements. This could potentially lower the precision instead of increasing it.

*As mentioned above, the authors suggest to determine one constant heating rate Ps, which is based on the expected highest wind speed and should also be sufficient for the lower wind speeds.*

I propose to derive the error prediction function in a more clear way. As shown in Fig. 8 for each ΔT the precision σp is following a $\frac{1}{\sqrt{n}}$-line. Hence, we assume the following dependency:

$$\sigma_p(n) = \frac{\alpha}{\sqrt{n}} \tag{3}$$

with α being a constant different for experiment set up. We found that α depends on ΔT and $T_{error}$:

$$\alpha = \left(\frac{\Delta T}{T_{error}}\right)^{-1} \tag{4}$$

with $T_{error}$ = 0.25 K being the performance of the DTS dependent constant and ΔT being

the measured temperature difference between the cables. Hence, α is representing the quality factor for the wind speed measurements. The lines derived from α could also be added in Fig. 8. When simplifying Eq. 18 of the submitted manuscript we can assume that ΔT is mainly depending on the following parameter:

$$\Delta T = \frac{AP_s}{Bu_n^m} \tag{5}$$

Combining my Eq. 4 and 5 and inserting that in Eq. 3, I derive the following error prediction equation:

$$\sigma_p(n, u_n, P_s) = \frac{BT_{error}u_n^m}{AP_s}\frac{1}{\sqrt{n}} \tag{6}$$

If I did not miss a point, no empirically derived intermediate constant has to be used for the error prediction equation.

*We compared your derived equation with our final equation and concluded that there is a difference, because in our equation there is an additional dependency on the wind speed, because σp on itself is also depended on wind speed (see our equation 14), which not included in your equation 3. With our C$_{DTS}$ we averaged for our wind speed range, which leads to a small difference in estimating the final σp and explains the difference between your eq. 6 and our eq. 22 (revised manuscript)*

**Terminology**

- p.2 l.28-29: "advection of cooler ambient air". I think convective heat loss is the correct phrase here. Please make sure the correct terms are used throughout the manuscript.

*We changed this accordingly.*

- p.6 l.14: There is a difference between the turbulent Prandtl number and the Prandtl number representing the ratio of momentum diffusivity (kinematic viscosity) and thermal diffusivity. Please clarify.

*We meant the latter, see line 20, page 7.*

- p.10 l.20:
  "The precision is an indication of the variability of the wind speed (e.g. RMSD),..."
  ! variability of the wind speed can also describe the deviation from the mean wind speed. However, I think in this context the authors refer to the precision of a measurement as- suming a constant wind speed.

*We agree, we indeed look at the precision of the wind speed measurements and not at the natural wind speed variability. We changed the sentence into:*

*"The measurement precision is an indication of the variability of wind speed measurements (e.g., RMSD), as opposed to accuracy which describes a systematic measurement error for which van be compensated (in our case expressed by the bias)."*

- What is RMSD?

*The root mean square deviation, in our case defined in equation 14.*

- "... as opposed to accuracy which describes a systematic error that can be removed through calibration (e.g., a bias)."! Accuracy it the combination of trueness (bias) and precision. The accuracy of a measurement can be low due to a poor trueness (high bias) or due to a poor precision. Please adjust throughout the manuscript.

*Accuracy can be defined in two ways.*

*"**Accuracy** has two definitions:*

1. *More commonly, it is a description of underline{systematic errors}, a measure of underline{statistical bias}; low accuracy causes a difference between a result and a "true" value. underline{ISO} calls this trueness.*
2. *Alternatively, ISO defines accuracy as describing a combination of both types of underline{observational error} above (random and systematic), so high accuracy requires both high precision and high trueness." - Wikipedia (underline{https://en.wikipedia.org/wiki/Accuracy_and_precision})*

*In our case we use the first definition as can be seen in equation 13*

DTS measurements need to be calibrated in post-processing to derive the actual temperatures from the ratio of intensities of Stokes and Anti-Stokes. Hence in this context I would avoid using the word calibration to not confuse the reader.

*We change calibration into "for which can be compensated"*

p.12 l.13: If RMSD is the root-mean-squares deviation, how was Eq. 15 derived and how does this equation represent the precision? In Eq. 15 the precision of both instruments are combined, even though $\sigma_p$ should represent the precision of only the AHFO technique.

*The RMSD is derived with the following reasoning:*

*The measurements of $U_{DTS}$ include the natural and DTS instrument variability*
*The measurements of $U_{sonic}$ include also the natural variability, and a small (negligible) sonic instrument variability. By subtracting both, the DTS instrument variability is the part which is left.*

*For clarification we added a line in the revised manuscript*

Throughout the manuscript $\sigma_p$ is used as a synonym for or parameter representing precision. However, it is counter-intuitive that $\sigma_p$ is decreasing for higher precisions.

*We understand the point, but mathematically what we do is correct. We will be more careful with the choice of words, by using words like improves or decreases.*

**Figures**

Figure 1: a and b are missing within the figure

*Will be changed accordingly.*

Figure 2: I would make this figure at least smaller. I don't think this figure is necessary as technical specification of the fiber-optic cable is given in the text.

*We think this is useful information for a reader less familiar with DTS. But could be removed on request.*

Figure 3: I like this figure very much, however I am missing the connection to Eq. 1. Breaking Eq. 1 into the relevant elements of the energy balance of the fiber and marking those elements with colors or boxes in Fig. 3 makes it even more powerful

*We will color the equation in a red and blue part, hence incoming and outgoing energy.*

Figure 4: Fig. 4a and Fig. 5a as well as Fig. 4d and 5d are identical. Directional sensitivity is not corrected for an attack angle of 90° angle as this is considered the optimal attack angle, hence showing 4a in this context does not make sense. Why are the symbols different between Fig. 4 and Fig. 5?

*We decide to remove figure 4a and only focus on 4b and 4c to visualize the difference in directional sensitivity equation.*

*For consistency throughout the paper we choose the select one marker per angle.*

Figure 5: comparing the subfigures is not easy because they are basically looking the same (also R2 is relatively similar). It is already shown in Fig. 4b and c that the directional sensitivity can be corrected by Eq. 13 and Fig. 5 is not adding new content. I also did not see further description of Fig. 5 nor discussion of it. So I suggest to take this figure out, unless Fig. 5 is described and discussed.

*The reason why we would like to add figure 5 is that we would like to show that AHFO works for all angles, without a big reduction of performance as you correctly mention. We added a sentence to the revised manuscript. And we propose to move it to the appendix. (as done in the revised manuscript)*

Figure 6: The effects of temporal averaging are shown and discussed in Fig. 8, however this is not done with Fig. 6. Take it out.

*We propose figure 6 to the appendix as well. (as done in the revised manuscript)*

Figure 7: The symbols are too small to see the filling you are using for different heating rates. Better use opaque colors like in ΔT = 2°C

*We will improve the figure.*

Figure 8: see comments on Fig. 7

*We will improve the figure.*

Figure 9: I would rather add $\frac{1}{\sqrt{n}}$ for each heating rate to Fig. 8 than adding Fig. 9a. Each heating rate is following a $\frac{1}{\sqrt{n}}$-decay, so when normalizing by the heating rate the only logical outcome is Fig. 9a. Hence Fig. 9a provides no new content in my opinion. The same appears to me for Fig. 9b.

*We think these figures help following the reader in understanding the steps we make to derive $C_{INT}$.*

Figure 10: I would show the same figure, but without spatial averaging. From this paper people should know that the precision can be further increased by averaging spatially or over time.

*Since all our figure are based on five sample points, we choose to keep this for consistency.*

**Specific comments**

p.1 l.5 : "operational conditions": I would rather name the conditions: heating rates and attack angles

*We add this to the abstract*

p.1 l.8-9: Under which conditions? For all conditions?

*We added for all conditions*

p.1 l.9-12: "We conclude...": no new content. We already know this from Sayde et al. (2015). What is your new contribution?

*We change conclude into we confirm. Furthermore, we add one sentence:*

*"We present a method to guide with AHFO settings in fieldwork preparation, such that data with acceptable precision is acquired."*

p.2 l.11: "High-resolution...": add spatially

*We changed accordingly*

p.2 l.31-35 & p.3 l.1-5: this paragraphs is not well organized

*We added a sentence*

p.3 l.19-20: "...to create the temperature difference needed to determine wind speed..." a minimum needed ΔT is not determined in this study. I think it is rather: "... to create the wanted temperature difference of at least 2°C..." or similar.

*We changed the sentence into:*

*"An electrical current (I) is passed through the heated cable, to create the temperature difference that is necessary to measure wind speed (ΔT, e.g., 2°C)."*

p.3 l.33: "steady state flow" -> maybe add variability of the wind speed from ultrasonic anemometer measurements

*Good suggestion, we will add it to the final paper.*

p.5 l.5: As you are mentioning the calibration set up here, which calibration set up did you use? single-ended, double-ended or duplexed?

*For a single ended setup. We added this to the revised manuscript.*

p.5 l.19-20: The captions are identical.

*We changed this*

p.6 l.11: Why is $T_s$ used for the heated cable while $T_f$ is used for the unheated cable/air temperature? f can be associated with fiber which can cause potential confusion. Maybe use $T_a$ for air temperature and $T_f$ for the temperature of the heated fiber-optic cable.

*For consistency we used the notations from Sayde et al. (2015), therefor we would like to keep it like this.*

p.8 l.13-15: I do not understand the meaning of this sentence.

*This sentence means that the used formula is valid for higher wind speeds.*

p.9 l.2-3: "...especially in outside atmospheric experiments." -> I disagree. When using a fiber-harp as a set up like the study of Thomas et al. (2012) to measure wind speeds, attack angles are 90° and no directional sensitivity has to be taken into account.

*We do not understand why the setup from Thomas et al. (2012) would completely compensate for the angle of attack. Or do you refer to Zeeman et al. (2015)?*

p.10 l.12-15: Why can the measurement error be assumed to be constant in the lab, but not in outdoor deployment? Further, here ΔT is the measurement error of the DTS, while later on p.13 l.13-14 the parameter $T_{error}$ is introduced as the constant concerning the performance of DTS. What is the difference between those two constants?

*We agree this was unclear, but σt = $T_{error}$. $T_{error}$ is depending on the DTS machine, and should also be constant outdoors. We remove the second part of the sentence.*

p.11 Eq. 14.: what does the overline indicate?

*The average over all measurements for one specific wind speed. We clarified this in the new text.*

p.12 l.1: what is the "uncalibrated data set"? Uncalibrated DTS measurements? Or were the directional sensitivity of the measurements not corrected?

*This is indeed unclear, we meant that the proposed directional sensitivity equation is not yet fully optimized (parameter $m_1$). To prevent miscommunication, we now leave out uncalibrated.*

p.12 l.1-3: why should the energy loss due to free convection be different between different attack angles?

*This could be for the same reason that we use a directional sensitivity analysis for calculating the wind speed.*

p.12 l.5: even when comparing the other attack angles to the 90°, I can see no clear dependency of the bias. 45° has the smallest bias, while 30° and 15° have more or less the same bias. Hence, I would argue: In hotwire anemometry the bias for small attack angles is high and decreases with increasing attack angle. However, even if this effect is taken into account for AHFO measurements, there seems to be more sources of error causing different behaviour of the bias for different attack angles, but with no clear pattern.

*We agree that the bias is not completely solved yet. We point at free convection because all measurements have a positive bias which suggests some energy term is neglected while could not be neglected. Some energy terms might be angle related and some not. Also different temperature differences might influence the amount of energy loss. This requires more detailed investigation in the future, for fully optimizing the method.*

p.12 l.8-9: What does "extensive calibration" mean? What else could be done or for what else can AHFO be corrected?

*Here is referred to equation 12, but also as mentioned in the comment above further improvement is expected to be possible.*

p.13 l.3: "The precision increases to a σp less than 5% by averaging over time" -> where is this shown?

*See figure 8*

p.15 l.6-7: "However, given the result that the increase in precision behaves independent of ΔT and the averaging time, it is possible to make a prediction for the precision of future work." -> this contradicts Fig. 8 and the derived error prediction equation. The precision only behaves independent of ΔT and the averaging time, when the precision is normalized by both parameter.

*What we mean here is that the behavior is the same for ΔT and averaging time, but not necessarily the values. We make use of the behavior to predict the values for other ΔT and averaging times.*

p.17 l.2-3: I do not understand the connection of Fig. 10 and the missing factor of free convection in the energy balance of the fiber. Please clarify.

*When there is no wind speed at all, the free convection term becomes more dominant compared to the forced convection term.*

*For clarification we changed to sentence.*

p.17 l.9-13: I would add, that either a set up like a fiber-harp has to be used, for which no reference device is needed as the attack angles are 90°, or a reference ultrasonic anemometer or even multiple ultrasonic anemometer have to be used to account for different angles of attack.

*If we look at the Zeeman et al. (2015) paper, we understand the fiber harp can compensate for horizontal wind variability (u and v), as we mention in our AHFO outdoors section. However, the vertical wind (w) variability component is not included, which might be important for more complex terrains, like forests.*

p.18 l.1-3: Why is complex terrain specifically mentioned, while field experiments with the AHFO technique is valuable in any terrain?

*That is true, but what is meant here is that by measuring distributed, more complex flow structures can be captured, compared to with a sonic anemometer, which assumes the point measurement holds for the whole measured area.*

**Review 2**

The manuscript describes a controlled laboratory evaluation of a recently introduced technique for wind speed measurements using actively heated fiber-optic cables combined with fiber-optic temperature sensing (AHFO-DTS), similar to hot-wire anemometry. The evaluation considers the wind speed, the angle between the mean flow and the sensing cable and the temperature offset between heated and unheated sections of the sensing cable in the experimental design. The results include a simplified model to help plan the heating requirements for experiments in similar conditions. The study highlights aspects to consider for planning real-world applications of AHFODTS and as a major outcome shows that the potential bias due to sensing cable pitch angle may be low and previously used constants should be reconsidered. However, not all laboratory outcomes can be immediately translated to a real-world field application. The authors added a section to discuss this, but should elaborate on the specific conditions in the wind tunnel (compared to real-world) in more detail, to help the reader. The need to include certain outcomes, particularly time averages of wind speed estimates, is unclear to me. Removing those would improve the focus of the text/figures and reduce redundancy. White noise can be mitigated by spatial/temporal averaging, but the introduction does not clearly mention the relevance for this study; I think that the outcomes for high-end resolutions of AHFO-DTS (currently 1Hz, 0.3m) are conclusive enough without it. After major revisions the manuscript should be considered for publication, and I expect it to be a very helpful contribution for those interested in spatial wind speed measurements using this novel technique.

*Thank you for your kind words and constructive comments.*

*Thanks you for your suggestion to elaborate on the specific conditions in the wind tunnel. We will add turbulence statistics of the wind tunnel in the new paper, as also suggested be reviewer 1.*

*We agree averaging over time and space was unclear, but in the revised paper we tried to solve this. By explaining the need to $C_{DTS}$ we need to average over time and space. Finally, with these results we present a method for estimating the precision of future experiments. We now emphasize this more throughout the manuscript.*

**Specific comments**

In their current form, Fig 4, 5 and 6 do not adequately highlight the differences between the applied corrections or experimental settings. Actually, the uncorrected regression (Fig. 4a) fits the range of observations in the center, making it look like a better fit. Please improve the figures.

*We agree and would like to point at the proposed changes to remove figure 4a and figure 5 and 6 to the appendix. See also see comments review 1.*

The reason to present results for different averaging periods was not clearly introduced. I suspect those averaging details can be left out and this would help focus the result/ discussion section (remove Fig 6 through 9). The precision and accuracy are most meaningful at the highest resolution, for combinations of different angle, wind speed and temperature offset (and heating rate) settings, and could perhaps be summarized in a single figure or table.

*Please consider the answer above why we average over time and space.*

A fixed temperature difference between the reference and heated probe would require a feedback system that adjusts the heating rate according to previous or expected wind conditions. 1.In the laboratory setup, was this adjustment in heating rate made manually or automatically How accurate could this be set and were heating rate readjustments made during the 10-min steady periods? The text suggests that under low angles of attack, the heat exchange is less efficient and, consequently, a lower heating rate can be applied to achieve a 2/4/6 K differences compared to the reference. Please discuss the impact of such variable heating rates on the results.

*For each wind speed setting we manually choose a heating rate which roughly gave a 2,4 or 6 °C difference. During this experiment the heating rate was not adjusted.*

*It is indeed correct that the heat exchange is less efficient with small angles. We compensate for this directional sensitivity with our directional sensitivity equation 12. Also $C_{DTS}$ for our final equation includes this.*

2. In real-world applications, with more variable wind and radiation conditions, such a feedback system may become challenging. A constant heating rate (variable temperature difference) is perhaps more practical. Therefore, could you also present your results expressed as a function of heating rate, instead of fixed temperature offset?

*In our experiments we already work with a fixed heating rate to create the temperature difference. As we discuss AFHO outdoor section we propose that for outdoor applications you estimate the maximum expected wind speed and based on this choose one desired precision, which results in one heating rate which should be sufficient.*

*In the end equation 22 (revised manuscript) contains the heating rate.*

Section 2.2.2 includes both a modification - a different set of constants - and a simplification of the set of equations from Sayde et al (2015) for applications in a wind tunnel. This is not reflected by the section title. Perhaps move the proposed modification (with figure) to the results section or rephrase the section title.

*Thanks you for this suggestion, this is changed in the revised manuscript*

Why were there no (additional) reference measurement made downwind of the heated cable? In a controlled environment, this could help identify feedback between ref and heated cables in relation to separation distance. In a real-world application, would this make a setup less sensitive to wind direction shifts, or is the relative position of both sections irrelevant? Given the long-standing expertise among the authors, could further recommendations be made for follow-up evaluations of AHFO-DTS inside or outside the wind tunnel?

*In this study was focused on a one-dimensional case to study in detail the precision of the method. In the AHFO outdoor section we already make some suggestion to use 3D AHFO set ups to make the method less sensible to non-perpendicular wind directions.*

**Technical corrections**
p1l1/p1l15: Either Active or Actively, choose one for the AHFO definition.

*We will use Actively.*

p2l21: remove the comma after '(2015)'

*Removed*

p3l16: 'cable (which encloses the FOs)': suggestion 'FO cable'

*We changed this into fiber optics*

p3l27: 'angles of attach': suggestion 'angles of attack with the mean flow'

*Good suggestion, we add angles of attack with the mean flow direction.*

p3l27: Were the ref and heated cable also 8 cm apart in mean wind direction under these slanted angles? From the following sentences this is not immediately clear. If not, would it matter if they would come closer together? Please explain.

*Yes, the separation distance was always 8cm. We will add a top view picture in figure 1.*

p3l33-p4l1: 'The wind speed in the wind tunnel was fixed at a constant value to create a steady state flow.': perhaps remove, or rephrase to be more specific. Do you mean: The engines in the wind tunnel were set to a constant rate, generating a steady (often laminar) flow after some time and, as a result, a dynamic steady state heat exchange of the FO cables with the moving air.

*We mean that the wind speed in the wind tunnel is at a fixed wind speed. In the final paper we will add some turbulence statistics.*

p4l2: This paragraph contains a listing of three different experimental settings. Please help the reader by presenting them as such. 'First,... Second,. . .' or other rephrasing.
 p4l2: 'Furthermore, for all wind speeds and angles the temperature difference between the reference and heated section, delta T, wat set at 2, 4 and 6 K in order to evaluate the importance of hotwire signal magnitude.'

*We changed this sentence*

p4l8: 'machine': maybe 'instrument'?

*We prefer machine for consistency.*

p4l9: 'One cable segment was heated.', suggestion: remove this first sentence and rephrase the second sentence 'The stainless-steel casing of the heated section was ...'

*We rephrased the sentence*

p4l10: Steel or other wire material?

*The AWG cables are from copper, we added this.*

p4l11: a temperature difference should be reported in K (Kelvin). Also, those values were already mentioned. suggestion: change ' to a fixed level, either 2, 4 and 6 °C, depending on the setup' to 'at fixed levels'.

*We will change degree C to K in the final manuscript. Also we changed the sentence.*

p5l4: what is an 'ambient bath'?

*A bath which is at room/air temperature, which is needed for internal DTS calibration.*

p4l7/p5l8: double ended DTS measurement was applied (p4) but not used (p5). Please put these details together in a single paragraph. Are there quantitative criteria to reject the double ended method using the arguments here, perhaps described in literature (citation)?

*During analysis at was found that the signal loss at the splice was not symmetric/clean. Therefor based on expert judgement we decided it was safer to not use this data.*

p5l21-p5l22: Somewhat vague. 'An energy balance ... advective energy transport from the heated cable, ...'. suggestion: 'An energy balance method ... heat dissipation from the heated section, ...'. The heat dissipation may not be a fully advective process.

*We rephrased the sentence.*

p6l14: please rephrase, introduce the abbreviation last, suggestion: 'The Nusselt number, Nu, is the ...'

*We rephrased it.*

Match the citation style according to journal guidelines, throughout the text. Particularly the year notation with nested parenthesis seems odd, p2l3 should be "... (e.g., Goodberlet et al., 1989)".

*We will check this and change if necessary*

- Fig. 4, Fig. 5, Fig. 6: Add a meaningful indication of the statistics of these point clouds - add means, or convert the presentation to boxplots - or remove the figures.

*We decided to remove and replace the figures to the appendix as mentioned.*

- p10l1-p10l21: contains details already mentioned before and details that should better be placed in an introduction.

*We restructured the whole method section in the revised paper.*

- Eq 12 and 13: where is $u_N$ defined?

*See equation 10 in the revised manuscript, we changed the description of this equation.*

- p10l26: Is the center of the wind tunnel cross section also the center of the ref/heated FO cable sections?
Why use a section of 0.9 m and not 0.3 m, the advertised resolution
of the system, nearest to the location of the sonic anemometer sensor path?

Please show that the extended spatial range did not have an impact on the outcomes. Also, by taking a fixed length of FO, the observed positions in the cross section of the tunnel changed with angle; the 15 degrees angle of attack setting would only integrate approximately 0.2 m vertically.

Since the bottom of the fiber was attached to the tunnel
(p3l26-30), were the positions of the center of the ref/heated sections determined at different positions along the optical-fiber length for each angle?

*Yes, this is always in the center.*

*Please consider the changes in the revised manuscript as mentioned in review 1.*

*We will add a table to the new manuscript which shows that this extended spatial range is justified.*

*We always used the 5 measurements close to the sonic and in the middle of the wind tunnel. So this changes for every angle.*

p10l26: 'Only the temperatures from the middle of the wind tunnel are used, to prevent using data with side/boundary effects.' If the center of the observed sections, independent of the angle of the cable, were centered at the same position in a cross-section of the tunnel as the sonic anemometer, please state this explicitly in the method section.

*We added a sentence stating this: We always used the 5 measurements closed to the sonic and in the middle of the wind tunnel.*

p10l27: Referring to 'data' here is not very specific, please rephrase. suggestion: 'AHFO-DTS derived wind speed estimates'.

*We changed this accordingly.*

p10l28: Averaging to 30 sec, including Fig. 6+7, shows no new information: better to leave it out?

*As mentioned Figure 6 will be moved to the appendix. Figure 7 is important because it shows the bias difference between different angles.*

Fig. 5c: In the review copy, it seems like the 16 m/s data included values that had not reached steady state yet (variability in Usonic, < 16 m/s). Could you please verify?

*We will check this for the final paper.*

p12l11: 'The precision is calculated for all 120 à T , angle and wind speed combination, using Eq. 15.' Please refer to the equation after it has been defined.

*We will move this equation.*

- p13;14: Are these numbers for Terror computed, based on the calibration bath sections? Or a specification of the instrument?

*These are given by the instrument specifications.*

**Revised manuscript**

The manuscript of van Ramshorst et al. (2019) is already partly revised based on the comments of review 1 and 2.

[revised manuscript text omitted]

---

## Referee Report (RR1)

**Review report: Wind speed measurements using distributed fiber optics: a wind tunnel study**

**Author of the paper: van Ramshorst et al.**
**Journal: Atmospheric Measurement Techniques**
**Manuscript DOI: 10.5194/amt-2019-63**

**General Comments**

The study of van Ramshorst et al. investigated the actively heated fiber-optic (AHFO) technique and estimated its accuracy and precision under controlled airflow conditions by comparing to a three-dimensional ultrasonic anemometer. A valuable error prediction equation for the wind speed measurements at different heating rates was developed, as the heating rate can be a limiting factor for long cables. This equation is also accounting for averaging over space or time which further increases precision. They conclude that AHFO measurements are reliable in outdoor deployments when correcting the measurements for directional sensitivity with a ultrasonic anemometer, choosing the right heating rate and spatial or temporal averaging.

Distributed temperature sensing (DTS) measures temperatures along a fiber-optic cable spatially continuously and can be used in various fields. Especially for atmospheric research this technique offers new insight into the temperature field and thus was implemented in many studies. By using the AHFO technique, wind speed measurements can be added to the system. As the community using the DTS and AHFO technique is growing, the study of van Ramshorst et al. is important for users to be aware of the accuracy, precision and limitation of this technique. Hence, the paper is valuable for our community.

The author did do a good job on the first revision. The reviewed manuscript nicely rearranged the sections and made the manuscript more reader friendly. Also the additional turbulence statistics of the wind tunnel makes it a stronger manuscript. However, there are still some issues regarding precise description of the setup, definition of parameter, presentation of some graphs, and the error prediction function. Details are given below for each section. I recommend to accept the submitted manuscript after major revisions.

**Terminology**

- advective heat loss vs. convective heat loss:
  the correct terminology as is also used in the referenced literature of this manuscript is "convective".

**Abstract & Section 1**

- p.1, l.2-3: "...better characterization of fine-scale processes." - this is a very specific comment, but I think sonic anemometers can determine fine-scale processes better in

time (20 Hz resolution vs. 1 Hz of DTS), while AHFO technique does have spatially continuously distributed measurements. Hence, AHFO can give inside into the wind field even a network of sonic anemometer might not be capable of on time scales > 1s. But I argue that a general statement like "better characterization" might not justify if not pointing out that the focus is on the spatial scale.

- p.1, l.4: "In this work, ..." this sentence is redundant
- p.1, l.6: "wind speed, angles of attack, and temperature differences" → Oxford comma!
- p.1, l.7: 1-s time scale
- p.1, l.9: the correlation numbers can only be found in the abstract and conclusions, but not in the results section. Please refer to that correlation numbers in the result section, otherwise those high correlation numbers seem to come out of nowhere.
- p.1, l.11-12: "AHFO allows for characterization...complex terrain..." - AHFO can be deployed in any terrain, so I do not understand the focus on complex terrain. I would remove that sentence and only mention the last sentence which is way more important.

- p.2, l.3: "Goodberlet et al. (1989)" - aren't there also more recent paper commenting on that?
- p.2, l.16: 1-s and 0.3-m resolution
- p.2, l.21-22: I do not understand the meaning of that sentence
- p.2, l.25-27: the statements in those lines are very specific and belong into material and methods
- p.2, l.29: magnitude of convective heat loss
- p.2, l.31-25: This paragraph can be shortened : "The heat transfer model assumes a flow normal to the fiber. Hence, non-normal angles of attack need to be accounted for by using directional sensitivity equations. Following the recommendations of Sayde et al. (2015) we tested different directional sensitivity equations from hotwire anemometry in the controlled setting of our experiments" - or similar

- p.3, l.7: remove "in complex terrain" - AHFO can be deployed in any terrain
- p.3, l.13: "precision of future experiments" - precision of AHFO experiments

**Section 2.1 & 2.2**

Both sections can be shortened by mentioning important literature and specifically focus on the parts which were changed in this manuscript to improve AHFO measurements. Section 2.2.1 has no new content and Figure 1 is from the paper of Sayde et al. (2015) altered in a incorrect way. The presented equations are also more or less a copy of Sayde et al. (2015). The most important changes are mentioned in Section 2.2.2, hence I would argue to remove Section 2.2.1 or only mention the most important points (like the energy balance) instead of the complete derivation of the equation determining the wind speed from AHFO measurements as already done by Sayde et al. (2015).

I also have the following specific comments for those two sections:

- p.3, l.17-25: I think this paragraph is unnecessary for the manuscript focusing on AHFO. One sentence and pointing to the publication of Selker et al. 2006 is enough to explain

the basic measurement principle of DTS. The last two sentences of the paragraph can be inserted somewhere else.

- p.4, l.1: "heat the fiber" - heat the FO cable
- p.4, l.2-3: "Also the creation of ..." - very confusing statement
- p.4, l.6: "which van be" - which can be
- p.4, l.10; p.5, l.5; p.5, l.13; p.5, l.14; ...: "advective" - "convective"

**Section 2.3**

The whole section needs to be revised. Some information is unnecessary and could be addressed in one paragraph instead of mentioning it in different paragraphs. For example p.7, l.17-23 define the heating rates $P_s$ creating $\Delta T$ for different wind speeds, however, the actually used $\Delta T$ are mentioned p.8 l.8 and the corresponding $P_s$ p.9, l.1. Also the total number of experiments could be mentioned at the end after defining/mentioning all setups instead of consecutively summing up the experiment setups (p.8 l.2, l.6, l.8). Further, in this section FOs is used as a synonym for FO cable or FO cores, however, was not defined as such.

The FO configuration as also introduced/proposed by Hausner et al. (2011) is introduced in p.8, l.10-11, then changed to a double-ended configuration (p.9, l.12), but actually a single-ended configuration (p.9, l.12) was used. This is more than confusing to the reader especially by mentioning it in different paragraphs. I would propose to mention the setup, only the actually chosen calibration method (as proposed by Hausner et al. (2011)), and step loss correction in one paragraph.

I also have the following comments and questions:

- p.7, l.17-23: This paragraph should be revised. It is confusing to the reader. The main point should be the definition of the heating rate $P_s$ fixing $\Delta T$ for different wind speeds. Accordingly, $\Delta T$ is representing $P_s$.
- p.7, l.17: "The angle of the fiber..., wind speed, and heating rate were systematically changed" - This sentence is fine, however you start defining heating rate, then the angle of attack and then your wind speed settings... A reorganization makes it more reader friendly.
- p.7, l.26-28: I am not sure if this sentence is important unless you add that those parts were excluded from analysis to avoid artifacts of the fiber touching the mounting material.

- p.8, l.2: "The AHFO wind speed measurements can be calibrated..." - calibrated is not the correct use here, as DTS data itself is also calibrated. I suggest "adjusted" or "verified". Also removing that sentence would not affect the meaning of the paragraph.
- p.8, l.10: The manufacturer of the FO cable is missing.
- Figure 3: The figure is not necessary for understanding the described method, Figure 2 is already showing the needed information. Either remove or put into Appendix.

- p.9, l.1-6: this paragraph is describing the heating rate and how it can be estimated from the resistance of the FO cable. However, the heating rate is first mentioned p.7, l.22. I suggest to reorganize the section and to shorten this paragraph. The most important information is in the last sentence.

- p.9, l.16. "splice loss": was a step loss correction performed?

**Section 2.4 & Section 3.1**

I do not completely agree with p.10, l.8-10. Sayde et al. (2015) never adjusted DTS measurements, they adjusted the sonic anemometer measurements and compared them to the AHFO measurements. Hence, the statement is not completely correct. However, I agree that the equation can be used to adjust for different attack angles and adjust/correct AHFO measurements if a sonic anemometer is near the AHFO setup.
Figure 4 should be adjusted by showing violin plots or boxplots for each wind speed configuration. The shown dots can not show the distribution of the DTS measurements in a clear way. Are real measurements of the sonics shown or the proposed fixed wind speeds?

**Section 3.2**

The definition of $\sigma_a$ and $\sigma_p$ should be moved into material and methods. The result that $\sigma_a$ and $\sigma_p$ are dependent on wind speed and averaging over space and time and the corresponding discussion can remain in Section 3.2.
On page 11 line 1-6 present details on the duplex FO configuration and that only the middle of the FO cable was used to estimate $u_{DTS}$, because otherwise the accuracy is decreased (Table C1). I think that this important piece of information, especially about the $90°$ attack angle, needs to be mentioned in Section 2.3. Further, shouldn't a step loss correction cover the effects of a splice?
I think Figure A1 and A2 are unnecessary as Figure 5 is giving the same information and derive the same conclusion. If the author decide to include those figures, they need to be adjusted in the same way as Figure 4.
Eq. 13 and 14 are presented in a confusing fashion by introducing dependencies which do not affect the definition of the introduced parameter. Eq. 13 and 14 define $\sigma_a$ and $\sigma_p$, however, the definition itself is not reflecting any dependency on $n_{space}$ nor $n_{time}$. This dependency is shown in Fig. 5 and Fig. 6 and should not be included in Eq. 13 and 14. The original equation from the first manuscript were better. Besides, the parameter $u_j$ remains unclear to me as it also doesn't define $\sigma_a$ and $\sigma_p$. Both are defined by $u_{DTS}$ and $u_{sonic}$. I think $u_j$ is introducing an unnecessary parameter. Further, Eq. 15 is introduced, which is changed to Eq. 16 without justification and both are different.

- p.12, l.12-14: those lines are redundant as they are already introduced on p.11, l.12-17
- p. 13, l.1-3: I do not see the justification of excluding the variation of sonic anemometer measurements by assuming it is small. If it was tested and did not change the results, I can see the justification of not considering that variation.

**Section 3.3 and Section 3.4**

The title of Section 3.3 is not representing the content. The use of an intermediate constant is still not justified. The authors response to my comment did not include a justification nor the

manuscript. The intermediate constant is defined as:

$$C_{int} = \overline{f(\overline{u_j}, n_{space}, n_{time}) \cdot \frac{\Delta T}{T_{error}} \cdot \sqrt{n_{time}}} \tag{1}$$

Hence, $C_{int}$ (or $C_{DTS} = C_{int} * 10$) is the mean of the precision over a variation of setup:

$$C_{int} = \overline{\sigma_p \cdot \frac{\Delta T}{T_{error}} \cdot \sqrt{n_{time}}} \tag{2}$$

Those constants are then included in the estimation of $\sigma_p$ again which is not justified.
Besides, how was the factor $T_{error}$ determined? Or is this number from literature?
The revised manuscript is describing the derivation of the intermediate constant very quickly without further explanation. This should be presented in further detail.
I still propose to derive the error prediction function in a more clear way. As shown in Fig. 6 for each $\Delta T$ the precision $\sigma_p$ is following a $\frac{1}{\sqrt{n}}$-line. Hence, we assume the following dependency:

$$\sigma_p(n) = \frac{\alpha}{\sqrt{n}} \tag{3}$$

with $\alpha$ being a constant different for experiment set up. We found that $\alpha$ depends on $\Delta T$ and $T_{error}$:

$$\alpha = \left(\frac{\Delta T}{T_{error}}\right)^{-1} \tag{4}$$

with $T_{error} = 0.25$ K being the performance of the DTS dependent constant and $\Delta T$ being the measured temperature difference between the cables. Hence, $\alpha$ is representing the quality factor for the wind speed measurements. The lines derived from $\alpha$ could also be added in Fig. 6 or 7. When simplifying Eq. 20 of the submitted manuscript we can assume that $\Delta T$ is mainly depending on the following parameter:

$$\Delta T = \frac{A P_s}{B u_n^m} \tag{5}$$

Combining my Eq. 4 and 5 and inserting that in Eq. 3, I derive the following error prediction equation:

$$\sigma_p(n, u_n, P_s) = \frac{B T_{error} u_n^m}{A P_s} \frac{1}{\sqrt{n}} \tag{6}$$

If I did not miss a point, no empirically derived intermediate constant has to be used for the error prediction equation.
I think the error prediction function could be tested with the existing data set by inserting $P_s$ in the error prediction function and plotting that against the 'real' $\sigma_p$ using Eq. 14 of the revised manuscript. This should show if $C_{DTS}$ is needed or not.

**Section 3.5 & conclusions & Appendix**

Section 3.5 is not introducing new content. It should be incorporated in the corresponding sections as a paragraph of discussion or incorporated into the conclusions if the statement is more an outreach than a finding.

- p.17, l.19-20: how can turbulence be fully captured by the AHFO technique? Which turbulence scales are you talking about? How should the setup look like?

- p.18, l.6-7: "Due to the way this design tool is constructed, it can be generalized for all kinds of fibers, DTS precisions, and user preferred spatial and temporal resolution."
I do not agree with this statement, because the accuracy and precision of the DTS measurements change with the use of FO cable, DTS performance, and the used calibration method. Further, the turbulence of the wind tunnel setup does not represent outdoor turbulence as also stated in this manuscript. Another point is the response time of the FO cable. The thicker the cable, the longer it takes to reach the FO core and measure the temperature change. Also the measurement location of the wind speed is important, as the noise of measurement increases with distance from the measurement device, hence with the location along the FO cable. So how should $C_{DTS}$ and the error prediction function be representative for outdoor deployments, different choice of FO cable, different setup of FO cable, or a different DTS machine? I think the statement is too strong and not justified.

- p.18, l.11: "...applications in complex terrain, allowing for..." -...applications, for example allowing for... - AHFO can be deployed in any terrain.

- Figure B2: turbulence intensity should be defined in the caption, as well as the location of the x-, y- and z-coordinate.

---

## Referee Report (RR2)

**Review report: Wind speed measurements using distributed fiber optics: a wind tunnel study**

**Author of the paper: van Ramshorst et al.**
**Journal: Atmospheric Measurement Techniques**
**Manuscript DOI: 10.5194/amt-2019-63**

**General Comments**

The study of van Ramshorst et al. investigated the actively heated fiber-optic (AHFO) technique and estimated its accuracy and precision under controlled airflow conditions by comparing to a three-dimensional ultrasonic anemometer. A valuable error prediction equation for the wind speed measurements at different heating rates was developed, as the heating rate can be a limiting factor for long cables. This equation is also accounting for averaging over space or time which further increases precision. They conclude that AHFO measurements are reliable in outdoor deployments when correcting the measurements for directional sensitivity with a ultrasonic anemometer, choosing the right heating rate and spatial or temporal averaging.

Distributed temperature sensing (DTS) measures temperatures along a fiber-optic cable spatially continuously and can be used in various fields. Especially for atmospheric research this technique offers new insight into the temperature field and thus was implemented in many studies. By using the AHFO technique, wind speed measurements can be added to the system. As the community using the DTS and AHFO technique is growing, the study of van Ramshorst et al. is important for users to be aware of the accuracy, precision and limitation of this technique. Hence, the paper is valuable for our community.

The manuscript improved substantially. It is well organized and leads the reader through the whole manuscript. I still have one major point: The manuscript propose to develop an error prediction function being valid for any kind of setup. However, the error prediction function is not tested or validated with the existing data set nor is the last point discussed accordingly. I did get a table in the authors' response to compare different approaches of the error prediction function, however, this table is not well explained. Further, no values derived from the prediction function is compared to actually measured quantities neither in the table nor in the manuscript. As the error prediction function is one main goal of the manuscript, either the authors need to explicitly state that the error prediction function needs to be validated in another experiment or another section validating the error prediction function is added to the manuscript. I recommend to accept the submitted manuscript after major revisions. More detailed comments are given below.

**Detailed comments**

- p3 l23: $T_{error}$: measurement error? how determined?

- p3 l27: choose dominant or important or be more specific

- p8 l18-31: nicely done!

- p9 l4: "duplexing" $\rightarrow$ duplexed FO core (I would stick with the earlier already mentioned phrase)

- p9 l6-8: the argumentation to treat the 90° angle different than the others is a bit thin in my eyes: maybe speculate why the 90° angle had lower precision (sharper bending of the FO cable maybe?) and thus justify your decision. Or treat all attack angles the same and shorten all data down. Why should the splice only affect the 90° angle? What if the others were also affected just a bit less?

- p9 l12-14: "indicating and..." $\rightarrow$ "indicating an"; How is the actual spatial resolution defined? Nyquist-frequency?

- p9 l14-17: I think mentioning the goal of an error prediction function is more useful than already mentioning the unique constant which is used in the error prediction function later. This will make it easier for the reader to follow your manuscript. Especially the last sentence is confusing to me. Is one constant more representative if I am averaging, but if I am not averaging, it isn't?

- Figure 3, B1, B2: a 1:1-line would be very helpful

- Figure B2 and p11 l1-5: It would be easier if B2 has two figures a) 1-s data and b) 30-s data. It is impossible for the reader to combine all four plots in B1 into one plot and compare it to Figure B2.

- Figure 4: Why is accuracy of 90° & $\Delta T = 2$K getting worse for higher averaging time? I would suspect the opposite as also proposed by the mansucript (p11 l16-17). I would like that this is at least mentioned/discussed.

- p11 l4: coefficients of determination: I guess that is a linear regression and you show the R-values? What is the derived slope and offset? Please also add this information or at least add the 1:1-lines to the graphs if slope is close to unity anyway.

- p11 l16-17: You need to discuss this statement further and use another phrase for "extensive calibration" which is not accurate as different calibration methods can be applied to the FO measurements before even computing wind speeds. I would also argue that maybe the temperature signal needs to be averaged over time before computing the FO wind speed. Would that also increase the accuracy? If not, why? Was this also tested?

- p11 l18: I would argue that the dependence between accuracy and averaging time is less pronounced than the one between the precision and averaging time scale, not that the accuracy is constant over time. Besides, it is confusing that $\sigma_a$ is given in percent while $\sigma_p$ is given in decimal numbers, but percentage values are given in the text. Please make uniform for both parameter.

- p11 l20: The last sentence is redundant. Either comment in further detail what Eq. 13 is stating and what dependencies can be determined from Figure 5 or remove the sentence.

- p12 l1-3: The meaning of those sentences for the analysis is hard to understand. Further, is $j$ used instead of the measured $u_{sonic}$?

- p12 l6: rephrase: "the precision was averaged over all wind speeds which is justified, because $\sigma_p$ is normalized by the mean wind speed, hence any linear dependency should be removed" or similar; "... for all $\Delta T$..." → "... for each $\Delta T$..."

- p12 l7: "..., with ..." The sentence is redundant to some degree. The colors and symbols are already showing why there are 12 different points for each $n_{time}$.

- The following is only a suggestion/thought: What about dropping the attack angles for $\sigma_p$? They are not further discussed as you already account for them in the earlier section. So should that maybe not be considered moving on? It is kind of distracting from the main object of different heating rates and averaging time/spatial scales.

- p13 l10-15ff: If this statement is true, then it needs to be further discussed and why it can be applied to different settings. The statement is also refering to an Equation which is introduced later in the manuscript. So I suggest to insert this paragraph at the corresponding location to Eq. 21 and into Section 3.4, respectively. Also $C_{int}$ has a wide spread and needs to be further discussed. Again I suggest to insert a plot using the prediction function for $\sigma_p$ and the actual derived $\sigma_p$ for all averaging scales to show the strength and accuracy of the prediction function.

- Figure 6a: y-axis label is not representing what is actually plotted: $\sigma_p \cdot \frac{\Delta T}{T_{error}}$

- Figure 6b: same y-axis problem as Fig. 6a. Further, how can you justify that your proposed constant has a spread from 1.1 to 2.2? This needs to be mentioned and discussed.

- p14 l6-7: This is a contradiction as Figure 5 is showing the exact opposite: Higher $\sigma_p$ for lower $\Delta T$, lower $\sigma_p$ with increasing $n_{time}$. $\sigma_p$ can be estimated from those variables, but $\sigma_p$ is not independent of those.

- Section 3.5: Maybe dew fall on the fiber needs to be considered? Water droplet on the fiber will for sure affect the measurements altering the heat loss of the unheated fiber (assuming the water droplets quickly evaporate from the heated fiber).

- p17 l10: directional sensitivity compensation can only be applied if the angle of attack is known demanding ancillary measurement devices. Please add.

---

## Referee Report (RR3)

**Review report: Wind speed measurements using distributed fiber optics: a wind tunnel study**

**Author of the paper: van Ramshorst et al.**
**Journal: Atmospheric Measurement Techniques**
**Manuscript DOI: 10.5194/amt-2019-63**

**General Comments**

The study of van Ramshorst et al. investigated the actively heated fiber-optic (AHFO) technique and estimated its accuracy and precision under controlled airflow conditions by comparing to a three-dimensional ultrasonic anemometer. A valuable error prediction equation for the wind speed measurements at different heating rates was developed, as the heating rate can be a limiting factor for long cables. This equation is also accounting for averaging over space or time which further increases precision. They conclude that AHFO measurements are reliable in outdoor deployments when correcting the measurements for directional sensitivity with a ultrasonic anemometer, choosing the right heating rate and spatial or temporal averaging.

Distributed temperature sensing (DTS) measures temperatures along a fiber-optic cable spatially continuously and can be used in various fields. Especially for atmospheric research this technique offers new insight into the temperature field and thus was implemented in many studies. By using the AHFO technique, wind speed measurements can be added to the system. As the community using the DTS and AHFO technique is growing, the study of van Ramshorst et al. is important for users to be aware of the accuracy, precision and limitation of this technique. The paper is very valuable for our community and I would like to see the manuscript being published.

The manuscript again improved substantially. Especially Figure 8 was the needed piece for validating the proposed prediction function. I have one major point: $T_{error}$ is an important value for the error prediction function, but is barely described. The number is given by the manufacturer but it is not specified if a calibration and which spatial and temporal resolution was applied. Accordingly, $T_{error}$ is not a universal number and can only be used if the setup of the manufacturer is known. I recommend to either ask the manufacturer or derive $T_{error}$ by the bias within the calibration bath which is a temperature controlled environment. I think that the error prediction function is only correct if $T_{error}$ is defined correctly. Further, I have three minor points: 1) the dependency between $\sigma_p$ and $n_{space}$ is not shown → I would recommend to add a second x-axis on Figure 4, Figure 5 and Figure 6 showing $n = n_{time} * n_{space}$; 2) Checking all equations for consistency and correctness; 3) Careful language: differentiate between what the paper can offer and what are potential future steps.

I am not a native speaker so I am not sure about the commas and other grammatical errors of the manuscript. I do not want to discourage the author by the amount of comments and number of reviews I demand. The manuscript is really improving. I recommend to accept the submitted manuscript after major revisions. I think that not another major revision is needed afterwards. More detailed comments are given below.

**Detailed comments**

- p1 l11-13: AHFO can measure the mean horizontal wind speed but not the horizontal wind speed. As the horizontal components are usual an order of magnitude stronger, it is so far not possible to measure the vertical wind speed component. At least I did not see a publication doing this so far. I also did not see this being done in your publication. So promising that AHFO combined with DTS can derive a turbulent heat flux estimation is promising more than can be done so far especially for reader which are new to this measurement technique.

- p2 l10: the "as well" can be taken out.

- p2 l11: spatial**ly**

- p2 l11-13: The underlying assumptions of what?

- p2 l21: I would remove "Recently"

- p2 l21-29: I like this paragraph very much.

- p3 l6: spatial**ly**

- p3 l8-9: as already mentioned above: how can you derrive the sensible heat flux from DTS + AHFO measurements

- p3 l15: the abbrevation DTS was already introduced earlier

- p3 l15-19: just a thought (does not need to be included): Isn't bending also a source of signal loss for the fiber? Further, when averaging over space is needed coiling up the fiber could potentially be useful. Hilgersom et al. 2016 "Practical considerations for enhanced-resolution coil-wrapped distributed temperature sensing". But maybe this is opening a new topic and is a bit too far off topic of this manuscript.

- p3 l23: $T_{error}$: machine specifications: How did the manufacturer define this? If this is not clear you can not use this number for your calculations! I think it is better to define $T_{error}$ by the mean bias within one of the calibration baths using the proposed averaging over time or space which might be more fair than a constant number independent of spatial and temporal averaging of your measurements. If $T_s - T_f$ is lower than the bias within the calibration baths then this can also be considered a low signal or high $\sigma_p$. If $T_{error}$ is defined this way, maybe the instantaneous constants are not necessary and your function could be applied to any setup.

- p3 l26-27: review this sentence. It contains "can" twice and "can cause that other ways of..." also sounds not logical to me.

- p3 l26: "radiation" $\rightarrow$ "radiative heat loss,"

- p3 l30: the measurement error can only be compensated when using another device. Please mention this. With a varying wind field or within a canopy a lot of reference devices might be necessary.

- p4 Figure 1: This is the figure of Sayde et al. (2015)! "...balance, based on..." $\rightarrow$ "...balance from..."

- p5 l9-10: The Nusselt number is defined incorrect: ratio of convective to conductive heat transfer.

- p5 l20: the unit of the temperature range is incorrect. I think it is degree Celcius, not Kelvin.

- p6 l1: I would start a new paragraph here

- p6 l9: "...we assume that there is a uniform..." → uniform in space or in time? In time could be verified by the ultrasonic anemometer data

- p7 l7: Is there a documentation of that windtunnel somewhere? Or a webpage to get further information?

- p7 l9-10: "... two segments of one cable (which encloses the FO cores) were placed 8cm apart..." → this formulated a bit confusing. "During the experiment the heated and the unheated reference cable segment were placed 8cm apart. The FO cable has two FO cores, hence, each cable segment could be sampled twice."

- p8 l6: duplexed FO core: was this splice checked for a step loss?

- p8 l11-12: so only offset correction of the FO cable was performed? Was the differential attenuation of the FO cores checked and accounted for?

- p8 l18-31: I like this presentation a lot, however, I think the bullet points could still be shortened by not using full sentences making the amount of experiments even more clear.

- p8 l27: Outside deployment: definitely turbulent conditions → please mention one sentence why this is not an issue comparing the wind tunnel experiments with outside deployments.

- p8 l33: "Splices between ends of fiber optic cables..." → "Splices connecting two fiber-optic cores..."

- p9 l1-2: "However, in processing of the raw DTS data...." → "But in our setup the signal loss of the splice connecting the fiber-optic cores of our cable at the end of the array was not the same in both directions." - Did you introduce earlier that two cores were spliced together to create a duplexed setup?

- p9 l2-3: "Due to this assymetrical structure..." → I think it was never introduced that potentially two channels can be used for this setup. Please be either more detailed about your setup (describe and add fiber-optic cores of the cable being connected to the DTS machine in text and Fig.2) or never mention this option. Otherwise it confuses the reader.

- p9 l4: Sorry if I missed it, but did you explain what "duplexed FO core" means?

- p9 l4-11: I would suggest in this paragraph to describe your setup by differentiating about the used number of FO cores increasing the measurement signal instead of writing "x2" or "x1". Further, in this paragraph the effect of bending the fiber is mentioned. So again Hilgersom et al. 2016 could be worth being mentioned instead of just stating that bends can cause signal loss. Besides, if a bend caused the signal loss why is only one FO core affected? Or are both affected and did combining them made the measurments worse?

- p9 l17-19: I would mention again that $n_{space}$=5 for the 90deg angle.

- p9 l14-17: I would not introduce the error prediction function in this paragraph. This will make it easier for the reader to follow your manuscript. How can $C_{DTS}$ be independent of DTS machine and settings? I highly doubt that as also explained earlier (p3 l23).

- p9 l23-24: "However, in reality the wind will not always have a 90deg angle..." $\rightarrow$ I agree that for a horizontal setup this is true but in a vertical harp the horizontal wind will have an attack angle of 90deg! It depends on the physical setup of the fiber.

- p10 Eq13: isn't it $\sum_{i=1}^{n}$ and the fraction $\frac{1}{n}$ or $\frac{1}{n-1}$? Further, $\sigma_p$ is defined here by $u_{DTS}$, but later in Eq.20 $u_N$ is inserted instead of $u_{DTS}$. Is this an inconsistency?

- p111 l1-2: Is that the only discussion you provide for the different attack angles? I think that the results of the use of the direction sensitivity formula needs to be described and discussed.

- p11 l3-6:
  "...is not yet fully calibrated..." $\rightarrow$ "...is not yet applied to the 30-s averages..."
  a range is given for the coefficients of determinations, slope, and intercept: what are the ranges for? Attack angles? Averaging? What is the averaging? What is $n$?
  Units are missing for the intercept
  In the abstract coefficients of determination are given: please also specify in the abstract on which setting those are derived or pick the best one and describe it fully. Otherwise those are just high numbers. Do not be overpromising.
  The coefficient of determination is high, but the intercept as well as the slope shows that there is a systematic underestimation (slope less than one). Why are the intercepts negative? Are they ranging from -0.7 to -0.6(ms$^{-1}$, I guess) or from -0.7 to 0.6? This needs to be discussed.
  "Finally, as expected, ...": this sentence seems out of place. If it is connected to Figure B2 I suggest to transfer the sentence to line 3. I would also change "wind speed angle" to "attack angle" as defined earlier in the manuscript.

- p11 l9: $\sigma_a$ is a dependent on the averaging time,..." $\rightarrow$ "As can be seen in Figure 4, $\sigma_a$ depends on the spatial and temporal averaging of the FO data. The averaging time $n_{time}$ is defined as..."

- p11 l10-11: you mention that $\sigma_a$ also depends on $n_{space}$ but this is not shown in your manuscript. Only plot showing different temporal averaging is shown. It needs at least to be mentioned that this was tested but it is not shown.

- p11 l14: I think $\sigma_a$ is so low because the data is averaged over $n = 10$ spatial points. Accordingly, when averaging over 30s this is already averaging over $n = 300$. Please at least mention again that $n = 10$. Otherwise it is again overpromising. Is the directional sensitivity adjusted for $\sigma_a$? This would explain the negative $\sigma_a$ as Fig.3b shows that for high wind speeds an overestimation of $u_{DTS}$ can be seen when applying the directional sensitivity.

- p12 Figure4: I recommend adding the second axis showing $n$

- p12 l4-8: The paragraph needs substantial revision. Especially the first two sentences I do not understand. What a natural variability? what was considered about it? Does it have influence on $\sigma_p$? Or is the inconsistency in the wind speed of the wind tunnel small enough so it does not matter for $\sigma_p$? The paragraph also further describes Eq. 13 so I suggest to move it to the same section

- p12 l12-15: This sentence seems out of place and the variables are already defined somewhere else. I would delete it.

- p13 Figure5: I would also add $n$ as a second axis.

- p13 Eq.15 & 16: Those equations seem weird to me as a dependency does not develop with the introduction of other variables in an equation:
  As can be seen in Figure 5, $\sigma_p$ is a function of $\Delta T$ and $n$, so $\sigma_p(\Delta T, n)$. So the question is if $\sigma_p$ can also be derived by the signal-to-noise ratio $\left(\frac{\Delta T}{T_{error}}\right)$ and the spatial and temporal averaging $n$:
  $\sigma_p(\Delta T, n) = \frac{\Delta T}{T_{error}} \frac{1}{\sqrt{n}}$
  As $\Delta T$ depends on the heating rate $P_s$ as well as on the mean wind speed the fiber "sees" $u_N$, $\Delta T$ can be reformulated:
  $\Delta T = \frac{A P_s}{B u_N^m}$
  $\Rightarrow \sigma_p(\Delta T, n) = \frac{B T_{error} u_N^m}{A P_s \sqrt{n}}$
  I suggest to use $T_{error}$ computed by the mean bias in the calibration bath and insert that into the above mentioned equation (without the use of $C_{DTS}$) and compare that to your suggested solution. I think $T_{error}$ is in the range of 0.7K to 0.9K which is a bit lower than $C_{DTS} * T_{error} = 1.25K$ of your manuscript, but as also seen in your manuscript those numbers are overestimating $\sigma_p$ (Figure 8). I think this is the most physical way to describe and derive $\sigma_p$.

- p15 Eq20 & 21: $u_n^m \rightarrow u_N^m$

- p15 l16: it is not shown or further mentionend that $\sigma_p$ also depends on $n_{space}$. Please provide corresponding graphs or describe in a view sentences if this was tested but is not shown.

- p15 Section "Verification of the precision prediction": This section needs to be explained and elaborated further. Why is twice the standard deviation used? Not just once? Why not using 90%- and 10%-percentiles? Is the 98%-percentile the same as twice the standard deviation? Figure 8 has many lines in it. Maybe using shading instead of several lines could be considered. To me it looks like $\sigma_p$ is overestimated by the prediction function. The points may lay in the 98% bound, but on the lower end. So I would doubt the general applicability.

- p16 Figure9: Where is this Figure described and discussed?

- p17 l23-26: I think it might be valuable to use a sonic anemometer to determine the attack angles. But depending on the wind field which can be very variable within canopies, within undulating terrain, even within a few meters. Directional sensitivity compensation can only be applied if the angle of attack is known demanding ancillary measurement devices. Please add. So I do not fully agree that wind speed measurements of horizontally put FO cables can always be fully corrected. I would rather recommend to string

the FO cables vertically, hence, no correction is needed as the attack angle is always perpendicular.

---

## Referee Report (RR4)

**Review report: Wind speed measurements using distributed fiber optics: a wind tunnel study**

**Author of the paper: van Ramshorst et al.**
**Journal: Atmospheric Measurement Techniques**
**Manuscript DOI: 10.5194/amt-2019-63**

**General Comments**

The study of van Ramshorst et al. investigated the actively heated fiber-optic (AHFO) technique and estimated its accuracy and precision under controlled airflow conditions by comparing to a three-dimensional ultrasonic anemometer. A valuable error prediction equation for the wind speed measurements at different heating rates was developed, as the heating rate can be a limiting factor for long cables. This equation is also accounting for averaging over space or time which further increases precision. They conclude that AHFO measurements are reliable in outdoor deployments when correcting the measurements for directional sensitivity with a ultrasonic anemometer, choosing the right heating rate and spatial or temporal averaging.

Distributed temperature sensing (DTS) measures temperatures along a fiber-optic cable spatially continuously and can be used in various fields. Especially for atmospheric research this technique offers new insight into the temperature field and thus was implemented in many studies. By using the AHFO technique, wind speed measurements can be added to the system. As the community using the DTS and AHFO technique is growing, the study of van Ramshorst et al. is important for users to be aware of the accuracy, precision and limitation of this technique. The paper is very valuable for our community and I would like to see the manuscript being published.

After a view rounds of review is still feel that a view issues are not addressed: 1) statements which needs further context for the reader & 2) Checking all equations for consistency and correctness.

I recommend to have another person check the manuscript and accept the submitted manuscript after major revisions.

**Detailed comments**

- p1 l9: a high correlation coefficient is presented. However, this correlation is based on correcting the wind speed measurements by the angle of attack. Without knowing the angle of attack the wind speed measurements by FODS perform by far not as good. I think this is a crucial point, especially in the varying wind field near the surface/within canopies/within the whole boundary layer. Depending on the setup, it is very hard to have enough reference devices to know the attack angle and then correct for it. Accordingly, I think the statement in p1 l9 should at least be reformulated and the reader pointed to that a correction for the attack angle was applied.

- p18 l12-14 two publications are mentioned giving an alternative to having multiple ultrasonic anemometer station along the fiber-optic setup. But to my knowledge Zeeman et al. 2015 only provides feature tracking which does not necessarily give the wind direction within the corresponding air masses (which is also stated in the publication under Section 3.1.2). While the outcome of the publication of Lapo et al 2020 is that FODS might be used to determine wind direction at some point, but field studies have to prove that and what features can actually be resolved by it. In this stage I would not present it as done by the authors.

- p3l9-11 the authors say that sensible heat flux can be estimated, however, there is no existing study proving that. Naming this and also the already mentioned publications is not incorrect, but I think they should be put in a better context.

- The mathematical correctness of Eq. 15-18 and how they are developed needs to be reviewed. I do not know the use of an intermediate constant, but maybe this is a mathematical derivation I am not aware of. As the authors show, the numbers do estimate $\sigma_p$ in a fairly good way, but the mathematical presentation of the derivation of the intermediate constants seems fuzzy to me. I would like another person to have a look on this.

- Equation 14 is introduced later than Equation 12 and 13, even though Equation 14 is used to determine the parameters derived in Equation 12 and 13. It would be more reader friendly to introduce Equation 14 together with Equation 11.

- Eq.21: As $\sigma_p$ is derived by using the corrected wind speeds $u_{DTS}$, I think Eq. 21 is incorrect: $u_{DTS}$ is used to derive $\sigma_p$, however, Eq. 19-20 use $u_N$ and then insert this into Eq.21. As stated in Equation 11 and 14 $u_N! = u_{DTS}$ and thus the derivation of Eq. 21 from Eq.20 is not correct. Even if the difference between $u_N$ and $u_{DTS}$ is only a factor, this needs to be mentioned and discussed in the text. Also, as $\sigma_p$ is derived for $u_{DTS}$ it is not justified in my opinion to say that the prediction function is then still true for perpendicular flow as the derivation is mostly based on corrected data.

- small editing comment: I think the definition of $n_{time}$ and $n_{space}$ was dropped in the most recent manuscript, but should be added. I am sorry if I over read the definition of those parameter.

**Detailed comments on manuscript after revision 4**

The following comments were not addressed

- p3 l8-9: as already mentioned above: how can you derrive the sensible heat flux from DTS + AHFO measurements
  $\Rightarrow$ even though this might be true, until there is no study I think it is a vague statement and should be reformulated or put in better context.

- p8 l6: duplexed FO core: was this splice checked for a step loss? ; p8 l11-12: so only offset correction of the FO cable was performed? Was the differential attenuation of the FO cores checked and accounted for?; p9 l1-2: "However, in processing of the raw DTS data...." $\rightarrow$ "But in our setup the signal loss of the splice connecting the fiber-optic cores of our cable at the end of the array was not the same in both directions." - Did you introduce earlier that two cores were spliced together to create a duplexed setup?; p9 l2-3:

"Due to this assymetrical structure..." $\rightarrow$ I think it was never introduced that potentially two channels can be used for this setup. Please be either more detailed about your setup (describe and add fiber-optic cores of the cable being connected to the DTS machine in text and Fig.2) or never mention this option. Otherwise it confuses the reader.

$\Rightarrow$ I think this still needs clarification and how the calibration was done. Single-ended, single-ended duplexed or double ended calibration? Hausner et al 2011 presents those three options. Maybe one paragraph specifically addressing calibration is beneficial instead of single sentences hinting to the calibration setup.

- p10 Eq13: isn't it $\sum_{i=1}^{n}$ and the fraction $\frac{1}{n}$ or $\frac{1}{n-1}$? Further, $\sigma_p$ is defined here by $u_{DTS}$, but later in Eq.20 $u_N$ is inserted instead of $u_{DTS}$.
  $\Rightarrow$ The authors responded that it is correct to as the only difference between $u_{DTS}$ and $u_N$ is a factor, however, I think this does justify inserting $u_N$ in Eq.20 instead of $u_{DTS}$. This clearly needs to be mentioned in the text and discussed (as also mentioned above).

- In the abstract coefficients of determination are given: please also specify in the abstract on which setting those are derived or pick the best one and describe it fully. Otherwise those are just high numbers.
  $\Rightarrow$ this is still not adjusted
  The coefficient of determination is high, but the intercept as well as the slope shows that there is a systematic underestimation (slope less than one). Why are the intercepts negative? Are they ranging from -0.7 to -0.6($\mathrm{ms}^{-1}$, I guess) or from -0.7 to 0.6? This needs to be discussed.
  $\Rightarrow$ as the coefficients are mentioned in the abstract I think the manuscript needs some discussion of the results in addition to the plots in the appendix.

- p11 l10-11: you mention that $\sigma_a$ also depends on $n_{space}$ but this is not shown in your manuscript. Only plot showing different temporal averaging is shown. It needs at least to be mentioned that this was tested but it is not shown.
  $\Rightarrow$ I do not think it is wrong that spatial averaging will influence $_a$, however, it is not shown. In my opinion it should be tested and then at least mentioned in the text. In Figure 4 the change of $\sigma_a$ is shown for increasing $n_{time}$ increasing the total $n$ while $n_{space}$ is kept constant. The difference between attack angles can not be used to show that spatial averaging does have an impact on $\sigma_a$. This should at least be mentioned in the text that similar behaviour is expected when increasing $n_{space}$ while $n_{time}$ is kept constant.

- p15 l16: it is not shown or further mentioned that $\sigma_p$ also depends on $n_{space}$. Please provide corresponding graphs or describe in a view sentences if this was tested but is not shown.
  $\Rightarrow$ same comment as above. I think it is only shown that $\sigma_p$ changes with temporal averaging while spatial averaging is kept constant.

- p13 Eq.15 & 16: Those equations seem weird to me as a dependency does not develop with the introduction of other variables in an equation:
  $\Rightarrow$ also see my comments in the first section.

- p17 l23-26: I think it might be valuable to use a sonic anemometer to determine the attack angles. But depending on the wind field which can be very variable within canopies, within undulating terrain, even within a few meters. Directional sensitivity compensation can only be applied if the angle of attack is known demanding ancillary measurement

devices.

⇒ see comment above. It is not easy to correct for attack angles and to have enough reference stations which should be mentioned accordingly for future users.

---

## Author Response (AR2)

**Point-by-point reply from van Ramshorst et al. (2019) to:**
**Review report: Wind speed measurements using distributed fiber optics: a wind tunnel study**

Author of the paper: van Ramshorst et al.
Journal: Atmospheric Measurement Techniques
Manuscript DOI: 10.5194/amt-2019-63

**General Comments**

The study of van Ramshorst et al. investigated the actively heated fiber-optic (AHFO) technique and estimated its accuracy and precision under controlled airflow conditions by comparing to a three dimensional ultrasonic anemometer. A valuable error prediction equation for the wind speed measurements at different heating rates was developed, as the heating rate can be a limiting factor for long cables. This equation is also accounting for averaging over space or time which further increases precision. They conclude that AHFO measurements are reliable in outdoor deployments when correcting the measurements for directional sensitivity with a ultrasonic anemometer, choosing the right heating rate and spatial or temporal averaging. Distributed temperature sensing (DTS) measures temperatures along a fiber-optic cable spatially continuously and can be used in various fields. Especially for atmospheric research this technique offers new insight into the temperature field and thus was implemented in many studies. By using the AHFO technique, wind speed measurements can be added to the system. As the community using the DTS and AHFO technique is growing, the study of van Ramshorst et al. is important for users to be aware of the accuracy, precision and limitation of this technique. Hence, the paper is valuable for our community.
The author did do a good job on the first revision. The reviewed manuscript nicely rearranged the sections and made the manuscript more reader friendly. Also the additional turbulence statistics of the wind tunnel makes it a stronger manuscript. However, there are still some issues regarding precise description of the setup, definition of parameter, presentation of some graphs, and the error prediction function. Details are given below for each section. I recommend to accept the submitted manuscript after major revisions.

*Thanks you for the positive feedback, we will give a point-by-point answers to your comments and questions below.*

**Terminology**

- advective heat loss vs. convective heat loss: the correct terminology as is also used in the referenced literature of this manuscript is "convective".

*We will change this accordingly.*

**Abstract & Section 1**

- p.1, l.2-3: "...better characterization of fine-scale processes." - this is a very specific comment, but I think sonic anemometers can determine fine-scale processes better in time (20 Hz resolution vs. 1 Hz of DTS), while AHFO technique does have spatially continuously distributed measurements. Hence, AHFO can give inside into the wind field even a network of sonic anemometer might not be capable of on time scales > 1s. But I argue that a general statement like "better characterization" might not justify if not pointing out that the focus is on the spatial scale.

*We agree that the benefits of the AHFO are mostly spatial, therefore we added the word spatial.*

- p.1, l.4: "In this work, ..." this sentence is redundant

*Agree, we removed the sentence*

- p.1, l.6: "wind speed, angles of attack, and temperature differences" ! Oxford comma!

*We changed this*

- p.1, l.7: 1-s time scale

*We changed this*

- p.1, l.9: the correlation numbers can only be found in the abstract and conclusions, but not in the results section. Please refer to that correlation numbers in the result section, otherwise those high correlation numbers seem to come out of nowhere.

*It is mentioned in the Accuracy and precision section on page 11, line 10 (old manuscript)*

- p.1, l.11-12: "AHFO allows for characterization...complex terrain..." - AHFO can be deployed in any terrain, so I do not understand the focus on complex terrain. I would remove that sentence and only mention the last sentence which is way more important.

*We removed the complex terrain part.*

- p.2, l.3: "Goodberlet et al. (1989)" - aren't there also more recent paper commenting on that?

*We added another more recent source*

- p.2, l.16: 1-s and 0.3-m resolution

*We changed this*

- p.2, l.21-22: I do not understand the meaning of that sentence

*We simplified the sentence*

- p.2, l.25-27: the statements in those lines are very specific and belong into material and methods

*We here describe the previous work of Sayde et al. and not our own set-up. Hence, we see this as an introduction part of our paper.*

- p.2, l.29: magnitude of convective heat loss

*We changed this*

- p.2, l.31-25: This paragraph can be shortened: "The heat transfer model assumes a flow normal to the fiber. Hence, non-normal angles of attack need to be accounted for by using directional sensitivity equations. Following the recommendations of Sayde et al. (2015) we tested different directional sensitivity equations from hotwire anemometry in he controlled setting of our experiments" - or similar

*We changed this sentence accordingly.*

- p.3, l.7: remove "in complex terrain" - AHFO can be deployed in any terrain

*We changed this*

- p.3, l.13: "precision of future experiments" - precision of AHFO experiments

*We changed this*

**Section 2.1 & 2.2**

Both sections can be shortened by mentioning important literature and specifically focus on the parts which were changed in this manuscript to improve AHFO measurements. Section 2.2.1 has no new content and Figure 1 is from the paper of Sayde et al. (2015) altered in an incorrect way. The presented equations are also more or less a copy of Sayde et al. (2015). The most important changes are mentioned in Section 2.2.2, hence I would argue to remove Section 2.2.1 or only mention the most important points (like the energy balance) instead of the complete derivation of the equation determining the wind speed from AHFO measurements as already done by Sayde et al. (2015).

*Section 2.2.1 contains indeed no new information and is a summary of the work of Sayde et al. (2015). We are already concise and we only introduce the parts we will change, as described in section 2.2.2. Accordingly, we think without this part the paper gets less readable. Also we changed advection to convection in figure 1.*

I also have the following specific comments for those two sections:

- p.3, l.17-25: I think this paragraph is unnecessary for the manuscript focusing on AHFO. One sentence and pointing to the publication of Selker et al. 2006 is enough to explain the basic measurement principle of DTS. The last two sentences of the paragraph can be inserted somewhere else.

*We agree and shorten this paragraph*

- p.4, l.1: "heat the fiber" - heat the FO cable

*We changed this*

- p.4, l.2-3: "Also the creation of ..." - very confusing statement

*We changed this*

- p.4, l.6: "which van be" - which can be

*We changed this*

- p.4, l.10; p.5, l.5; p.5, l.13; p.5, l.14; ...: "advective" - "convective"

*We changed this*

**Section 2.3**

The whole section needs to be revised. Some information is unnecessary and could be addressed in one paragraph instead of mentioning it in different paragraphs. For example p.7, l.17-23 define the heating rates $P_s$ creating $\Delta T$ for different wind speeds, however, the actually used $\Delta T$ are mentioned p.8 l.8 and the corresponding $P_s$ p.9, l.1. Also the total number of experiments could be mentioned at the end after defining/mentioning all setups instead of consecutively summing up the experiment setups (p.8 l.2, l.6, l.8). Further, in this section FOs is used as a synonym for FO cable or FO cores, however, was not defined as such. The FO configuration as also introduced/proposed by Hausner et al. (2011) is introduced in p.8, l.10-11, then changed to a double-ended configuration (p.9, l.12), but actually a single-ended configuration (p.9, l.12) was used. This is more than confusing to the reader especially by mentioning it in different paragraphs. I would propose to mention the setup, only the actually chosen calibration method (as proposed by Hausner et al. (2011)), and step loss correction in one paragraph.

*We revised the whole section and made it more concise according to your suggestions. We clarified if we mean the FO cable or FO core. We now only mention the single ended configuration.*

I also have the following comments and questions:

- p.7, l.17-23: This paragraph should be revised. It is confusing to the reader. The main point should be the definition of the heating rate $P_s$ fixing $\Delta T$ for different wind speeds. Accordingly, $\Delta T$ is representing $P_s$.

*We changed this*

- p.7, l.17: "The angle of the fiber..., wind speed, and heating rate were systematically changed" - This sentence is fine, however you start defining heating rate, then the angle of attack and then your wind speed settings... A reorganization makes it more reader friendly.

*We changed this*

- p.7, l.26-28: I am not sure if this sentence is important unless you add that those parts were excluded from analysis to avoid artifacts of the fiber touching the mounting material.

*We think this should be part of the explanation of our set-up*

- p.8, l.2: "The AHFO wind speed measurements can be calibrated..." – calibrated is not the correct use here, as DTS data itself is also calibrated. I suggest "adjusted" or "verified". Also removing that sentence would not affect the meaning of the paragraph.

*We changed this*

- p.8, l.10: The manufacturer of the FO cable is missing.

*We added this*

- Figure 3: The figure is not necessary for understanding the described method, Figure 2 is already showing the needed information. Either remove or put into Appendix.

*We added this as an appendix*

- p.9, l.1-6: this paragraph is describing the heating rate and how it can be estimated from the resistance of the FO cable. However, the heating rate is first mentioned p.7, l.22. I suggest to reorganize the section and to shorten this paragraph. The most important information is in the last sentence.

*We changed this*

- p.9, l.16. "splice loss": was a step loss correction performed?

*Yes, according to Hausner et al. (2011).*

**Section 2.4 & Section 3.1**

I do not completely agree with p.10, l.8-10. Sayde et al. (2015) never adjusted DTS measurements, they adjusted the sonic anemometer measurements and compared them to the AHFO measurements. Hence, the statement is not completely correct. However, I agree that the equation can be used to adjust for different attack angles and adjust/correct AHFO measurements if a sonic anemometer is near the AHFO setup. Figure 4 should be adjusted by showing violin plots or boxplots for each wind speed configuration. The shown dots cannot show the distribution of the DTS measurements in a clear way. Are real measurements of the sonics shown or the proposed fixed wind speeds?

*We agree and rewrote the sentence about Sayde's correction method.*

*We changed figure 4 to boxplots. In our old plots the real measurements of the sonic was used, however in a boxplot this is not possible, and the proposed fixed wind speeds are used.*

**Section 3.2**
The definition of $\sigma_a$ and $\sigma_p$ should be moved into material and methods. The result that $\sigma_a$ and $\sigma_p$ are dependent on wind speed and averaging over space and time and the corresponding discussion can remain in Section 3.2.

*We added this to the method and a new short section.*

On page 11 line 1-6 present details on the duplex FO configuration and that only the middle of the FO cable was used to estimate $u_{DTS}$, because otherwise the accuracy is decreased (Table C1). I think that this important piece of information, especially about the 90° attack angle, needs to be mentioned in Section 2.3. Further, shouldn't a step loss correction cover the effects of a splice?

*We added this to the method. The reason we don't use part of the data is because of an asymmetrical splice loss. A step loss correction could correct also for the channel which looks wrong, however we do not want to introduce extra uncertainties.*

I think Figure A1 and A2 are unnecessary as Figure 5 is giving the same information and derive the same conclusion. If the author decides to include those figures, they need to be adjusted in the same way as Figure 4.

*We changed Figure A1 and A2 to boxplots.*

Eq. 13 and 14 are presented in a confusing fashion by introducing dependencies which do not affect the definition of the introduced parameter. Eq. 13 and 14 define $\sigma_a$ and $\sigma_p$, however, the definition itself is not reflecting any dependency on $n_{space}$ nor $n_{time}$. This dependency is shown in Fig. 5 and Fig. 6 and should not be included in Eq. 13 and 14. The original equation from the first manuscript were better. Besides, the parameter $u_j$ remains unclear to me as it also doesn't define $\sigma_a$ and $\sigma_p$. Both are defined by $u_{DTS}$ and $u_{sonic}$. I think $u_j$ is introducing an unnecessary parameter. Further, Eq. 15 is introduced, which is changed to Eq. 16 without justification and both are different.

*We agree and removed $n_{space}$ and $n_{time}$. Also we changed the notation of Eq. 15 and Eq. 16. Furthermore, we changed uj to j, meaning a specific wind speed setting (1,3,5,7,9,11,13,15,16,17 m/s)*

- p.12, l.12-14: those lines are redundant as they are already introduced on p.11, l.12-17

*We agree and changed this*

- p. 13, l.1-3: I do not see the justification of excluding the variation of sonic anemometer measurements by assuming it is small. If it was tested and did not change the results, I can see the justification of not considering that variation.

*We show in appendix B (old manuscript) that there is low turbulence in the wind tunnel, therefore this term will be negligible, as our turbulence statistics are calculated with the sonic anemometer data.*

**Section 3.3 and Section 3.4**

The title of Section 3.3 is not representing the content. The use of an intermediate constant is still not justified. The authors response to my comment did not include a justification nor the manuscript. The intermediate constant is defined as:

$$C_{int} = \overline{f(\bar{u}_j, n_{space}, n_{time}) * \frac{\Delta T}{T_{error}} * \sqrt{n_{time}}} \tag{1}$$

Hence, $C_{int}$ (or $C_{DTS} = C_{int} * 10$) is the mean of the precision over a variation of setup:

$$C_{int} = \overline{\sigma_p * \frac{\Delta T}{T_{error}} * \sqrt{n_{time}}} \tag{2}$$

Those constants are then included in the estimation of $\sigma_p$ again which is not justified. Besides, how was the factor $T_{error}$ determined? Or is this number from literature? The revised manuscript is describing the derivation of the intermediate constant very quickly without further explanation. This should be presented in further detail. I still propose to derive the error prediction function in a more clear way. As shown in Fig. 6 for each $\Delta T$ the precision $\sigma_p$ is following a $\frac{1}{\sqrt{n}}$-line. Hence, we assume the following dependency:

$$\sigma_p(n) = \frac{\alpha}{\sqrt{n}} \tag{3}$$

with $\alpha$ being a constant different for experiment set up. We found that $\alpha$ depends on $\Delta T$ and $T_{error}$:

$$\alpha = (\frac{\Delta T}{T_{error}})^{-1} \tag{4}$$

with $T_{error}$ = 0.25 K being the performance of the DTS dependent constant and $\Delta T$ being the measured temperature difference between the cables. Hence, $\alpha$ is representing the quality factor for the wind speed measurements. The lines derived from $\alpha$ could also be added in Fig. 6 or 7. When simplifying Eq. 20 of the submitted manuscript we can assume that $\Delta T$ is mainly depending on the following parameter:

$$\Delta T = \frac{AP_s}{Bu_n^m} \tag{5}$$

Combining my Eq. 4 and 5 and inserting that in Eq. 3, I derive the following error prediction equation:

$$\sigma_p(n, u_n, P_s) = \frac{BT_{error}u_n^m}{AP_s}\frac{1}{\sqrt{n}} \qquad (6)$$

If I did not miss a point, no empirically derived intermediate constant has to be used for the error prediction equation.
I think the error prediction function could be tested with the existing data set by inserting $P_s$ in the error prediction function and plotting that against the 'real' $\sigma_p$ using Eq. 14 of the revised manuscript. This should show if $C_{DTS}$ is needed or not.

*We changed the title into Normalized precision independent of sampling settings, this represents the content of the section. $T_{error}$ is determined from the machine specifications (page 14, line 8).*

*We understand your confusion, however we think our method is justified and the difference lies in Eq. 3 from you review. You state that σp is only dependent on the sampling n, however σp is also dependent on the wind speed (u). As can be seen in our eq. 14 (old manuscript). To confirm this with numbers we made some short calculations shown below, which shows that our CDTS factor makes the σp (for $n_{time}$ = 30 sec) prediction more similar to the calculated observations from figure 6 (old manuscript). Therefore, we conclude that our method is correct and justified.*

*Table 1: Comparing van Ramshorst et al. and review report way of calculating σ$_p$, for comparison with Figure 6 (old manuscript); Values calculated for $n_{time}$ = 30.*

| Angle | ΔT | σ$_p$ - van Ramshorst et al.; eq. 22 | σ$_p$ - Review report; eq. 6 |
|-------|-----|------------------------------------|-----------------------------|
| 90°   | 6   | 0,0162                             | 0,0032                      |
| 45°   | 4   | 0,0190                             | 0,0038                      |
| 30°   | 2   | 0,0418                             | 0,0084                      |
| 15°   | 4   | 0,0345                             | 0,0069                      |

**Section 3.5 & conclusions & Appendix**

Section 3.5 is not introducing new content. It should be incorporated in the corresponding sections as a paragraph of discussion or incorporated into the conclusions if the statement is more an outreach than a finding.

*Since section 3 is results and discussion we think it is appropriate to have a separate section which puts the results in perspective and gives a short outreach.*

- p.17, l.19-20: how can turbulence be fully captured by the AHFO technique? Which turbulence scales are you talking about? How should the setup look like?

*We clarified this, we meant larger scale turbulence depending on your spatial and temporal sampling resolution, but at least > 1s and > 0.3m.*

- p.18, l.6-7: "Due to the way this design tool is constructed, it can be generalized for all kinds of fibers, DTS precisions, and user preferred spatial and temporal resolution." I do not agree with this statement, because the accuracy and precision of the DTS measurements change with the use of FO cable, DTS performance, and the used calibration method. Further, the turbulence of the wind tunnel setup does not represent outdoor turbulence as also stated in this manuscript. Another point is the response time of the FO cable. The thicker the cable, the longer it takes to reach the FO core and measure the temperature change. Also the measurement location of the wind speed is important, as the noise of measurement increases with distance from the measurement device, hence with the location along the FO cable. So how should C$_{DTS}$ and the error prediction function be representative for outdoor deployments, different choice of FO cable, different setup of FO cable, or a different DTS machine? I think the statement is too strong and not justified.

*We partly agree and downscaled our statement a bit. However, we also partly disagree and in our opinion the following things are included:*
- *the thickness of the cable is in parameter A and B; eq. 22 (old manuscript)*
- *The DTS performance, calibration and cable length can be adjusted with T$_{error}$*

*We agree that the turbulence is not included, although we should take into account that this is a first estimate, not a fully describing function of σ$_p$.*

- p.18, l.11: "...applications in complex terrain, allowing for..." -...applications, for example allowing for... - AHFO can be deployed in any terrain.

*We changed this*

- Figure B2: turbulence intensity should be defined in the caption, as well as the location of the x-, y- and z-coordinate.

*We clarified this in the appendix*

[revised manuscript text omitted]

---

## Author Response (AR3)

**Point-by-point reply from van Ramshorst et al. (2019) to:**
**Review report: Wind speed measurements using distributed fiber optics: a wind tunnel study**

Author of the paper: van Ramshorst et al.
Journal: Atmospheric Measurement Techniques
Manuscript DOI: 10.5194/amt-2019-63

**General Comments**

The study of van Ramshorst et al. investigated the actively heated fiber-optic (AHFO) technique and estimated its accuracy and precision under controlled airflow conditions by comparing to a three-dimensional ultrasonic anemometer. A valuable error prediction equation for the wind speed measurements at different heating rates was developed, as the heating rate can be a limiting factor for long cables. This equation is also accounting for averaging over space or time which further increases precision. They conclude that AHFO measurements are reliable in outdoor deployments when correcting the measurements for directional sensitivity with an ultrasonic anemometer, choosing the right heating rate and spatial or temporal averaging.

Distributed temperature sensing (DTS) measures temperatures along a fiber-optic cable spatially continuously and can be used in various fields. Especially for atmospheric research this technique offers new insight into the temperature field and thus was implemented in many studies. By using the AHFO technique, wind speed measurements can be added to the system. As the community using the DTS and AHFO technique is growing, the study of van Ramshorst et al. is important for users to be aware of the accuracy, precision and limitation of this technique. Hence, the paper is valuable for our community.

The manuscript improved substantially. It is well organized and leads the reader through the whole manuscript. I still have one major point: The manuscript proposes to develop an error prediction function being valid for any kind of setup. However, the error prediction function is not tested or validated with the existing data set nor is the last point discussed accordingly. I did get a table in the authors' response to compare different approaches of the error prediction function, however, this table is not well explained. Further, no values derived from the prediction function is compared to actually measured quantities neither in the table nor in the manuscript. As the error prediction function is one main goal of the manuscript, either the authors need to explicitly state that the error prediction function needs to be validated in another experiment or another section validating the error prediction function is added to the manuscript. I recommend to accept the submitted manuscript after major revisions. More detailed comments are given below.

*Thanks you for the positive feedback, we will give a point-by-point answers to your comments and questions below.*

**Detailed comments**

- p3 l23: Terror: measurement error? how determined?

*The measurement error is defined by the machine specifications. We added a line to clarify.*

- p3 l27: choose dominant or important or be more specific

*We meant dominant, and changed this.*

- p8 l18-31: nicely done!

*Thanks*

- p9 l4: "duplexing" -> duplexed FO core (I would stick with the earlier already mentioned phrase)

*We changed this accordingly.*

- p9 l6-8: the argumentation to treat the 90° angle different than the others is a bit thin in my eyes: maybe speculate why the 90° angle had lower precision (sharper bending of the FO cable maybe?) and thus justify your decision. Or treat all attack angles the same and shorten all data down. Why should the splice only affect the 90° angle? What if the others were also affected just a bit less?

*We clarified possible causes in the manuscript by mentioning the sharper bend and the effect of fixation of the cable. The splice is a different issue and we don't argue that this causes the difference.*

- p9 l12-14: "indicating and..." -> "indicating an"; How is the actual spatial resolution defined? Nyquist-frequency?

*We use the 10-90% rule to determine the spatial resolution as described in: Tyler, S. W., J. S. Selker, M. B. Hausner, C. E. Hatch, T. Torgersen, C. E. Thodal, and S. G. Schladow (2009), Environmental temperature sensing using Raman spectra DTS fiber-optic methods, Water Resour. Res.,45, W00D23, doi:10.1029/2008WR007052. We clarified this in the paper.*

- p9 l14-17: I think mentioning the goal of an error prediction function is more useful than already mentioning the unique constant which is used in the error prediction function later. This will make it easier for the reader to follow your manuscript. Especially the last sentence is confusing to me. Is one constant more representative if I am averaging, but if I am not averaging, it isn't?

*We clarified this in the manuscript. We mean that by averaging it becomes more accurate, we changed this accordingly.*

- Figure 3, B1, B2: a 1:1-line would be very helpful

*We agree and added this to the figures*

- Figure B2 and p11 l1-5: It would be easier if B2 has two figures a) 1-s data and b) 30-s data. It is impossible for the reader to combine all four plots in B1 into one plot and compare it to Figure B2.

*We agree and created a figure B2 a and b.*

- Figure 4: Why is accuracy of 90° & ΔT = 2K getting worse for higher averaging time? I would suspect the opposite as also proposed by the manuscript (p11 l16-17). I would like that this is at least mentioned/discussed.

*First of all, we need to clarify that a bias of 0, not necessary means it is perfect. This would assume the energy balance is perfect, however this is not the case as mentioned in the paper. This "error" is most likely different for each angle and heating rate. Secondly all the measurements show the same behavior, they all "drop" for times higher than 1. We see a bigger drop if the ΔT is smaller, which relates to the signal to noise ratio. Finally, the drop for the 90° measurement is bigger because we only have 5 instead of 10 measurements, which doubles the drop in the bias. This shows the precision also has an effect on the bias.*

*Summarizing:*

$$\sigma_a = \frac{(\bar{u}_{DTS}(j) \pm \varepsilon) - \bar{u}_{sonic}(j)}{\bar{u}_{sonic}(j)}$$

*With ε being dependent on the precision and error in the energy balance.*

- p11 l4: coefficients of determination: I guess that is a linear regression and you show the R-values? What is the derived slope and offset? Please also add this information or at least add the 1:1-lines to the graphs if slope is close to unity anyway.

*We plotted the 1:1 line in the figures. Also we clarified the slope and offset in the manuscript and the values are indeed the $r^2$ (coefficient of determination).*

- p11 l16-17: You need to discuss this statement further and use another phrase for "extensive calibration" which is not accurate as different calibration methods can be applied to the FO measurements before even computing wind speeds. I would also argue that maybe the temperature signal needs to be averaged over time before computing the FO wind speed. Would that also increase the accuracy? If not, why? Was this also tested?

*We rephrased this sentence into further analysis. Further analysis can be done on improving the energy balance. Considering your second suggestion by averaging over time, this is how we did it.*

- p11 l18: I would argue that the dependence between accuracy and averaging time is less pronounced than the one between the precision and averaging time scale, not that the accuracy is constant over time. Besides, it is confusing that σa is given in percent while σp is given in decimal numbers, but percentage values are given in the text. Please make uniform for both parameters.

*We changed the sentence into fairly constant. Also we changed all σ's to decimal numbers.*

- p11 l20: The last sentence is redundant. Either comment in further detail what Eq. 13 is stating and what dependencies can be determined from Figure 5 or remove the sentence.

*We removed this sentence*

- p12 l1-3: The meaning of those sentences for the analysis is hard to understand. Further, is j used instead of the measured usonic?

*We clarified these sentences by:*

*For the calculation of the precision $u_{DTS}$, we considered the natural variability of the wind. We assumed that this natural variability is measured by the sonic and we assume that this per definition is smaller than the variability of the DTS. After applying eq. 13 the variability of the DTS machine $u_{DTS}$ are obtained.*

*j indicates the wind speed setting, while calculating the actual measured wind speeds are used.*

- p12 l6: rephrase: "the precision was averaged over all wind speeds which is justified, because σp is normalized by the mean wind speed, hence any linear dependency should be removed" or similar; "... for all ΔT..." -> "... for each ΔT..."
- p12 l7: "..., with ..." The sentence is redundant to some degree. The colors and symbols are already showing why there are 12 different points for each ntime.

*We rephrased these two sentences into:*

*The precision was averaged over all wind speeds for each ΔT and angle combinations in Figure 5, which is justified because σp is normalized by the mean wind speed, hence any linear dependency should be removed.*

- The following is only a suggestion/thought: What about dropping the attack angles for σp? They are not further discussed as you already account for them in the earlier section. So should that maybe not be considered moving on? It is kind of distracting from the main object of different heating rates and averaging time/spatial scales.

*We think it is valuable to show the variability between the angles.*

- Figure 6a: y-axis label is not representing what is actually plotted: $\sigma_p \frac{\Delta T}{T_{error}}$

*We changed this.*

- p13 l10-15: If this statement is true, then it needs to be further discussed and why it can be applied to different settings. The statement is also referring to an Equation which is introduced later in the manuscript. So I suggest to insert this paragraph at the corresponding location to Eq. 21 and into Section 3.4, respectively. Also Cint has a wide spread and needs to be further discussed. Again I suggest to insert a plot using the prediction function for σp and the actual derived σp for all averaging scales to show the strength and accuracy of the prediction function.
- Figure 6b: same y-axis problem as Fig. 6a. Further, how can you justify that your proposed constant has a spread from 1.1 to 2.2? This needs to be mentioned and discussed.

*We created a new subsection in Section 3.4 showing the validation of our prediction equation. Additionally, we also discuss the sensitivity of the constant $C_{int}$. The following (subsection) is added to the manuscript:*

*"**Verification of the precision prediction section:***

*For verification purposes the calculated precision (Eq. 15) is combined with the predicted precision (Eq. 21) in Figure 8. As can be seen in Figure 8 we underestimate the precision of the AHFO system using $C_{int}$=1.6, meaning that better performance can be expected. This difference can be explained by three causes. First, in Figure 6b it is visible that the 90° data is only averaged over 5 data points instead of 10, resulting in a higher $C_{int}$. Second, we see the effect of ΔT: how higher the heating the less spread in the precision distribution (Figure 5). Third, we neglect the smaller energy terms in Eq. 19, which leads to an increased σp. To investigate the sensitivity of using a constant $C_{int}$, the 98% confidence bounds (two times standard deviation) of $C_{int}$ are determined. It is projected (dotted lines) in Figure 8 that the calculated precision is within the 98% confidence interval of the predicted precision. Concluding, with our prediction equation we can predict all our settings within a 98% confidence interval, showing the general applicability."*

- p14 l6-7: This is a contradiction as Figure 5 is showing the exact opposite: Higher σp for lower ΔT, lower σp with increasing ntime. σp can be estimated from those variables, but σp is not independent of those.

*We understand the confusion; our sentence was wrongly stated. We clarified this in the manuscript by: "... behaves similar for each…." instead of independent.*

- Section 3.5: Maybe dew fall on the fiber needs to be considered? Water droplet on the fiber will for sure affect the measurements altering the heat loss of the unheated fiber (assuming the water droplets quickly evaporate from the heated fiber).

*We agree and added a part about wet fibers.*

- p17 l10: directional sensitivity compensation can only be applied if the angle of attack is known demanding ancillary measurement devices. Please add.

*We added this.*

[revised manuscript text omitted]
 8 we underestimate the precision of the AHFO system using $C_{int} = 1.6$, meaning that better performance can be expected. This difference can be explained by three causes. First, in Figure 6b it is visible that the 90° data is only averaged

5   over 5 data points instead of 10, resulting in a higher $C_{int}$. Second, we see the effect of $\Delta T$: how higher the heating the less spread in the precision distribution (Figure 5). Third, we neglect the smaller energy terms in Eq. 19, which leads to an increased $\sigma_p$. To investigate the sensitivity of using a constant $C_{int}$, the 98% confidence bounds (two times standard deviation) of $C_{int}$ are determined. It is projected (dotted lines) in Figure 8 that the calculated precision is within the 98% confidence interval of the predicted precision. 
[revised manuscript text omitted]

---

## Author Response (AR4)

**Point-by-point reply from van Ramshorst et al. (2019) to:**
**Review report: Wind speed measurements using distributed fiber optics: a wind tunnel study; March 4, 2020**

Author of the paper: van Ramshorst et al.
Journal: Atmospheric Measurement Techniques
Manuscript DOI: 10.5194/amt-2019-63

**General Comments**

The study of van Ramshorst et al. investigated the actively heated fiber-optic (AHFO) technique and estimated its accuracy and precision under controlled airflow conditions by comparing to a three-dimensional ultrasonic anemometer. A valuable error prediction equation for the wind speed measurements at different heating rates was developed, as the heating rate can be a limiting factor for long cables. This equation is also accounting for averaging over space or time which further increases precision. They conclude that AHFO measurements are reliable in outdoor deployments when correcting the measurements for directional sensitivity with an ultrasonic anemometer, choosing the right heating rate and spatial or temporal averaging. Distributed temperature sensing (DTS) measures temperatures along a fiber-optic cable spatially continuously and can be used in various fields. Especially for atmospheric research this technique offers new insight into the temperature field and thus was implemented in many studies. By using the AHFO technique, wind speed measurements can be added to the system. As the community using the DTS and AHFO technique is growing, the study of van Ramshorst et al. is important for users to be aware of the accuracy, precision and limitation of this technique. The paper is very valuable for our community and I would like to see the manuscript being published.

The manuscript again improved substantially. Especially Figure 8 was the needed piece for validating the proposed prediction function. I have one major point: $T_{error}$ is an important value for the error prediction function, but is barely described. The number is given by the manufacturer but it is not specified if a calibration and which spatial and temporal resolution was applied. Accordingly, $T_{error}$ is not a universal number and can only be used if the setup of the manufacturer is known. I recommend to either ask the manufacturer or derive $T_{error}$ by the bias within the calibration bath which is a temperature controlled environment. I think that the error prediction function is only correct if $T_{error}$ is defined correctly. Further, I have three minor points: 1) the dependency between $\sigma_p$ and $n_{space}$ is not shown -> I would recommend to add a second x-axis on Figure 4, Figure 5 and Figure 6 showing $n = n_{time} * n_{space}$; 2) Checking all equations for consistency and correctness; 3) Careful language: differentiate between what the paper can offer and what are potential future steps.

I am not a native speaker so I am not sure about the commas and other grammatical errors of the manuscript. I do not want to discourage the author by the amount of comments and number of reviews I demand. The manuscript is really improving. I recommend to accept the submitted manuscript after major revisions. I think that not another major revision is needed afterwards. More detailed comments are given below.

*Thank you for the positive feedback, we will give point-by-point answers to your comments and questions below.*

**Detailed comments**

- p1 l11-13: AHFO can measure the mean horizontal wind speed but not the horizontal wind speed. As the horizontal components are usual an order of magnitude stronger, it is so far not possible to measure the vertical wind speed component. At least I did not see a publication doing this so far. I also did not see this being done in your publication. So promising that AHFO combined with DTS can derive a turbulent heat flux estimation is promising more than can be done so far especially for reader which are new to this measurement technique.
- p3 l8-9: as already mentioned above: how can you derive the sensible heat flux from DTS + AHFO measurements

*We indeed don't measure the vertical wind speed in our paper. Using Monin-Obukhov similarity it is theoretically possible to estimate the sensible heat flux by using the vertical profile of the mean horizontal wind speed and temperature. We added Businger et al. (1971) as a source on page 3 for clarity, however, the possibility with DTS still has to be tested.*

- p2 l10: the "as well" can be taken out.

*We removed this*

- p2 l11: spatially

*We changed this*

- p2 l11-13: The underlying assumptions of what?

*We changed this sentence and added: underlying **spatial** assumptions. With DTS/AHFO one is able to get high resolution spatial data, which allows for direct measurements of for example temperature and horizontal wind profiles. Also seepage in rivers can be identified and shown.*

- p2 l21: I would remove "Recently"

*We changed this into: In 2015,*

- p2 l21-29: I like this paragraph very much.

*Thanks!*

- p3 l6: spatially

*We changed this*

- p3 l15: the abbreviation DTS was already introduced earlier

*We changed this*

- p3 l15-19: just a thought (does not need to be included): Isn't bending also a source of signal loss for the fiber? Further, when averaging over space is needed coiling up the fiber could potentially be useful. Hilgersom et al. 2016 "Practical considerations for enhanced-resolution coil-wrapped distributed temperature sensing". But maybe this is opening a new topic and is a bit too far off topic of this manuscript.

*We added a sentence about sharp bends and added Hilgersom et al. (2016).*

- p3 l23: Terror: machine specifications: How did the manufacturer define this? If this is not clear you cannot use this number for your calculations! I think it is better to define Terror by the mean bias within one of the calibration baths using the proposed averaging over time or space which might be more fair than a constant number independent of spatial and temporal averaging of your measurements. If Ts-Tf is lower than the bias within the calibration baths then this can also be considered a low signal or high σp. If Terror is defined this way, maybe the instantaneous constants are not necessary and your function could be applied to any setup.
- p9 l14-17: I would not introduce the error prediction function in this paragraph. This will make it easier for the reader to follow your manuscript. How can CDTS be independent of DTS machine and settings? I highly doubt that as also explained earlier (p3 l23).

*We understand the confusion and added a complete new paragraph on how we calculated $T_{error}$. In this part we mention how we calculated $T_{error}$. We also explain how $T_{error}$ can be calculated for other studies, which is necessary to use our prediction equation.*

*"DTS temperature measurements contain a measurement error, which follows a normal distribution Selker et al. 2006a). With long FO cables this measurement error changes over the length of the cable and this error is also different for each DTS machine. In this experiment a short FO cable is used, which is close to the calibration bath. Therefore, the measurement error is calculated based on the calibration baths, by taking the average of two baths where the mean standard deviation over the whole experiment is calculated. Given the fact the signal used ΔT, containing the difference of two temperature measurements of $T_s$ and $T_f$, $T_{error}$ becomes: $T_{error} = \sqrt{\sigma_{Ts}^2 + \sigma_{Tf}^2}$. In this experiment $\sigma_{Ts} = \sigma_{Tf}$, resulting in $T_{error} = \sigma_T * \sqrt{2}$. In this experiment we used a single value, however in experiments with longer FO cables, one could calculate a $T_{error}$ changing along the cable (Des Tombe and Schilperoort (2020)). "*

*We also understand the confusion about C_DTS being independent and changed this sentence. C_DTS is indeed a value based on our experiment. However, with calculating C_DTS we try to generalize the precision of the DTS based on the used ΔT, angles and the number of n. This single value can then be used in Eq. 21 to predict the precision for independent other experiments. C_DTS is a fixed number, however in Eq. 21 we use the generalization in combination with experiment specific parameters ($T_{error}$ for example) to be able to give predictions for all kinds of DTS machines and settings.*

- p3 l26-27: review this sentence. It contains "can" twice and "can cause that other ways of..." also sounds not logical to me.

*We changed the sentence*

- p3 l26: "radiation" -> "radiative heat loss,"

*Changed*

- p3 l30: the measurement error can only be compensated when using another device. Please mention this. With a varying wind field or within a canopy a lot of reference devices might be necessary.

*Added*

- p4 Figure 1: This is the figure of Sayde et al. (2015)!"...balance, based on..." -> "... balance from..."

*Changed*

- p5 l9-10: The Nusselt number is defined incorrect: ratio of convective to conductive heat transfer.

*Changed*

- p5 l20: the unit of the temperature range is incorrect. I think it is degree Celsius, not Kelvin.

*Good catch, changed to ℃.*

- p6 l1: I would start a new paragraph here

*Changed*

- p6 l9: "...we assume that there is a uniform..." -> uniform in space or in time? In time could be verified by the ultrasonic anemometer data

*We refer here to 'in space'. In time the temperature is not constant. In space is added.*

- p7 l7: Is there a documentation of that wind tunnel somewhere? Or a webpage to get further information?

*We have no documentation. We added the name of the wind tunnel, so information can be found online or the people in charge of the wind tunnel can be contacted.*

*http://research.engr.oregonstate.edu/liburdygroup/Facilities*

- p7 l9-10: "... two segments of one cable (which encloses the FO cores) were placed 8cm apart..." -> this formulated a bit confusing. "During the experiment the heated and the unheated reference cable segment were placed 8cm apart. The FO cable has two FO cores, hence, each cable segment could be sampled twice."

*Changed*

- p8 l6: duplexed FO core: was this splice checked for a step loss?
- p8 l11-12: so only offset correction of the FO cable was performed? Was the differential attenuation of the FO cores checked and accounted for?

*Yes, that is one of the reasons we found out about the asymmetrical splice loss.*

*We performed the complete calibration method of Hausner et al. (2011) using the CTEMPs package (https://ctemps.org/data-processing). So this includes also the differential attenuation.*

- p8 l18-31: I like this presentation a lot, however, I think the bullet points could still be shortened by not using full sentences making the amount of experiments even more clear.

*We were satisfied with your suggestion to make those bullet points, and think that the sentences clearly explain our experiments.*

- p8 l27: Outside deployment: definitely turbulent conditions -> please mention one sentence why this is not an issue comparing the wind tunnel experiments with outside deployments.

*This is indeed a good point, however we think this is not the right section to discuss this and would only confuse the reader. We have a whole section (3.5) discussing this and other issues for outdoor measurements.*

- p8 l33: "Splices between ends of fiber optic cables..." -> "Splices connecting two fiber optic cores..."

*Changed*

- p9 l1-2: "However, in processing of the raw DTS data...." -> "But in our setup the signal loss of the splice connecting the fiber-optic cores of our cable at the end of the array was not the same in both directions." - Did you introduce earlier that two cores were spliced together to create a duplexed setup?
- p9 l4: Sorry if I missed it, but did you explain what "duplexed FO core" means?

*We changed the sentence as suggested and indeed we spliced two cores together and clarified this on page 7.*

- p9 l2-3: "Due to this asymmetrical structure..." -> I think it was never introduced that potentially two channels can be used for this setup. Please be either more detailed about your setup (describe and add fiber-optic cores of the cable being connected to the DTS machine in text and Fig.2) or never mention this option. Otherwise it confuses the reader.

*We added a sentence on page 7, where is stated that we connected both FO cores, however we used a single-ended configuration.*

*"Both the FO cores were connected to a Silixa Ultima DTS machine (Ultima S, 2 km range, Silixa, London, UK) outside the wind tunnel, however afterwards a single-ended configuration was used due to asymmetrical signal loss."*

- p9 l4-11: I would suggest in this paragraph to describe your setup by differentiating about the used number of FO cores increasing the measurement signal instead of writing "x2" or "x1". Further, in this paragraph the effect of bending the fiber is mentioned. So again Hilgersom et al. 2016 could be worth being mentioned instead of just stating that bends can cause signal loss. Besides, if a bend caused the signal loss why is only one FO core affected? Or are both affected and did combining them made the measurements worse?

*We added a table in the appendix (D2) where we explain how many measurements are taken.*

*We added the paper of Hilgersom et al. (2016) as a reference. Furthermore, both cores are affected, one more than the other, therefore combining all measurements made things worse. We use the best measurements from both cores, the ones right in the middle, as the edge effect is bigger for the 90° setup.*

p9 l17-19: I would mention again that nspace=5 for the 90deg angle.

*We added a table to the appendix for clarity and added nspace=5 for the 90deg angle.*

- p9 l23-24: "However, in reality the wind will not always have a 90deg angle..." -> I agree that for a horizontal setup this is true but in a vertical harp the horizontal wind will have an attack angle of 90deg! It depends on the physical setup of the fiber.

*We changed the sentence*

- p10 Eq13: isn't it $\sum_{i=1}^{n}$ and the fraction $\frac{1}{n}$ or $\frac{1}{n-1}$? Further, $\sigma_p$ is defined here by $u_{DTS}$, but later in Eq.20 $u_N$ is inserted instead of $u_{DTS}$. Is this an inconsistency?
- p15 Eq20 & 21: $u_n^m \rightarrow u_N^m$

*Good remark and we changed the symbol inconsistency.*

*We changed the sum sign of Eq. 13.*

*We understand the confusion. First we made sure all $u_N$ have a capital N as it should. Second, we indeed use $U_{DTS}$ in eq. 13, however this is not an inconsistency, as the difference between $U_{DTS}(i,j) - \bar{U}_{DTS}(j) = U_N(i,j) - \bar{U}_N(j)$ because, the measurements for each $\sigma_p$ (Fig.5) are converted with the same factor (Eq.14).*

- p11 l1-2: Is that the only discussion you provide for the different attack angles? I think that the results of the use of the direction sensitivity formula needs to be described and discussed.

*We added some discussion stating the results are satisfying for all four angles.*

- p11 l3-6: "...is not yet fully calibrated..." -> "...is not yet applied to the 30-s averages..."

*We refer to Eq. 14 in this sentence. We made this clearer.*

a range is given for the coefficients of determinations, slope, and intercept: what are the ranges for? Attack angles? Averaging? What is the averaging? What is n?

*In Figure B1 you can now find the detailed slope, intercept and $R^2$ per setup, as well as the n. In the text we only summarize the results.*

Units are missing for the intercept.

*Added*

In the abstract coefficients of determination are given: please also specify in the abstract on which setting those are derived or pick the best one and describe it fully. Otherwise those are just high numbers. Do not be overpromising.

*We clarified this*

The coefficient of determination is high, but the intercept as well as the slope shows that there is a systematic underestimation (slope less than one). Why are the intercepts negative? Are they ranging from -0.7 to -0.6(ms-1, I guess) or from -0.7 to 0.6? This needs to be discussed.

*For the 15° angle there is a negative intercept, which is compensated by the higher slope. The rest are positive intercepts (see B1).*

"Finally, as expected, ...": this sentence seems out of place. If it is connected to Figure B2 I suggest to transfer the sentence to line 3. I would also change "wind speed angle" to "attack angle" as defined earlier in the manuscript.

*We rewrote this sentence.*

- p11 l9: σa is a dependent on the averaging time,..." -> "As can be seen in Figure 4, σa depends on the spatial and temporal averaging of the FO data. The averaging time ntime is defined as..."

*We changed the sentence*

- p11 l10-11: you mention that σa also depends on nspace but this is not shown in your manuscript. Only plot showing different temporal averaging is shown. It needs at least to be mentioned that this was tested but it is not shown.

*In the next paragraph we extended the discussion of Figure 4 and described the effect of $n_{space}$.*

*"For the data set (n=5-300), the maximum $σ_a$ is ± 0.03, which is promising for future applications. The ΔT = 6 K should be the best performing heating setting, however this is not always the case and there are fluctuations between the heating settings, which could be due to neglecting small energy losses, like free convection due to heating of air close to the heated cable (Sayde et al. (2015)), which is temperature dependent. With such an energy loss included, the bias of each angle might change. Nevertheless, the*

*bias is fairly constant after n=50 with increasing averaging time, which means further analysis can probably increase the accuracy. The change in bias from n=5 to n=50 is due to the precision of our AHFO measurements, which increases with averaging over longer time (n increases) and is higher for a greater $\Delta T$. This difference is bigger for the 90° cases, as $n_{space}=5$ instead of $n_{space}=10$ for the other angles, indicating that spatial averaging also has an effect on the bias."*

- p11 l14: I think σa is so low because the data is averaged over n = 10 spatial points. Accordingly, when averaging over 30s this is already averaging over n = 300. Please at least mention again that n = 10. Otherwise it is again overpromising. Is the directional sensitivity adjusted for σa? This would explain the negative σa as Fig.3b shows that for high wind speeds an overestimation of uDTS can be seen when applying the directional sensitivity.

*We added (n = 5-300). And yes, the new directional sensitivity equation (Eq. 14) is used to calculate σa.*

- p12 Figure4: I recommend adding the second axis showing n

*We changed the x-axis of Figure 4 to n. Thanks for this suggestion.*

- p12 l4-8: The paragraph needs substantial revision. Especially the first two sentences I do not understand. What a natural variability? what was considered about it? Does it have influence on σp? Or is the inconsistency in the wind speed of the wind tunnel small enough so it does not matter for σp? The paragraph also further describes Eq. 13 so I suggest to move it to the same section.

*With natural variability we meant the variability of the wind speed. And indeed we meant to say that this variability is small enough so that it does not matter for σp. We changed the text accordingly.*

- p12 l12-15: This sentence seems out of place and the variables are already defined somewhere else. I would delete it.

*Yes, we agree and removed the sentence.*

- p13 Figure5: I would also add n as a second axis.

*We changed the x-axis of Figure 5 to n. Thanks for this suggestion.*

- p13 Eq.15 & 16: Those equations seem weird to me as a dependency does not develop with the introduction of other variables in an equation:
  As can be seen in Figure 5, $\sigma_p$ is a function of $\Delta T$ and n, so $\sigma_p(\Delta T, n)$. So the question is if $\sigma_p$ can also be derived by the signal-to-noise ratio $\frac{\Delta T}{T_{error}}$ and the spatial and temporal averaging n:

  $$\sigma_p(\Delta T, n) = \frac{\Delta T}{T_{error}} \frac{1}{\sqrt{n}}$$

  As $\Delta T$ depends on the heating rate Ps as well as on the mean wind speed the fiber "sees" $u_N$, $\Delta T$ can be reformulated:

  $$\Delta T = \frac{AP_s}{Bu_n^m}$$

  $$\rightarrow \sigma_p(\Delta T, n) = \frac{Bu_n^m T_{error}}{AP_s \sqrt{n}}$$

I suggest to use Terror computed by the mean bias in the calibration bath and insert that into the above mentioned equation (without the use of $C_{DTS}$) and compare that to your suggested solution. I think Terror is in the range of 0.7K to 0.9K which is a bit lower than $C_{DTS}$ * $T_{error}$ = 1.25K of your manuscript, but as also seen in your manuscript those numbers are overestimating $\sigma_p$ (Figure 8). I think this is the most physical way to describe and derive $\sigma_p$.

*As already discussed in review round 2. $\sigma_p$ is also dependent on the wind speed (j), and this dependency is missing from your first equation. $\Delta T$ and $\sigma_p$ both have the wind speed as a dependency. Furthermore, Figure 8 shows our equation works (new manuscript). The suggested equation by you would be too low, as $T_{error}$ is 0.32 and $C_{DTS}$ is 3.57.*

- p15 l16: it is not shown or further mentioned that σp also depends on nspace. Please provide corresponding graphs or describe in a view sentences if this was tested but is not shown.

*The effect of $n_{space}$ can be seen in Figure 5. Also the $\frac{1}{\sqrt{n}}$ doesn't differentiate between $n_{space}$ or $n_{time}$. Finally, n is defined by $n_{space} \times n_{time}$.*

- p15 Section "Verification of the precision prediction": This section needs to be explained and elaborated further. Why is twice the standard deviation used? Not just once? Why not using 90%- and 10%-percentiles? Is the 98%-percentile the same as twice the standard deviation? Figure 8 has many lines in it. Maybe using shading instead of several lines could be considered. To me it looks like σp is overestimated by the prediction function. The points may lay in the 98% bound, but on the lower end. So I would doubt the general applicability.

*We created a new Figure 8 with shading and once the standard deviation. Also we investigated the overestimation and found out why this was happening. We rewrote the whole section, which clarifies earlier errors.*

*New text paragraph:*

*"For verification purposes the calculated precision (Eq. 13) is combined with the predicted precision (Eq. 21) in Figure 8.  As can be seen in Figure 8, the precision of the AHFO system is estimated well and the one time standard deviation covers all calculated precisions. When using Eq. 21 one should consider $U_N$ is derived for a 90° angle. If wind speeds with other angles of attack are expected, one should use Eq. 14. $U_N$ is the measured wind speed normal to the FO cable and the measured wind speed is lower in case of an angle < 90°. In this case one should use $U_N = U_{DTS} \times cos(\varphi - 90°)^{m1}$. Concluding, with our prediction equation we can predict all our settings within a one standard deviation interval, showing general applicability."*

- p16 Figure9: Where is this Figure described and discussed?

*We don't understand this question. We don't have a Figure 9.*

- p17 l23-26: I think it might be valuable to use a sonic anemometer to determine the attack angles. But depending on the wind field which can be very variable within canopies, within undulating terrain, even within a few meters. Directional sensitivity compensation can only be applied if the angle of attack is known demanding ancillary measurement devices. Please add. So I do not fully agree that wind speed measurements of horizontally put FO cables can always be fully corrected. I would rather recommend to string the FO cables vertically, hence, no correction is needed as the attack angle is always perpendicular.

*We made some changes in this paragraph and added Lapo et al. (2020) as a reference.*

[revised manuscript text omitted]
 90° ~~data is only averaged over 5 data points instead of 10, resulting in a higher $C_{int}$. Second, we see the effect of $\Delta T$: how higher the heating the less spread in the precision distribution (Figure 5). Third, we neglect the smaller energy terms in Eq. 19, which leads to an increased $\sigma_p$. To investigate the sensitivity of using a constant $C_{int}$, the 98% confidence bounds (two times standard deviation) of $C_{int}$ are determined. It is projected (dotted lines) in Figure 8 that the calculated precision is within the 98% confidence interval of the predicted precision.98% confidencethe~~ general applicability.

[revised manuscript text omitted]

---

## Author Response (AR5)

**Point-by-point reply from van Ramshorst et al. (2019) to:**
**Review report: Wind speed measurements using distributed fiber optics: a wind tunnel study; May 11, 2020**

Author of the paper: van Ramshorst et al.
Journal: Atmospheric Measurement Techniques
Manuscript DOI: 10.5194/amt-2019-63

**General Comments**

The study of van Ramshorst et al. investigated the actively heated fiber-optic (AHFO) technique and estimated its accuracy and precision under controlled airflow conditions by comparing to a three-dimensional ultrasonic anemometer. A valuable error prediction equation for the wind speed measurements at different heating rates was developed, as the heating rate can be a limiting factor for long cables. This equation is also accounting for averaging over space or time which further increases precision. They conclude that AHFO measurements are reliable in outdoor deployments when correcting the measurements for directional sensitivity with an ultrasonic anemometer, choosing the right heating rate and spatial or temporal averaging. Distributed temperature sensing (DTS) measures temperatures along a fiber-optic cable spatially continuously and can be used in various fields. Especially for atmospheric research this technique offers new insight into the temperature field and thus was implemented in many studies. By using the AHFO technique, wind speed measurements can be added to the system. As the community using the DTS and AHFO technique is growing, the study of van Ramshorst et al. is important for users to be aware of the accuracy, precision and limitation of this technique. The paper is very valuable for our community and I would like to see the manuscript being published. After a view rounds of review, I still feel that a view issues are not addressed: 1) statements which needs further context for the reader & 2) Checking all equations for consistency and correctness. I recommend to have another person check the manuscript and accept the submitted manuscript after major revisions.

*Thank you for the feedback, we hope that in this point-by-point reply we can answer the last concerns raised.*

**Detailed comments**

- p1 l9: a high correlation coefficient is presented. However, this correlation is based on correcting the wind speed measurements by the angle of attack. Without knowing the angle of attack the wind speed measurements by FODS perform by far not as good. I think this is a crucial point, especially in the varying wind field near the surface/within canopies/within the whole boundary layer. Depending on the setup, it is very hard to have enough reference devices to know the attack angle and then correct for it. Accordingly, I think the statement in p1 l9 should at least be reformulated and the reader pointed to that a correction for the attack angle was applied.

*We added that these correlation coefficients are obtained after correcting for the angle of attack:*

*"The AHFO measurements are compared to sonic anemometer measurements and show a high coefficient of determination (0.92-0.96) for all individual angles, after correction the AHFO measurements for the angle of attack."*

- p18 l12-14 two publications are mentioned giving an alternative to having multiple ultrasonic anemometer station along the fiber-optic setup. But to my knowledge Zeeman et al. 2015 only provides feature tracking which does not necessarily give the wind direction within the corresponding air masses (which is also stated in the publication under Section 3.1.2). While the outcome of the publication of Lapo et al 2020 is that FODS might be used to determine wind direction at some point, but field studies have to prove that and what features can actually be resolved by it. In this stage I would not present it as done by the authors.

*Thank you for pointing this out. We agree that still work has to be done to measure the angle of attack under all conditions. Therefore, we changed the text accordingly:*

*"However, in complex terrains as for example inside canopies, one ancillary device could be not enough due to the high variability of the wind field. In such a case, a more complex 3D set-up of DTS/AHFO (Zeeman et al. (2015)) could be an indication of the angle of attack. Also recently, a new method is under development which tries to measure the angle of attack with a single cable, using microstructures attached to the fiber (Lapo et al. (2020))."*

- p3 l9-11 the authors say that sensible heat flux can be estimated, however, there is no existing study proving that. Naming this and also the already mentioned publications is not incorrect, but I think they should be put in a better context.

*We agree this is not being published yet, however, we think that when the wind and temperature profile is known, it is theoretically possible that one can estimate the sensible heat flux under the assumptions of Monin-Obukhov similarity theory. To clarify, we added that the Monin-Obukhov similarity theory should be used following the method of Businger 1971. With the citation to Businger 1971, one can get all the details which are needed:*

*"Moreover, the ability to measure spatial varying wind fields has the potential to be useful for estimating sensible heat fluxes in a variety of atmosphere-vegetation-soil continuums, by applying Monin-Obukhov similarity theory (assuming no violation of its assumptions) to the measured vertical profile of the mean wind speed and temperature (Businger 1971)"*

- The mathematical correctness of Eq. 15-18 and how they are developed needs to be reviewed. I do not know the use of an intermediate constant, but maybe this is a mathematical derivation I am not aware of. As the authors show, the numbers do estimate $\sigma_p$ in a fairly good way, but the mathematical presentation of the derivation of the intermediate constants seems fuzzy to me. I would like another person to have a look on this.

*We are confident that our mathematical derivation is correct and appropriate, which was verified by the editor.*

- Equation 14 is introduced later than Equation 12 and 13, even though Equation 14 is used to determine the parameters derived in Equation 12 and 13. It would be more reader friendly to introduce Equation 14 together with Equation 11.

*It is true that in the end we proposed Equation 14 as the directional sensitivity equation, however the derivation of Equation 14 is part of our results section (based on your recommendation for dividing old and new work). Therefore, Equation 14 is shown in the next chapter as this is a new addition compared to the "original method" by Sayde et al. (2015).*

- Eq.21: As $\sigma_p$ is derived by using the corrected wind speeds $u_{DTS}$, I think Eq. 21 is incorrect: $u_{DTS}$ is used to derive $\sigma_p$, however, Eq. 19-20 use $u_N$ and then insert this into Eq.21. As stated in Equation 11 and 14 $u_N! = u_{DTS}$ and thus the derivation of Eq. 21 from Eq.20 is not correct. Even if the difference between $u_N$ and $u_{DTS}$ is only a factor, this needs to be mentioned and discussed in the text. Also, as $\sigma_p$ is derived for $u_{DTS}$ it is not justified in my opinion to say that the prediction function is then still true for perpendicular flow as the derivation is mostly based on corrected data.

*As our aim is to predict the precision of AHFO compared to a sonic, we used the corrected $u_{DTS}$ data instead of $u_n$ since the wind is not always at an angle of 90 degrees. Additionally, this approach is also consistent with our bias calculation. Furthermore, the difference between $u_n(i,j)-u_n(j)$ and $U_{DTS}(i,j)-U_{DTS}(j)$ is minimal, especially considering our non-turbulent flow conditions inside the wind tunnel. We think that our prediction function is also valid for perpendicular flow, as our data set to derive $C_{DTS}$ consists of perpendicular measurements, also it is clear in Figure 8 that the perpendicular measurements are within one standard deviation of $C_{DTS}$.*

*To clarify we added the following sentence:*

*"Knowing this expression of ΔT, Eq. 18 can again be rewritten into Eq. 21 (assuming the difference between $u_n(i,j)-u_n(j)$ and $U_{DTS}(i,j)-U_{DTS}(j)$ is negligible), which expresses the precision estimate, with $P_s$ as only parameter which can be changed during an experiment."*

*Furthermore, we noticed a typing error in L21. Instead of referring to equation 19, this should have been equation 18. We corrected this as well.*

- small editing comment: I think the definition of $n_{time}$ and $n_{space}$ was dropped in the most recent manuscript, but should be added. I am sorry if I over read the definition of those parameter.

*The definition of $n_{time}$ and $n_{space}$ can be found on page 9 line 23-26.*

**Detailed comments on manuscript after revision 4**
*The following comments were not addressed*

- p3 l8-9: as already mentioned above: how can you derive the sensible heat flux from DTS + AHFO measurements -> even though this might be true, until there is no study I think it is a vague statement and should be reformulated or put in better context.

*See our previous comment in this document where we clarified this.*

- p8 l6: duplexed FO core: was this splice checked for a step loss? ; p8 l11-12: so only offset correction of the FO cable was performed? Was the differential attenuation of the FO cores checked and accounted for? ; p9 l1-2: "However, in processing of the raw DTS data...." -> "But in our setup the signal loss of the splice connecting the fiber-optic cores of our cable at the end of the array was not the same in both directions." - Did you introduce earlier that two cores were spliced together to create a duplexed setup? ; p9 l2-3: "Due to this asymmetrical structure..." -> I think it was never introduced that potentially two channels can be used for this setup. Please be either more detailed about your setup (describe and add fiber-optic cores of the cable being connected to the DTS machine in text and Fig.2) or never mention this option. Otherwise it confuses the reader.

*This already answered in the previous point-to-point reply (amt-2019-63-author_response-version4.pdf page 5).*

- I think this still needs clarification and how the calibration was done. Single-ended, single-ended duplexed or double ended calibration? Hausner et al 2011 presents those three options. Maybe one paragraph specifically addressing calibration is beneficial instead of single sentences hinting to the calibration setup.

*We think everything is in the paper to reproduce the used method. Adding a separate paragraph will also not benefit the readability of the paper and is also not the main scope of this paper. To clarify which method of Hausner 2011 we used we added to the following sentence "single-ended".*

*"For calibration and validation of the DTS data, approximately 6 m of the FO cables was placed in a well-mixed ambient bath to calibrate the DTS temperature according to the single-ended method described by Hausner 2011."*

- p10 Eq13: isn't it $\sum_{i=1}^{n}$ and the fraction $\frac{1}{n}$ or $\frac{1}{n-1}$? Further, $\sigma_p$ is defined here by $u_{DTS}$, but later in Eq.20 $u_N$ is inserted instead of $u_{DTS}$) The authors responded that it is correct to as the only difference between $u_{DTS}$ and $u_N$ is a factor, however, I think this does justify inserting $u_N$ in Eq.20 instead of $u_{DTS}$. This clearly needs to be mentioned in the text and discussed (as also mentioned above).

*See our previous comment in this document where we clarified this.*

- In the abstract coefficients of determination are given: please also specify in the abstract on which setting those are derived or pick the best one and describe it fully. Otherwise those are just high numbers. -> this is still not adjusted

*This is now corrected in the abstract.*

The coefficients of determination are high, but the intercept as well as the slope shows that there is a systematic underestimation (slope less than one). Why are the intercepts negative? Are they ranging from -0.7 to -0.6($ms^{-1}$, I guess) or from -0.7 to 0.6? This needs to be discussed. -> as the coefficients are mentioned in the abstract I think the manuscript needs some discussion of the results in addition to the plots in the appendix.

*Negative intercepts are inherent when using linear regression. All slopes and intercepts can be found in Figure B1. We added the Figure B1 reference in the text to emphasize this.*

- p11 l10-11: you mention that $\sigma_a$ also depends on $n_{space}$ but this is not shown in your manuscript. Only plot showing different temporal averaging is shown. It needs at least to be mentioned that this was tested but it is not shown.
  -> I do not think it is wrong that spatial averaging will influence $\sigma_a$, however, it is not shown. In my opinion it should be tested and then at least mentioned in the text. In Figure 4 the change of $\sigma_a$ is shown for increasing $n_{time}$ increasing the total n while $n_{space}$ is kept constant. The difference between attack angles cannot be used to show that spatial averaging does have an impact on $\sigma_a$. This should at least be mentioned in the text that similar behavior is expected when increasing $n_{space}$ while $n_{time}$ is kept constant.

*Clearly our definition of "dependent on" led to miscommunication. In the paper (version amt-2019-63-author_response-version4.pdf) we indeed mention $\sigma_a$ depends on $n_{space}$, as in our experiment we average over multiple measurements in space. We don't necessarily imply that $\sigma_a$ improves, however we do use $n_{space}$, so we need to define this quantity because it has an effect on how $\sigma_a$ is calculated. In the previous manuscript version (version amt-2019-63-author_response-version5.pdf) we already changed this sentence to hopefully prevent miscommunication, stating:*

*"$\sigma_a$ depends on the spatial and temporal averaging of the FO data. The averaging time $n_{time}$ is defined as $n_{time} = t_{avg}/t_{sample}$, where $t_{avg}$ can only be an integer which is a multiple of $t_{sample}$. Spatial averaging is defined as $n_{space} = x_{avg}/x_{sample}$, where $x_{avg}$ can only be an integer which is a multiple of $x_{sample}$."*

- p15 l16: it is not shown or further mentioned that $\sigma_p$ also depends on $n_{space}$. Please provide corresponding graphs or describe in a view sentences if this was tested but is not shown. -> same comment as above. I think it is only shown that $\sigma_p$ changes with temporal averaging while spatial averaging is kept constant.

*Per definition the precision decreases by the relation of $\sqrt{n}$, when the wind speed is the same over time or space (n is defined as $n_{time}$ x $n_{space}$). The effect of $n_{time}$ is mostly shown in most figures, but in Appendix D1 we show the effect of $n_{space}$ in the wind tunnel.*

- p13 Eq.15 & 16: Those equations seem weird to me as a dependency does not develop with the introduction of other variables in an equation: -> also see my comments in the first section.

*See our previous comment in this document where we clarified this.*

- p17 l23-26: I think it might be valuable to use a sonic anemometer to determine the attack angles. But depending on the wind field which can be very variable within canopies, within undulating terrain, even within a few meters. Directional sensitivity compensation can only be applied if the angle of attack is known demanding ancillary measurement devices. -> see comment above. It is not easy to correct for attack angles and to have enough reference stations which should be mentioned accordingly for future users.

*See our previous comment in this document where we clarified this. We repeated this difficulty now as well on page 18, line 23 in the conclusions:*

[revised manuscript text omitted]